# Stress adaptation of mitochondrial protein import by OMA1-mediated degradation of DNAJC15

Lara Kroczek[1,5], Hendrik Nolte[1,5], Yvonne Lasarzewski[1], Ishita Agrawal[1], Thibaut Molinié[2], Daniel Curbelo Piñero[1], Kathrin Lemke[1], Elena Rugarli[2,3,4] & Thomas Langer [1,4] ✉

Mitochondria dynamically adapt to cellular stress to ensure cell survival. The stress-regulated mitochondrial peptidase OMA1 orchestrates these adaptive responses, which limit mitochondrial fusion and promote mitochondrial stress signaling and metabolic rewiring. Here, we show that cellular stress adaptation involves OMA1-mediated regulation of mitochondrial protein import and OXPHOS biogenesis. OMA1 cleaves the mitochondrial chaperone DNAJC15 and promotes its degradation by the m-AAA protease AFG3L2. Loss of DNAJC15 impairs mitochondrial protein import and restricts OXPHOS biogenesis under conditions of mitochondrial dysfunction. Non-imported mitochondrial preproteins accumulate at the endoplasmic reticulum, inducing an unfolded protein response. Our results demonstrate stress-dependent changes in mitochondrial protein import as part of the OMA1-mediated mitochondrial stress response and highlight the interdependence of proteostasis regulation between different organelles.

Mitochondrial stress responses orchestrated by the mitochondrial peptidase OMA1 ensure the adaptation of mitochondrial functions to meet changing physiological demands and maintain cell survival. Mitochondrial deficiencies, such as OXPHOS defects or dissipation of the membrane potential, activate OMA1 in the mitochondrial inner membrane (IM) and trigger the mitochondrial integrated stress response (ISR[mt]) by processing DELE1, which promotes metabolic reprogramming and enhances cellular defense against oxidative stress[1–6]. Simultaneously, OMA1 activation limits mitochondrial fusion through excessive processing of the dynamin-like GTPase OPA1, leading to the fragmentation of the mitochondrial network under stress[7–10]. Recent studies have demonstrated that OMA1 can perform quality-control functions by degrading polypeptides that block protein translocases[11]. The activation of OMA1 destabilizes the protease and causes its degradation, terminating the OMA1-mediated stress response after restoration of mitochondrial functions[6,12]. Although impairment of OMA1-mediated

stress pathways by deletion of *Oma1* or *Dele1*, or by prevention of Opa1 processing, is well-tolerated in mice[8,13–18], protective effects have been observed under pathophysiological conditions, such as mitochondrial (cardio-)myopathies[16,18–20]. However, the consequences of OMA1 deficiency and of impaired OMA1-mediated stress responses are cell- and tissue-specific and depend on the physiological context[14,21–25].

Here, to better understand how OMA1 affects mitochondrial proteostasis and stress responses, we performed a proteomic survey for proteolytic substrates of OMA1. We demonstrate that OMA1 cleaves the mitochondrial chaperone DNAJC15 facilitating its degradation by the mitochondrial m-AAA protease. DNAJC15 loss alters protein import by TIM23 protein translocases in the IM and limits the accumulation of OXPHOS-related mitochondrial matrix and IM proteins. Non-imported mitochondrial preproteins accumulate at the endoplasmic reticulum (ER) and trigger an unfolded protein response. These results demonstrate that OMA1 allows to adapt mitochondrial protein biogenesis to

[1]Max Planck Institute for Biology of Ageing, Cologne, Germany. [2]Institute for Genetics, University of Cologne, Cologne, Germany. [3]Center for Molecular Medicine, University of Cologne, Cologne, Germany. [4]Cologne Excellence Cluster on Cellular Stress Responses in Aging-Associated Diseases (CECAD), University of Cologne, Cologne, Germany. [5]These authors contributed equally: Lara Kroczek, Hendrik Nolte. ✉e-mail: tlanger@age.mpg.de

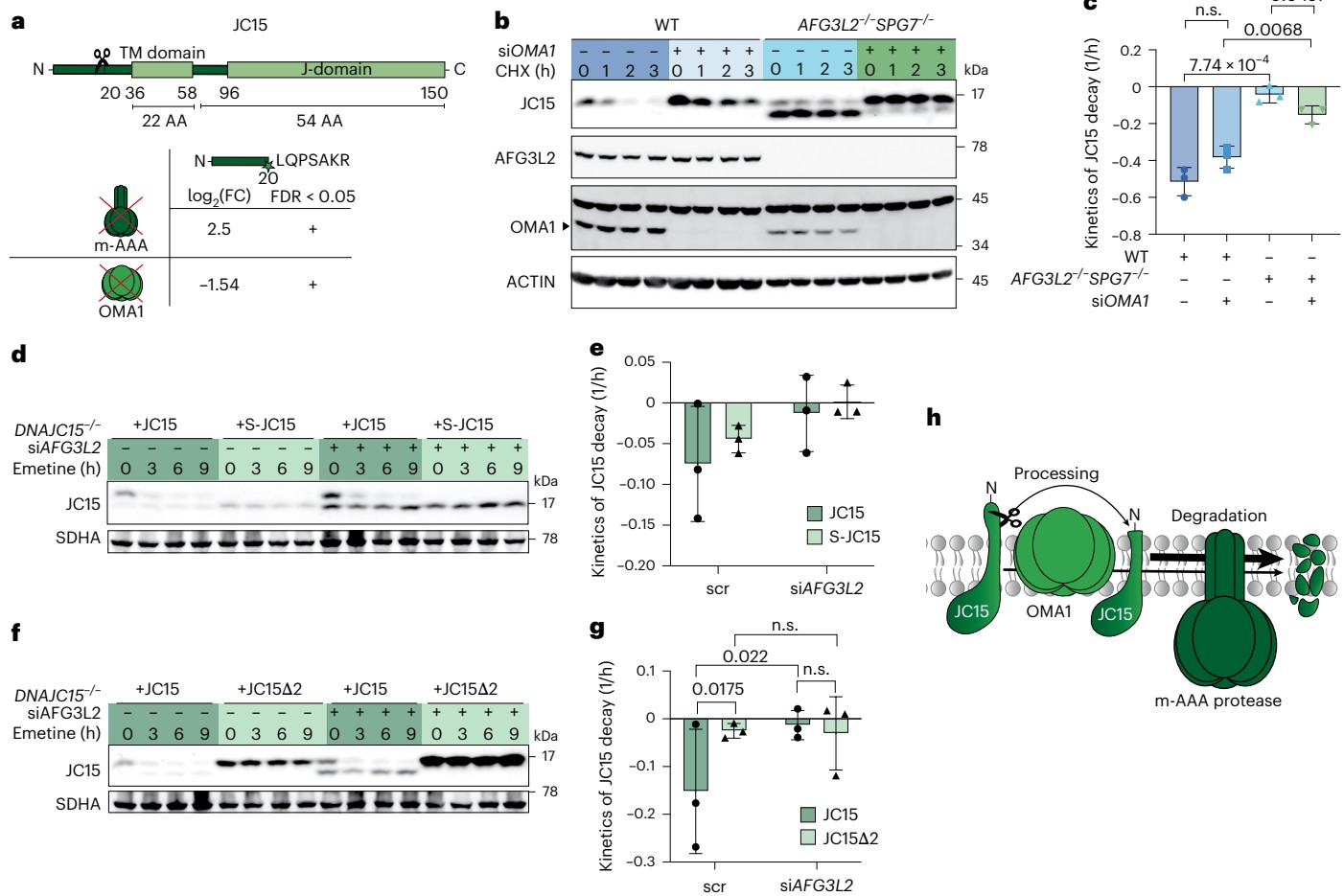

**Fig. 1 | OMA1 cleavage of DNAJC15 facilitates degradation by the m-AAA protease. a**, Comparison of the Neo-N-terminal proteome datasets of WT HeLa cells with *OMA1*[−/−] and *AFG3L2*[−/−]*SPG7*[−/−] cells reveals the OMA1-dependent cleavage of DNAJC15 (JC15) at amino acid position 19 (UniProt ID: Q9Y5T4) (*n* = 5, biologically independent samples). FDR, false discovery rate; FC, fold change. **b**, Stability of DNAJC15 in WT and *AFG3L2*[−/−]*SPG7*[−/−] HeLa cells depleted of OMA1 (si*OMA1*) for 72 h after inhibition of cytosolic translation with cycloheximide (CHX). A representative experiment is shown (*n* = 3, biologically independent samples). **c**, Degradation rates of DNAJC15 in **b**, normalized to time point 0 h. Data are means ± s.d. (*n* = 3, biologically independent samples). *P* values were calculated using a two-sided unpaired *t*-test. h, hour. **d**, Stability of DNAJC15 in *DNAJC15*[−/−] cells expressing DNAJC15 (JC15) or DNAJC15 cleaved by OMA1 (S-JC15)

and depleted of AFG3L2 (si*AFG3L2*) after inhibition of cytosolic translation with emetine. **e**, Degradation rates of DNAJC15 in **d**, normalized to time point 0 h. Data are means ± s.d. (*n* = 3, biologically independent samples). Statistics were calculated with a two-way ANOVA (genotype × treatment) test. **f**, Stability of DNAJC15 in *DNAJC15*[−/−] cells expressing DNAJC15 (JC15) or a non-cleavable DNAJC15 variant lacking amino acid 19 and 20 (JC15Δ2), depleted of AFG3L2 (si*AFG3L2*) after inhibition of cytosolic translation with emetine. **g**, Degradation rates of DNAJC15 in **f**, normalized to time point 0 h. Data are means ± s.d. (*n* = 3, biologically independent samples). *P* values were calculated using a two-way ANOVA test (genotype × treatment). **h**, Schematic of the OMA1-mediated DNAJC15 processing leading to a degradation by the m-AAA protease.

stress and reveal an intricate network of cellular stress responses to proteostasis disturbances.

## Results

### OMA1 cleavage facilitates DNAJC15 degradation by the m-AAA protease

To identify potential OMA1 substrate proteins, we searched for amino-terminal peptides of mitochondrial proteins that accumulate in an OMA1-dependent manner in a proteome-wide survey in wild-type (WT) and *OMA1*[−/−] human cervical cancer cells (Fig. 1a and Supplementary Table 1). Among the four Neo-N peptides of MitoCarta 3.0 proteins[26], levels of which were significantly decreased in *OMA1*[−/−] cells, was one amino-terminal peptide of the protein DNAJC15 (MCJ), a cochaperone of the human TIM23 translocase in the mitochondrial IM[27–29] (Extended Data Fig. 1a). We detected a peptide corresponding to amino acids 20–35, indicating that OMA1 cleaves DNAJC15 (Q9Y5T4) after amino acid 19.

DNAJC15 has been identified as a short-lived mitochondrial protein[30,31], and we have recently observed that DNAJC15 accumulates

in HeLa cells lacking the m-AAA protease subunit AFG3L2, raising the possibility that the m-AAA protease mediates proteolysis of DNAJC15 (refs. 30,31). Cycloheximide (CHX)-chase experiments confirmed the rapid turnover of DNAJC15, which was completely halted in cells lacking both m-AAA protease subunits, AFG3L2 and SPG7 (Fig. 1b,c, Extended Data Fig. 1b and Supplementary Table 2). DNAJC15 accumulated in a shorter form in these cells (Fig. 1b). Analysis of the Neo-N-terminal proteome of m-AAA protease-deficient cells revealed the accumulation of the DNAJC15 amino-terminal peptide spanning residues 20–35 (Fig. 1a and Supplementary Table 3), suggesting that OMA1 mediates cleavage of full-length DNAJC15 (hereafter termed L-DNAJC15) after amino acid 19 into a shorter form (hereafter termed S-DNAJC15) that is degraded by the m-AAA protease. In accordance, DNAJC15 cleavage is prevented by OMA1 depletion in *AFG3L2*[−/−]*SPG7*[−/−] (Fig. 1b,c). However, antimycin A and oligomycin treatment activated OMA1 in cells, inducing cleavage of L-DNAJC15 and resulting in the formation of S-DNAJC15 (Extended Data Fig. 1c,d). Removal of the stress-inducing agents restores wild-type DNAJC15 levels.

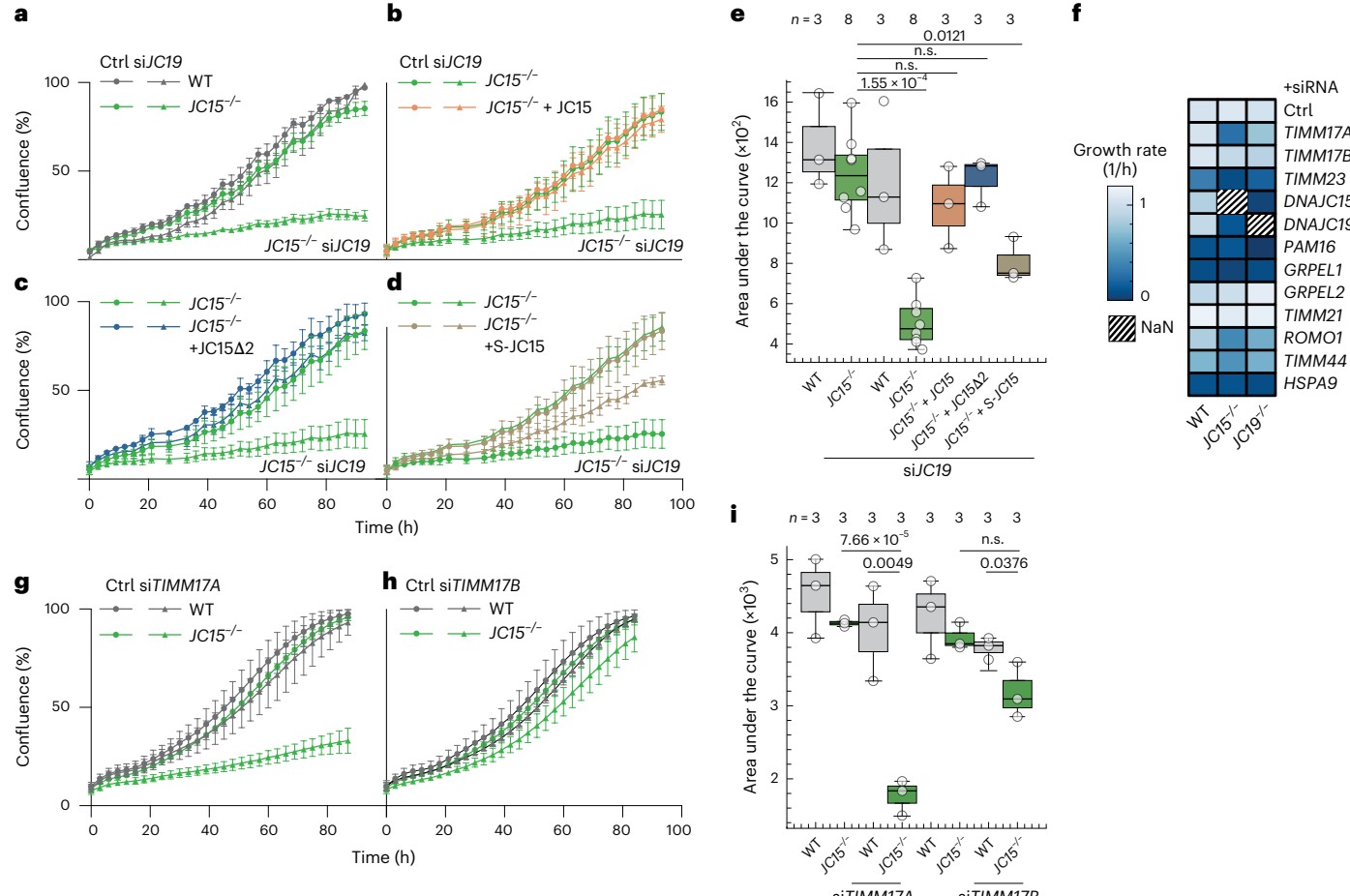

**Fig. 2 | Genetic interactions of DNAJC15 with subunits of the TIM23 complex.**
**a**, Cell growth of WT and *DNAJC15*[−/−] HeLa cells after siRNA-mediated depletion of DNAJC19 (WT, *n* = 3; *DNAJC15*[−/−] *n* = 8, biologically independent samples). Data are means ± s.d. **b**, Cell growth of *DNAJC15*[−/−] HeLa cells expressing DNAJC15 (JC15) after siRNA-mediated depletion of DNAJC19 (*n* = 3, biologically independent samples). Data are means ± s.d. **c**, Cell growth of *DNAJC15*[−/−] HeLa cells expressing non-cleavable DNAJC15 (JC15Δ2) after siRNA-mediated depletion of DNAJC19 (*n* = 3, biologically independent samples). Data are means ± s.d. **d**, Cell growth of *DNAJC15*[−/−] HeLa cells expressing OMA1-cleaved DNAJC15 (S-JC15) after siRNA-mediated depletion of DNAJC19 (*n* = 3, biologically independent samples). Data are means ± s.d. **e**, Quantification of cell growth analysis, determining the area under the curve in **a**–**d**. *P* values were calculated using a two-sided Mann–Whitney *U*-test. Quantile box plot show median (center line) and the 25th and 75th percentiles, and whiskers show minimum

and maximum values (1.5 × IQR distance from median, outliers not shown) (WT, *n* = 3; *DNAJC15*[−/−], *n* = 8, *DNAJC15*[−/−] + JC15Δ2, *n* = 3, *DNAJC15*[−/−] + S-JC15, *n* = 3, biologically independent samples). **f**, Heatmap of growth rates of WT, *DNAJC15*[−/−] and *DNAJC19*[−/−] HeLa cells after siRNA depletion of the indicated subunits of the TIM23 complex. Data are means (*n* = 3, biologically independent samples). **g,h**, Cell growth of WT and *DNAJC15*[−/−] HeLa cells after siRNA-mediated depletion of TIMM17A (**g**) or TIMM17B (**h**) for 72 h (*n* = 3, biologically independent samples). Data are mean values ± s.d. **i**, Quantification of cell growth monitoring the area under the curve in (**g,h**). *P* values were calculated using a two-sided unpaired *t*-test. Quantile box plot show the median (center line) and 25th and 75th percentiles, and whiskers show the minimum and maximum values (1.5 × IQR distance from median, outliers not shown, *n* = 3, biologically independent samples).

---

To confirm these findings, we generated *DNAJC15*[−/−] HeLa cell lines expressing DNAJC15, S-DNAJC15 (lacking amino acids 2–19) or a DNAJC15 variant lacking amino acids 19 and 20 (hereafter referred to as DNAJC15Δ2) (Extended Data Fig. 1e). Experiments in *DNAJC15*[−/−] cells expressing DNAJC15 variants lacking 2, 6 or 10 amino acids at the OMA1 cleavage site have shown that deletion of amino acids 19 and 20 completely abrogated OMA1 cleavage, even after membrane hyperpolarization, which activates OMA1 (Extended Data Fig. 1f,g). S-DNAJC15 was targeted to mitochondria and stabilized upon depletion of the m-AAA protease in cells expressing DNAJC15 or S-DNAJC15 (Fig. 1d,e). Remarkably, impaired OMA1 cleavage led to a significant stabilization of DNAJC15Δ2, which accumulated in WT and m-AAA-protease-deficient cells (Fig. 1f,g).

We conclude that cleavage of DNAJC15 after amino acid 19 by OMA1 leads to the destabilization of DNAJC15 and rapid proteolysis by the m-AAA protease (Fig. 1h).

## Genetic interactions link DNAJC15 to mitochondrial protein import

Complementation studies in yeast have shown functional conservation between DNAJC15 and its yeast ortholog Pam18 (Tim14)[29,32], suggesting that proteolysis of DNAJC15 by OMA1 and the m-AAA protease regulate mitochondrial protein import. Although Pam18 (Tim14) is essential for cell growth and mitochondrial protein import in yeast[33–35], the basal characterization of *DNAJC15*[−/−] cells did not reveal global changes in cell growth or oxygen consumption, nor did it reveal broad changes in the cellular proteome (Extended Data Fig. 2a–c). Human cells express two Pam18 paralogs, DNAJC15 and DNAJC19. Both proteins are highly homologous in the J-domain and the transmembrane region, but DNAJC15 exposes an extended N-terminal region to the mitochondrial intermembrane space (IMS), which is cleaved by OMA1 (Extended Data Fig. 2d). We therefore hypothesized that DNAJC19 could compensate for the loss of DNAJC15. Indeed, depletion of DNAJC19

abolished cell proliferation of *DNAJC15*[−/−] cells but did not affect the growth of WT HeLa cells (Fig. 2a,e). Re-expression of DNAJC15 or DNAJC15Δ2 restored cell growth (Fig. 2b,c,e), whereas only partial recovery was observed after expression of S-DNAJC15 (Fig. 2d,e). Thus, DNAJC15 and DNAJC19 show functional redundancy, suggesting that both play a part in mitochondrial protein import.

We extended the analysis of the genetic interaction landscape of DNAJC15 to other components of the human TIM23 translocase[28]. Consistent with their central role in protein translocation through TIM23 complexes, downregulation of PAM16, HSPA9 or GRPEL1 strongly inhibited cell growth in WT cells independent of the presence of DNAJC15 or DNAJC19 (Fig. 2f). Similarly, TIMM23, TIMM44 or ROMO1 depletion limited the proliferation of WT cells (Fig. 2f). Cell growth was further reduced in the absence of DNAJC15, providing additional genetic support for a role for DNAJC15 in protein import.

Notably, depletion of TIMM17A strongly impaired the growth of *DNAJC15*[−/−] cells, while only moderately affecting WT and *DNAJC19*[−/−] cell growth (Fig. 2f,g,i). Re-expression of DNAJC15, DNAJC15Δ2 or S-DNAJC15, restored the growth of *DNAJC15*[−/−] cells depleted of TIMM17A (Extended Data Fig. 2e–h). By contrast, DNAJC15 did not genetically interact with the TIMM17A-homolog TIMM17B (Fig. 2h,i). The genetic interaction of DNAJC15 with TIMM17A, but not TIMM17B, was further confirmed by acute knockdown experiments (Extended Data Fig. 3a–g). These studies link the function of DNAJC15 to mitochondrial protein import.

## The interactome of DNAJC15

To define the molecular environment of DNAJC15 in mitochondria, we used immunoprecipitation coupled with chemical cross-linking to isolate untagged DNAJC15 and associated proteins. Liquid chromatography coupled to tandem mass spectrometry (LC−MS/MS) analysis revealed a significant enrichment of 185 proteins, 179 of which were mitochondrial (according to MitoCop[31]). The vast majority of these are localized to the mitochondrial matrix and IM (Fig. 3a,b and Supplementary Table 4). Consistent with previous reports[29], PAM16 emerged as the strongest interactor of DNAJC15 (Fig. 3c). Moreover, we identified TIMM23, TIMM44, HSPA9, TIMM50, TIMM21 and DNAJC19 as significant DNAJC15 interactors, further supporting the association of DNAJC15 with the TIM23 translocase (Fig. 3c).

An interaction network based on MitoCop pathways revealed close proximity of DNAJC15 to mitochondrial quality control and biogenesis factors (Fig. 3d). These include the m-AAA protease subunit AFG3L2 and its interactors MAIP1 and TMBIM5 (GHITM), the i-AAA protease YME1L and the CLPB chaperone and the scaffold proteins PHB1, PHB2 and STOML2 (SLP2), which are associated with AAA proteases and affect OXPHOS biogenesis (Fig. 3a,d)[36–40]. Strikingly, the vast majority of proteins in the vicinity of DNAJC15 function in mitochondrial gene expression and respiratory chain biogenesis (Fig. 3d), highlighting the emerging coupling between protein translocation and OXPHOS biogenesis[41,42]. Strikingly, the most enriched interactors of DNAJC15 include enzymes involved in coenzyme Q metabolism, components of the respiratory complex I assembly and subunits of complex I, in particular those in the Q module and the ND4 module (Fig. 3d). Another prominent interactor of DNAJC15 is the microprotein NCBP2−AS2, which has been found to interact with COQ8 in a high-throughput screen and, more recently, with OXPHOS and TIM translocase subunits[43,44].

## DNAJC15 loss impairs mitochondrial protein import

Given that the genetic and protein-interaction landscapes linked the function of DNAJC15 to mitochondrial protein import, we monitored mitochondrial protein distribution using stable-isotope labelling with amino acids in cell culture (SILAC)-based proteomics combined with cellular fractionation. We labeled WT HeLa cells with different stable isotopes of lysine and arginine before transfection with DNAJC15-specific,

DNAJC19-specific or scrambled short interfering RNA (siRNA) (Fig. 4a). This allowed us to pool the three samples before cell fractionation, to reduce experimental variation. We then isolated the mitochondrial fraction by differential centrifugation and removed potential non-imported mitochondrial preproteins with exogenously added protease. The total cell fraction and mitochondrial fractions with and without protease treatment were subjected to LC−MS/MS analysis in four independent biological replicates (Fig. 4a). In total, 8,342 proteins were quantified, covering 952 (83.8%) of the MitoCarta 3.0 annotated proteins. The SILAC ratios of individual proteins between depleted and non-depleted cells reflected changes in their steady-state levels under the different experimental conditions (Supplementary Table 5).

Depletion of DNAJC15 or DNAJC19 only moderately affected the distribution of mitochondrial proteins in the total cell fraction (Fig. 4b,c). We observed just a moderate increase in 110 and 77 mitochondrial proteins in cells depleted of DNAJC15 or DNAJC19, respectively (Fig. 4b,c and Extended Data Fig. 8d). Thus, mitochondrial proteins are synthesized at similar rates regardless of the presence or absence of DNAJC15 or DNAJC19. However, 188 MitoCarta 3.0 annotated proteins were reduced in the untreated mitochondrial fraction after DNAJC15 depletion, and there was a significant systematic decrease of mitochondrial mass in DNAJC15-depleted cells compared with WT cells (Fig. 4b,c). After the mitochondrial fraction was treated with protease, the steady-state level of 75 more proteins was significantly reduced, indicating that these proteins were not imported but remained associated with the outer mitochondrial membrane (OM) in DNAJC15-depleted mitochondria (Fig. 4b,c). By contrast, the depletion of DNAJC19 did not significantly reduce steady-state levels of mitochondrial proteins (Fig. 4b,c). Notably, mitochondrial membrane potential was not altered, and the turnover rate of short-lived mitochondrial proteins was unaffected by acute DNAJC15 depletion (Extended Data Fig. 4a–d). To substantiate these findings, we performed protein-import experiments using isolated mitochondria. We observed a pronounced impairment in the import of mitochondrial proteins, including COQ10B and TIMMDC1, upon DNAJC15 depletion (Fig. 4d,e).

These data and our proteome-wide analysis suggest that DNAJC15 depletion impairs mitochondrial import and broadly rewires the mitochondrial proteome.

## DNAJC15 and TIMM17A ensure the import of OXPHOS-related proteins

The loss of DNAJC15 mainly impaired the import of proteins localized to the matrix and IM, but also affected a few IMS- and OM-localized proteins (Extended Data Fig. 4e). The steady-state levels of proteins imported through the TIM22 complex and of the subunits of this translocase remained unchanged upon DNAJC15 depletion (Extended Data Fig. 4f–h). Gene enrichment analysis of significantly changed proteins revealed broad effects on mitochondrial proteins with diverse functions in mitochondrial gene expression (Extended Data Fig. 5a). Among the downregulated mitochondrial proteins, those associated with mitochondrial ribosome assembly and RNA granules were most affected, as shown by a MitoPathway analysis (Fig. 4f). These results reveal that DNAJC15 supports the mitochondrial import of many matrix and IM proteins with OXPHOS-related functions. In accordance, 71 proteins, which were present at reduced levels after DNAJC15 depletion, are significantly enriched in the interactome of DNAJC15 and might bind to it during membrane translocation into the mitochondria (Extended Data Fig. 5b).

Considering the genetic interaction of DNAJC15 with TIMM17A, but not TIMM17B, we also depleted TIMM17A or TIMM17B from WT and *DNAJC15*[−/−] HeLa cells and used LC−MS/MS to analyze the mitochondrial proteome (Fig. 4g and Supplementary Table 6). Depletion of one TIMM17 paralog did not affect the accumulation of the other (Extended Data Fig. 5c,d). Levels of 204 and 50 mitochondrial proteins were significantly reduced after depletion of TIMM17A and

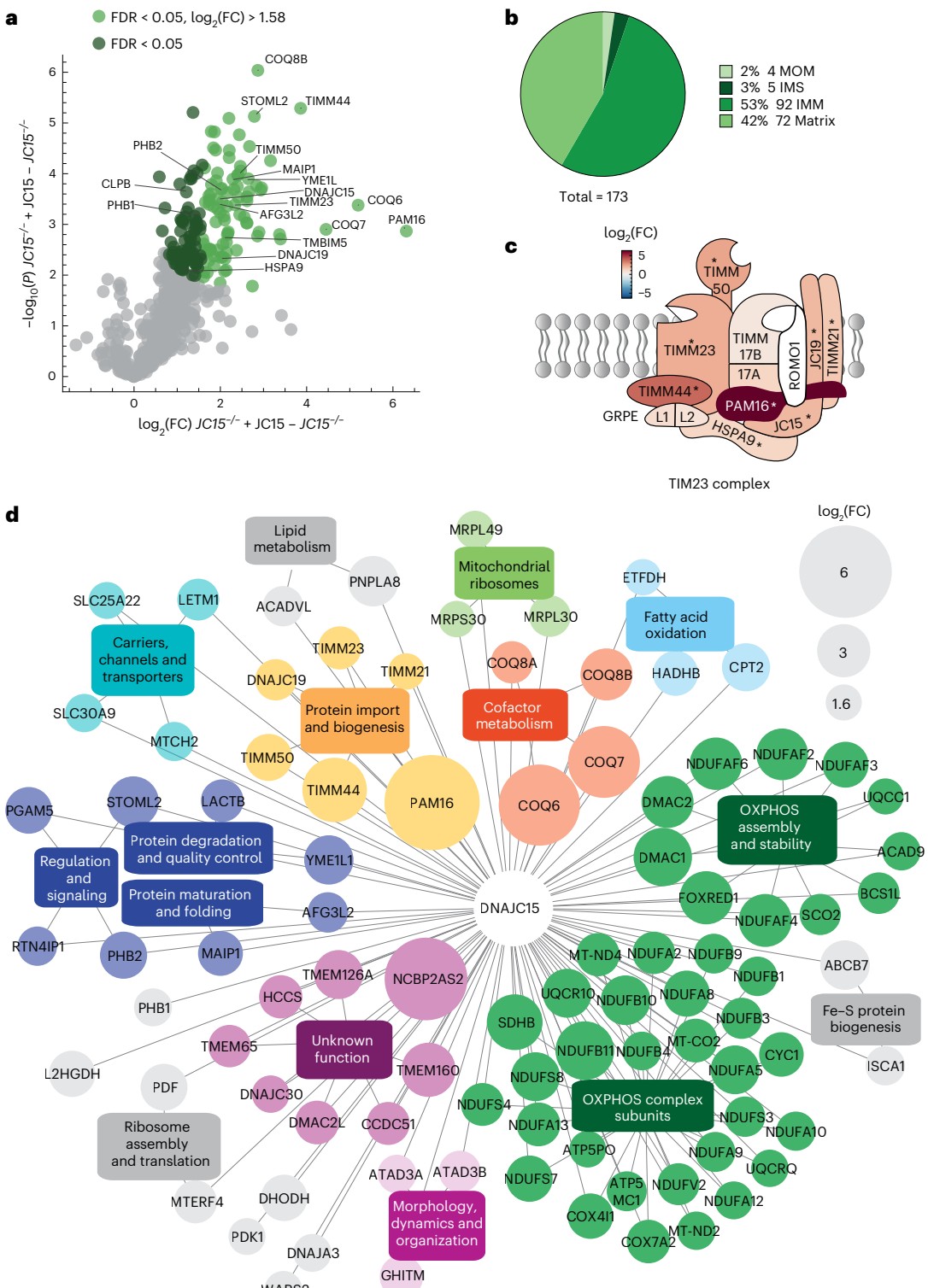

**Fig. 3 | The DNAJC15 interactome.** DNAJC15 immunoprecipitation in *DNAJC15*−/− HeLa cells expressing DNAJC15 and in *DNAJC15*−/− HeLa cells after chemical crosslinking (DSP, 2 mM) under non-denaturing conditions. **a**, Volcano plot showing the mitochondrial proteins that interact with DNAJC15, as defined by MitoCarta 3.0. A total of 174 mitochondrial proteins (according to MitoCarta 3.0) exhibited a significant interaction, and these are highlighted in green (permutation-based FDR < 0.05, *n* = 4, biologically independent samples); those with at least a 1.5-fold enrichment are shown in dark green (FDR < 0.05, log₂(FC) < 1.58). *P* values (*y* axis) were calculated using an unpaired two-sided *t*-test. **b**, Pie chart visualizing the known submitochondrial localization of the 173 significant mitochondrial DNAJC15 interaction partners (according to MitoCarta

3.0) (FDR < 0.05). MOM, mitochondrial outer membrane; IMS, intermembrane space; IMM, inner mitochondrial membrane. **c**, Schematic of the TIM23 complex subunits interacting with DNAJC15. The blue–white–red color scale indicates fold-enrichment between *DNAJC15*−/− cells expressing DNAJC15 and *DNAJC15*−/− HeLa cells (log₂(FC)). The asterisk indicates all significantly affected proteins (FDR < 0.05, *n* = 4, biologically independent samples). **d**, Network analysis of mitochondrial DNAJC15 interactors with a fold-enrichment higher than 1.58 (FDR < 0.05). The box size is adapted to their fold-enrichment (log₂(FC)) in *DNAJC15*−/− HeLa cells expressing DNAJC15 compared to *DNAJC15*−/− HeLa cells. Pathways associated with only one gene name were removed.

TIMM17B, respectively (Extended Data Fig. 5e–g). Notably, depletion of TIMM17A in *DNAJC15*[−/−] cells decreased the steady-state levels of an additional 135 mitochondrial proteins, leading to a significant shift in mitochondrial mass (Extended Data Fig. 6a). TIMM17B depletion in *DNAJC15*[−/−] cells reduced the steady-state levels of further 73 proteins (Extended Data Fig. 5e–g). A two-way ANOVA analysis revealed 42 mitochondrial proteins showing a significant genotype-by-knockdown interaction effect on abundance. Performing hierarchical clustering revealed a significant downregulation of many mitochondrial ribosome and OXPHOS-related proteins only in cells lacking both TIMM17A and DNAJC15, not those lacking TIMM17B and DNAJC15 (Fig. 4g and Extended Data Fig. 6b). These experiments suggest that both DNAJC15- and TIMM17A-containing complexes support the mitochondrial import of OXPHOS-related proteins into the matrix and IM.

We did not observe differences in various parameters, such as protein abundance, protein length, mitochondrial targeting sequence (MTS) length or MTS score, between affected and non-affected mitochondrial proteins (Extended Data Fig. 6c–f). However, the half-life of mitochondrial proteins that were significantly decreased after DNAJC15 depletion was lower than the median stability of the mitochondrial proteome (Fig. 4h and Extended Data Fig. 6g)[31]. Similarly, short-lived proteins were enriched among those present at reduced levels upon TIMM17A depletion in *DNAJC15*[−/−] cells (Fig. 4i). It is therefore conceivable that, along with differences in import specificity, the high turnover of specific mitochondrial proteins also contributes to the observed proteomic changes, when the general protein-import capacity of mitochondria is reduced upon degradation of DNAJC15.

## DNAJC15 preserves OXPHOS activity and coenzyme Q synthesis

Given the profound effect of DNAJC15 on the steady-state levels of numerous proteins involved in mitochondrial gene expression and OXPHOS biogenesis, we determined cellular oxygen consumption rates (OCRs) in isolated DNAJC15-depleted mitochondria. Consistent with previous findings[27], the OCR was reduced upon DNAJC15 depletion, which was restored upon re-expression of DNAJC15 (Fig. 5a,b and Extended Data Fig. 6h,k). Expression of DNAJC15Δ2 only partially rescued this deficiency, indicating that DNAJC15 processing affects respiration (Fig. 5c and Extended Data Fig. 6k). Loss of DNAJC15 significantly impaired complex I activity for both phosphorylation-coupled and phosphorylation-uncoupled states (Fig. 5a and Extended Data Fig. 6h). Similarly, we observed moderately reduced complex II activity in DNAJC15-depleted mitochondria (Fig. 5d and Extended Data Fig. 6i). It should be noted that the activities of respiratory complexes were not affected in the absence of DNAJC15 at the cellular level (Fig. 5e and Extended Data Figs. 2b and 6j), probably owing to the moderately increased mitochondrial mass in the cells indicated by our proteomic analysis (Fig. 4b,c). Thus, DNAJC15 preserves OXPHOS function, ensuring the mitochondrial protein import of many matrix and IM proteins whose function is linked to mitochondrial gene expression and OXPHOS biogenesis.

The respiratory deficiency after depleting DNAJC15 is consistent with the binding of DNAJC15 to OXPHOS-related proteins (Fig. 3d). However, this interaction analysis does not distinguish between an association of DNAJC15 with polypeptides during import or after completion of translocation. Indeed, the yeast DNAJC15 homolog Pam18 has dual functions in protein translocation and assembly of respiratory-chain complexes[41]. Because the N-terminal region of DNAJC15 that is cleaved by OMA1 is required for the chaperone to form associations with TIM23 complexes[27], we investigated how OMA1 cleavage affects the interactome of DNAJC15. However, the rapid turnover of S-DNAJC15 during chemical crosslinking precluded the direct identification of S-DNAJC15-interacting proteins (Extended Data Fig. 7a,b). We therefore compared the DNAJC15 interactome in *DNAJC15*[−/−] cells expressing DNAJC15 to non-cleavable DNAJC15Δ2. We identified 96 mitochondrial proteins as interacting partners of DNAJC15Δ2, all of which were also found in association with DNAJC15 (Extended Data Fig. 7c). Interacting proteins that did not depend on OMA1 cleavage included the majority of TIM subunits, such as PAM16, TIMM44, TIMM23 and TIMM50, supporting an import function of L-DNAJC15 (Extended Data Fig. 7d). A further 83 proteins were detected exclusively in association with cleavable DNAJC15 (Extended Data Fig. 7c and Supplementary Table 4). A direct comparison of the interactomes of DNAJC15 and DNAJC15Δ2 revealed that binding of coenzyme Q (CoQ) biosynthetic enzymes was most strongly affected and significantly reduced by impairment of OMA1 cleavage (Fig. 5f), suggesting that these proteins bind preferentially to S-DNAJC15.

To corroborate these findings, we determined the DNAJC15 interactome in *AFG3L2*[−/−] *SPG7*[−/−] cells, which lack the m-AAA protease and in which S-DNAJC15 accumulates (Fig. 1b). Strikingly, we identified COQ4,

**Fig. 4 | Mitochondrial import defects after depletion of DNAJC15. a**, WT HeLa cells were labeled with different stable isotopes through SILAC for a minimum of five splits, to adapt cellular pools of stable-isotope-labeled lysine and arginine. Differently labeled cells were depleted of DNAJC19 (si*JC19*) and DNAJC15 (si*JC15*) for 48 h, and an equal number of cells corresponding to the three conditions were pooled before subcellular fractionation, to minimize experimental variation. Samples were analyzed by LC–MS/MS (*n* = 5, biologically independent samples). **b**, Boxplot visualizing the distribution of mitochondrial proteins (MitoCarta 3.0) in the whole cell and in the mitochondrial fractions, which were treated with Proteinase K (PK) where indicated. A two-sided Mann–Whitney *U*-test was performed. Quantile box plot showing the median, 25th and 75th percentiles; whiskers show minimum and maximum values after outlier removal (defined as distance from median 0.5 × IQR, outliers not shown) (*n* = 5, biologically independent samples). **c**, Significantly altered mitochondrial proteins (MitoCarta 3.0) in the different fractions (FDR < 0.05) after depletion of DNAJC19 (si*JC19*, top) and DNAJC15 (si*JC15*; bottom), separated according to their SILAC ratio compared with WT (SILAC ratio > 0 for upregulation and SILAC ratio < 0 for downregulation). **d,e**, The indicated radiolabeled proteins, COQ10B (**d**) and TIMMDC1 (**e**), were incubated with the indicated mitochondria for the indicated times at 30 °C. Import was blocked with a 5 min incubation of 8 μM antimycin A, 1 μM valinomycin and 20 μM oligomycin. After treatment with Proteinase K (20 μg ml⁻¹) for 20 min on ice, mitochondria were subjected to SDS–PAGE and radioimaging. Imported proteins were quantified, and the amounts of the radiolabeled proteins were normalized to the control at time point 1 h. Values are means ± s.d. (*n* = 3, biologically independent samples). The area under the curve for each replicate and *P* values were calculated using a two-sided unpaired *t*-test. **f**, Mitochondrial pathways enrichment analysis (MitoCarta 3.0) of significantly changed mitochondrial proteins after DNAJC15 depletion relative to WT (Fisher exact test, FDR < 0.02). **g**, Heat map of the two-way ANOVA significant protein groups (−log₁₀(*P*) > 2.9). Proteins corresponding to the significantly enriched Gene Ontology term 'mitochondrial ribosome' are highlighted. The dataset was filtered for minimum required values of more than 3 (*n* = 5, biologically independent samples). LFQ, label-free quantification. **h**, Boxplot visualizing the half-life distribution of the mostly affected proteins (log₂FC < −0.58) in siRNA-depleted DNAJC15 HeLa cells compared with MitoCoP-annotated proteins. The quantile box plot shows the median (center line) and 25th and 75th percentiles, and whiskers show the minimum and maximum values after outlier removal (0.5 × IQR distance from median, outliers not shown). A two-sided Mann–Whitney *U*-test was performed (*n* = 5, biologically independent samples, including outliers). The half-life was extracted from Morgenstern et al.[31]. **i**, Boxplot visualizing the distribution of the half-life of significantly downregulated and mitochondrial proteins (log₂(FC) < −0.58, FDR < 0.05) in *DNAJC15*[−/−] HeLa cells after siRNA-mediated depletion of TIMM17A compared with all MitoCop-annotated proteins. Quantile box plots show the median (center line) and 25th and 75th percentiles, and whiskers show minimum and maximum values after outlier removal (0.5 × IQR distance from median, outliers not shown). Mann–Whitney *U*-test was performed (*n* = 5, biologically independent samples, including outliers). The half-life was extracted from Morgenstern et al.[31].

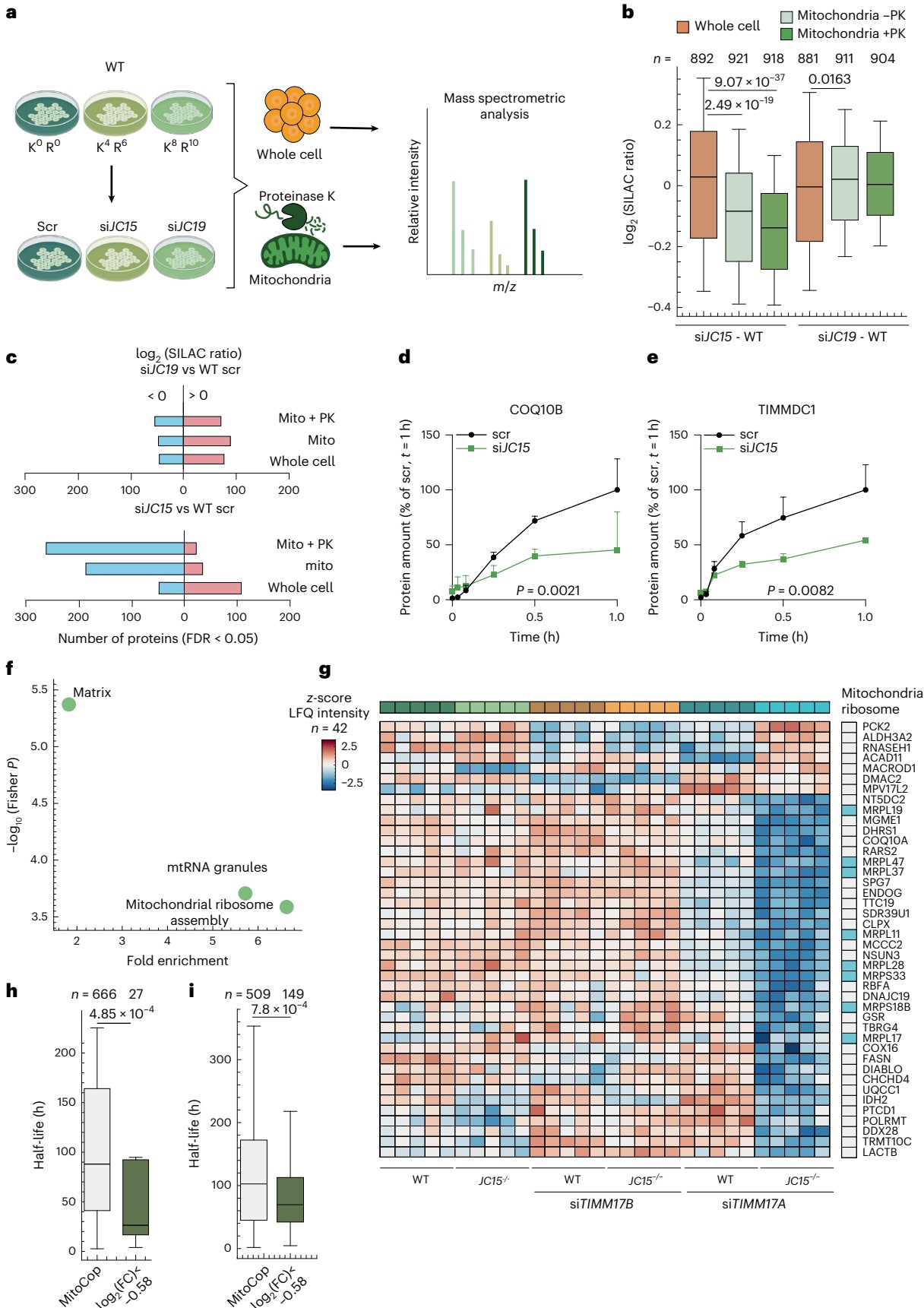

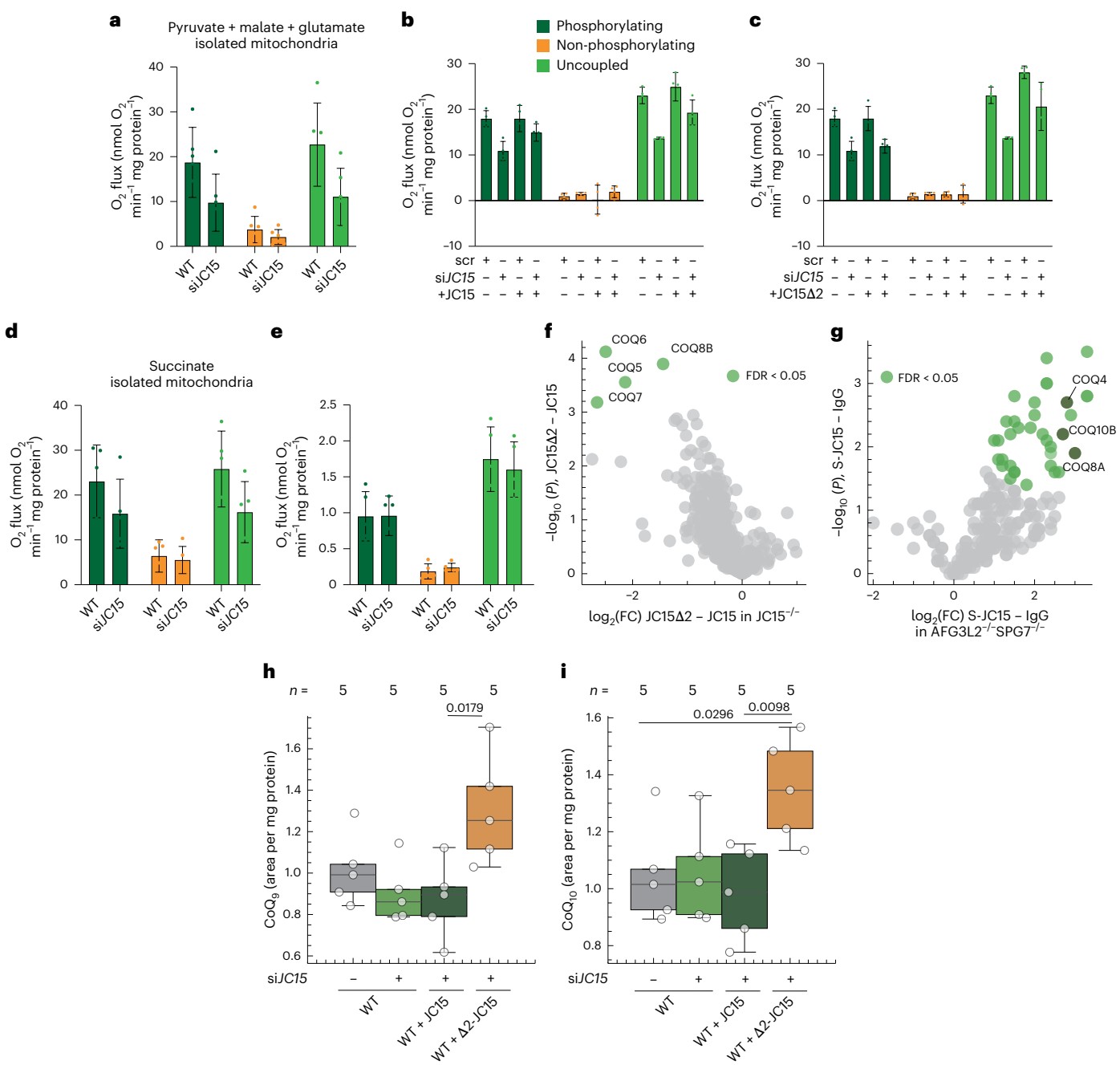

COQ8A and COQ10B exclusively in $AFG3L2^{-/-}SPG7^{-/-}$ cells, strongly suggesting that S-DNAJC15 binds CoQ biosynthetic enzymes (Fig. 5g). Of note, COQ4 and COQ10B are candidate substrates of AFG3L2 and accumulate in $AFG3L2^{-/-}SPG7^{-/-}$ cells, which likely explains why they were not identified in the crosslinking experiments in WT cells (Fig. 5f and Supplementary Table 7).

The interaction of S-DNAJC15 with CoQ biosynthetic enzymes might affect CoQ synthesis. We therefore determined the cellular levels of conenzyme Q9 ($CoQ_9$) and Q10 ($CoQ_{10}$) by metabolomics in WT cells, which were depleted of DNAJC15 and expressed siRNA-resistant variants of DNAJC15 and DNAJC15Δ2 (Fig. 5h,i and Extended Data Fig. 7e). Depletion of DNAJC15 did not affect the levels of $CoQ_9$ or $CoQ_{10}$. By contrast, $CoQ_9$ and $CoQ_{10}$ levels were significantly increased upon expression of DNAJC15Δ2, but not of cleavable DNAJC15 (Fig. 5h,i). These observations suggest that S-DNAJC15 might have an inhibitory role in CoQ metabolism and balance CoQ synthesis with OXPHOS biogenesis.

## Mitochondrial proteins mislocalize to the ER upon loss of DNAJC15 and trigger the unfolded stress response

Since the loss of DNAJC15 reduced the steady-state levels of many mitochondrial proteins in the organelle, but not at the cellular level, we performed further cell-fractionation experiments to determine the localization of the non-imported mitochondrial proteins (Fig. 6a). After SILAC labeling and downregulation of DNAJC15 and DNAJC19, we isolated mitochondria by low-speed centrifugation (8,000$g$; fraction A) and further fractionated the supernatant by differential centrifugation, to enrich the endoplasmic reticulum (ER) and plasma membrane (40,000$g$; fraction B) or other vesicular structures (100,000$g$; fraction C). The remaining supernatant was designated as the cytosolic fraction (fraction D) (Fig. 6a). LC–MS/MS analysis of the cell fractions confirmed the enrichment of different organellar structures in the different fractions (Extended Data Fig. 8a–c) and showed an accumulation of 130 mitochondrial proteins in the fraction B containing the ER membrane only after depletion of DNAJC15

**Fig. 5 | Impaired mitochondrial import after loss of DNAJC15 limits OXPHOS activity. a**, Oxygen flux was measured in intact mitochondria isolated from DNAJC15-depleted HeLa cells and WT cells, using an Oroboros respirometer. Oxygen flux was assessed in the presence of mitochondrial complex I substrates (pyruvate 10 mM, glutamate 5 mM, malate 5 mM), under phosphorylating conditions (ADP+$P_i$), under non-phosphorylating conditions (oligomycin, 10 μM) and after uncoupling with CCCP ($n = 5$, biologically independent samples). Values are means ± s.d. **b**, Oxygen flux was measured in intact mitochondria isolated from DNAJC15-depleted HeLa WT cells and those re-expressing DNAJC15 (JC15), using an Oroboros respirometer. Oxygen flux was assessed in the presence of mitochondrial complex I substrates (pyruvate 10 mM, glutamate 5 mM, malate 5 mM), under phosphorylating conditions (ADP+$P_i$), under non-phosphorylating conditions (oligomycin 10 μM) and after uncoupling with CCCP ($n = 4$, biologically independent samples). Data were normalized to the average of oxygen flux of WT HeLa cells under phosphorylating conditions. Values are means ± s.d. **c**, Oxygen flux was measured in intact mitochondria isolated from DNAJC15-depleted HeLa WT cells and those re-expressing uncleavable DNAJC15 (JC15Δ2), using an Oroboros respirometer. Oxygen flux was assessed in the presence of mitochondrial complex I substrates (pyruvate 10 mM, glutamate 5 mM, malate 5 mM), under phosphorylating conditions (ADP+$P_i$), non-phosphorylating conditions (oligomycin 10 μM) and after uncoupling with CCCP ($n = 4$, biologically independent samples). Data were normalized to the average of oxygen flux of wild-type HeLa cells under phosphorylating conditions. Values are means ± s.d. **d**, Oxygen flux was measured in intact mitochondria isolated from DNAJC15-depleted HeLa cells and WT cells, using an Oroboros respirometer. Oxygen flux was assessed in the presence of mitochondrial complex II substrate (succinate 10 mM) and rotenone (140 nM) under phosphorylating and non-phosphorylating conditions and after uncoupling (n = 4, biologically

independent samples). Values are means ± s.d. **e**, Oxygen flux was measured in DNAJC15-depleted HeLa cells and WT cells under phosphorylating and non-phosphorylating conditions and after uncoupling ($n = 4$, biologically independent samples). Values are means ± s.d. **f**, Volcano plot showing the interaction between mitochondrial proteins and DNAJC15 weakening after the OMA1-cleavage site was mutated (JC15Δ2) (according to MitoCarta 3.0), compared to DNAJC15$^{-/-}$ cells expressing DNAJC155 (JC15Δ2 – JC15 in JC15$^{-/-}$). Four significant mitochondrial interactors are highlighted in green (FDR < 0.05; $n = 4$, biologically independent samples). $P$ values ($y$ axis) were calculated using an unpaired two-sided $t$-test. **g**, Volcano plot showing mitochondrial S-DNAJC15 (S-JC15) interactors, identified by DNAJC15 bait pulldown, compared with the IgG control in *AFG3L2$^{-/-}$ SPG7$^{-/-}$* cells (S-JC15 – IgG in AFG3L2$^{-/-}$ SPG7$^{-/-}$). Proteins that were also significantly enriched in wild-type DNAJC15 pulldown samples relative to IgG were excluded (FDR < 0.05, $n = 4$ biologically independent samples). $P$ values ($y$ axis) were calculated using an unpaired two-sided $t$-test. Significantly changed CoQ proteins were highlighted in dark green. **h**, CoQ$_9$ abundance, normalized to the protein content, in cells expressing siRNA-resistant DNAJC15 (JC15) and DNAJC15Δ2 (JC15Δ2), after siRNA-mediated depletion of DNAJC15 (si*JC15*). $P$ values were calculated using a two-sided unpaired $t$-test. Quantile box plots show the median (center line) and 25th and 75th percentiles, and whiskers show minimum and maximum values (1.5 × IQR distance from median, outliers not shown, $n = 5$, biologically independent samples). **i**, CoQ$_{10}$ abundance, normalized to the protein content, in cells expressing siRNA-resistant DNAJC15 (JC15) and DNAJC15Δ2 (JC15Δ2), after siRNA-mediated depletion of DNAJC15 (si*JC15*). $P$ values were calculated using a two-sided unpaired $t$-test. Quantile box plots show the median (center line) and 25th and 75th percentiles, and whiskers show minimum and maximum values (1.5 × IQR distance from median, outliers not shown, $n = 5$, biologically independent samples).

(Fig. 6b,c and Supplementary Table 8). IM proteins and matrix proteins, which are imported along the TIM23$^{motor}$ pathway[45], were enriched among proteins accumulating in fraction B (Extended Data Fig. 8e–g). Fifty-three mitochondrial proteins that accumulated in fraction B were also significantly downregulated in fraction A. This is consistent with the observed protein-import deficiency in the absence of DNAJC15 (Extended Data Fig. 8d). By contrast, we did not observe reduced steady-state levels of mitochondrial proteins in fraction A or accumulation of mitochondrial proteins in fraction B after depletion of DNAJC19 (Fig. 6d). Because mitochondrial proteins detected in the ER fraction included many membrane proteins, we performed carbonate extraction to determine their mode of association with the ER. We observed that the mitochondrial preproteins accumulating at

the ER were predominantly detected in the pellet fraction, indicating membrane insertion (Fig. 6e and Supplementary Table 9). Together, these results suggest that non-imported mitochondrial preproteins accumulate at the ER specifically in cells lacking DNAJC15.

The accumulation of misfolded proteins and disturbances in ER proteostasis trigger ER stress responses[46,47]. Therefore, we performed RNA sequencing and observed that Gene Ontology terms associated with the unfolded protein response (UPR) were upregulated (Fig. 6f). To decipher which branch of the UPR is activated under DNAJC15-depleted conditions, we used a gene-set-profiling approach, which was developed to identify the activation of stress-responsive signaling pathways[48,49]. These pathways included the integrated stress response (ISR), heat shock response (HSR) and oxidative stress response (OSR),

**Fig. 6 | Non-imported mitochondrial proteins after DNAJC15 loss induce an unfolded protein response at the ER. a**, Subcellular fractionation of SILAC-labeled cells depleted of DNAJC15 (si*JC15*) and DNAJC19 (si*JC19*) (see Fig. 4a) by differential centrifugation, followed by LC–MS/MS analysis ($n = 5$, biologically independent samples). Figure created in BioRender; Kroczek, L.; https://biorender.com/i68gx8v (2025). **b**, Volcano plot of mitochondrial proteins accumulating in the 40,000$g$ fraction between cells depleted of DNAJC15 (si*JC15*) compared to wild-type HeLa cells (si*JC15* – WT) (130 significant proteins (FDR < 0.05, determined by Benjamini–Hochberg procedure) are labeled in green) ($n = 5$, biologically independent samples). $P$ values ($y$ axis) were calculated using an unpaired two-sided $t$-test. **c**, Boxplot visualizing the distribution of mitochondrial proteins (MitoCarta 3.0) in the different cellular fractions as a SILAC ratio (log$_2$) between DNAJC15-depleted and WT cells. Quantile box plots show the median (center line) and 25th and 75th percentiles, and whiskers show minimum and maximum values after outlier removal (0.5 × IQR distance from median, outliers not shown). A two-sided Mann–Whitney $U$-test was performed ($n = 5$, biologically independent samples, including outliers). $n$ indicates the number of proteins identified in each fraction. **d**, Boxplot visualizing the distribution of mitochondrial proteins (MitoCarta 3.0) in all different fractions as in **c** after DNAJC19 depletion. $n$ indicates the number of proteins identified in each fraction. Quantile box plots show the median (center line) and 25th and 75th percentiles, and whiskers show the minimum and maximum values after outlier removal (0.5 × IQR distance from median, outliers not shown) ($n = 5$, biologically independent samples, including outliers). **e**, Boxplots visualizing the distribution

of mitochondrial proteins (MitoCarta 3.0) in the pellet and supernatant fraction after carbonate extraction of the 40,000$g$ fraction. The SILAC ratio (log$_2$) compares siRNA-mediated depletion of DNAJC15 with wild-type (WT) control. $n$ indicates the number of proteins identified in each fraction. Quantile box plots show the median (center line) and 25th and 75th percentiles, and whiskers show the minimum and maximum values after outlier removal (0.5 × IQR distance from median, outliers not shown). Two-sided Mann–Whitney $U$-test was performed ($n = 5$, biologically independent samples, including outliers). **f**, Gene Ontology enrichment (Fisher exact test by Benjamini–Hochberg procedure, FDR < 0.02) analysis of all significant gene groups after DNAJC15 depletion (si*JC15*) relative to WT ($P < 0.05$). **g**, Gene expression analysis of WT HeLa cells and HeLa cells depleted of DNAJC15 (si*JC15*) for 48 h by RNA sequencing. Box plot visualizing different transcriptionally regulated stress responses according to gene sets regulated downstream of the following signaling pathways[49]: the UPR depending on IRE1 or ATF6; the ISR; the HSR and the OSR. Quantile box plots show the median (center line) and 25th and 75th percentiles, whiskers show minimum and maximum values after outlier removal (0.5 × IQR distance from median, outliers not shown). A two-sided Mann–Whitney $U$-test was performed ($n = 5$, biologically independent samples, including outliers). **h**, Heat-map of ATF6-related UPR targets[49] whose expression in DNAJC15-depleted relative to wild-type (WT) cells (log$_2$(FC)) is shown. Genes whose expression was significantly (unpaired $t$-test followed by Benjamini–Hochberg correction) changed after DNAJC15 depletion (adjusted $P < 0.05$) are shown in green.

and the different branches of UPR of the ER mediated by IRE1 or ATF6 (Fig. 6g and Supplementary Table 10).

Loss of DNAJC15 did not induce an ISR, as indicated by the unaltered expression of the known target genes of the ISR or the ISR$^{mt}$ (Fig. 6g and Extended Data Fig. 9a,b). OMA1, which can trigger the ISR$^{mt}$ through DELE1 processing, was not activated in these cells, nor

was OPA1 processing increased (Extended Data Fig. 9c). However, DNAJC15-depleted cells exhibited the expected ISR upon oligomycin treatment, demonstrating that these cells are not refractory to ISR induction (Extended Data Fig. 9b). Similar to the ISR, the loss of DNAJC15 did not induce either the OSR or the HSF1-dependent HSR, which was observed after HSP90 inhibition and perturbation of

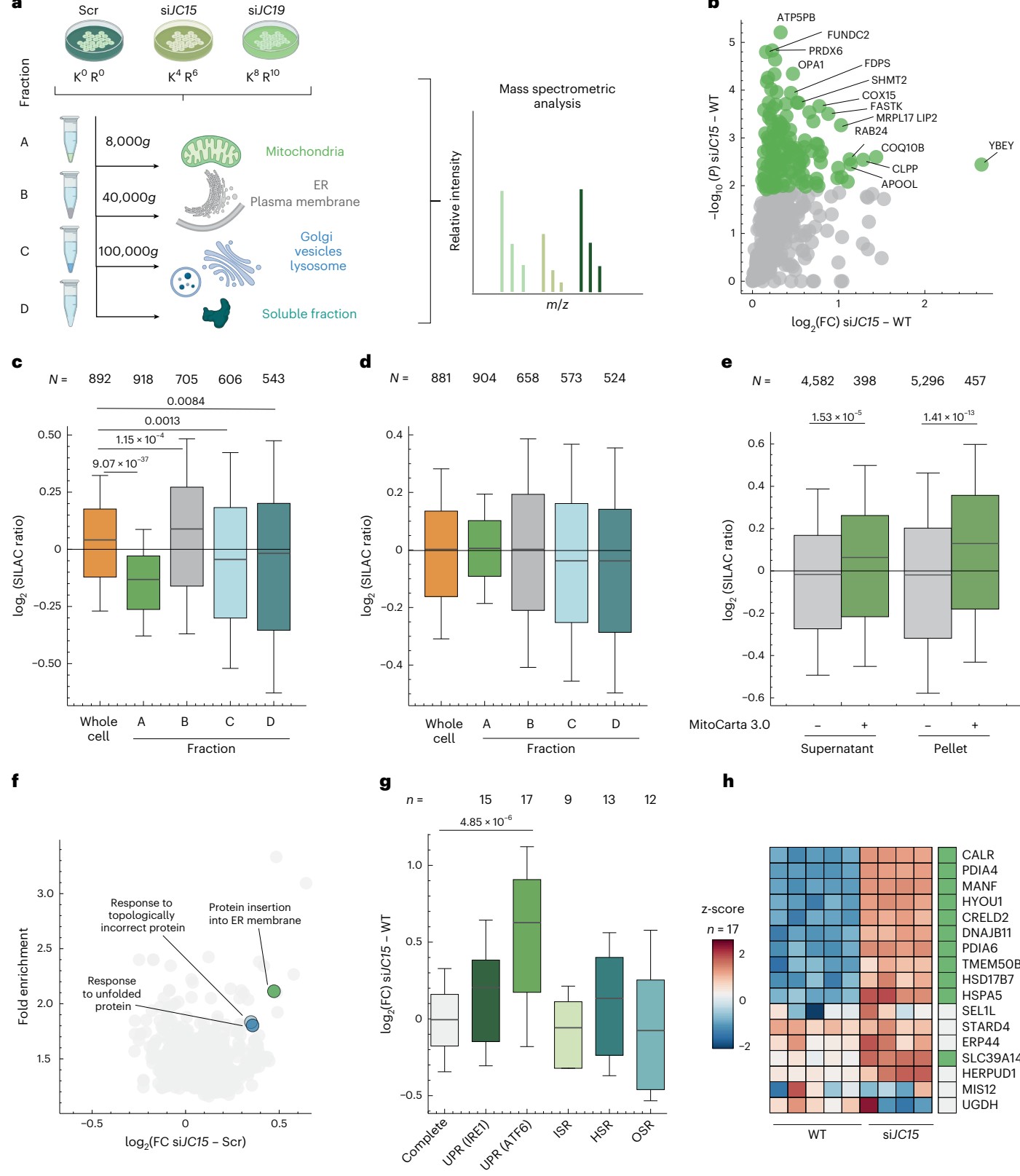

mitochondrial proteostasis leading to UPR[mt] (ref. 50). By contrast, we observed activation of the ATF6-mediated UPR, as indicated by the increased expression of 11 downstream genes (Fig. 6h), whereas the IRE1/XBP1-mediated UPR was not induced (Fig. 6g).

We conclude from these experiments that the loss of DNAJC15 impairs mitochondrial protein import and leads to the accumulation of mitochondrial preproteins at the ER, disrupting ER proteostasis and triggering the UPR.

## Discussion

We have identified DNAJC15 turnover as part of the OMA1-mediated mitochondrial stress response, which integrates changes in mitochondrial morphology and stress signalling with protein import regulation. In response to mitochondrial deficiencies, OMA1 cleavage at the N terminus of DNAJC15 initiates its degradation by the m-AAA protease, thereby limiting the import of OXPHOS-related proteins into mitochondria. We propose that this reduces OXPHOS biogenesis under stress, until mitochondrial function can be restored or damaged mitochondria are removed by mitophagy[51].

DNAJC15, together with its paralog DNAJC19, supports mitochondrial protein import through TIM23 complexes as part of the import motor at the matrix side of the IM[27,52]. Consistently, we identified PAM16 (MAGMAS), DNAJC19, HSPA9, TIMM23, TIMM44 and TIMM21 in the vicinity of DNAJC15. DNAJC15 binding to TIM23 complexes did not depend on OMA1 cleavage, identifying L-DNAJC15 as the import-competent form. PAM16 recruits DNAJC15 (and DNAJC19) to TIM23 complexes and ensures HSPA9-dependent protein import[27,32,53]. Loss of DNAJC15 broadly affects protein biogenesis and impairs the import of proteins with OXPHOS-related functions. Depletion of TIMM17A, but not of TIMM17B, aggravates this effect, suggesting that both DNAJC15 and TIMM17A ensure mitochondrial-import-dependent OXPHOS biogenesis.

How do DNAJC15 and TIMM17A specifically affect the biogenesis of OXPHOS-related proteins? Previous biochemical experiments have identified different protein translocases harboring TIMM17A and TIMM17B that cooperate with DNAJC15 and DNAJC19 (ref. 27). Therefore, it is possible that DNAJC15 and TIMM17A are part of the same translocase complex, whose import specificity differs from that of the TIMM17B-containing complex. The identification of numerous OXPHOS-related proteins as DNAJC15 interaction partners suggests import specificity. Another possible explanation, consistent with the observed genetic interaction between the two proteins, is that the loss of either DNAJC15 or TIMM17A affects different translocase complexes. Notably, both DNAJC15 and TIMM17A are unstable proteins. Whereas OMA1 cleavage of DNAJC15 facilitates its degradation by AFG3L2, TIMM17A is rapidly degraded by the i-AAA protease YME1L in a highly regulated process that is accelerated under various stress conditions[54–57]. The proteolytic breakdown of DNAJC15, TIMM17A or both reduces the general import capacity of mitochondria. Under conditions of limited import, intramitochondrial protein turnover and differences in the half-lives of mitochondrial proteins could reshape the mitochondrial proteome. Indeed, we noted that the half-lives of mitochondrial proteins that accumulated in mitochondria in a DNAJC15-dependent manner were shorter overall.

Although OMA1 promotes the degradation of DNAJC15 by AFG3L2, our interaction studies suggest that S-DNAJC15 has additional functions. The stabilization of DNAJC15 in $AFG3L2^{-/-} SPG7^{-/-}$ cells promoted its interaction with CoQ-biosynthetic enzymes. This interaction was lost in cells expressing only non-cleavable DNAJC15. Furthermore, we observed elevated CoQ levels in these cells, which indicates that S-DNAJC15 may inhibit CoQ synthesis. CoQ serves as an electron carrier in the respiratory chain and as cofactor of oxidoreductases and dehydrogenases. Therefore, regulating AFG3L2-mediated proteolysis of S-DNAJC15 could allow cellular CoQ levels to be adjusted in response to different metabolic demands.

Loss of DNAJC15 does not completely inhibit mitochondrial protein import nor block translocation by precursor stalling, but leads to accumulation of mitochondrial preproteins at the ER. Disruption of ER proteostasis in the absence of DNAJC15 triggers the UPR, demonstrating coupling of mitochondrial and ER proteostasis. The accumulation of mitochondrial preproteins at the ER is reminiscent of studies in yeast suggesting that some mitochondrial preproteins encounter the ER membrane on their way to the mitochondria[58,59]. Accordingly, the accumulation of mitochondrial preproteins at the ER upon loss of DNAJC15 during mitochondrial stress would allow a rapid recovery of mitochondrial import upon relief of the stress, which is accompanied by OMA1 degradation and restoration of DNAJC15 protein levels. Although it remains to be established whether ER-associated preproteins remain competent for mitochondrial import, our findings reveal a close link in organellar proteostasis regulation and an integrated cellular stress response to mitochondrial dysfunction.

## Online content

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

## Methods

### Cell culture

HeLa (CCL-2) cells were purchased from American Type Culture Collection (ATCC) and cultured in Dulbecco's Modified Eagle's Medium (DMEM), containing 4.5 g L$^{-1}$ D-glucose (Thermo Fisher, cat. no. 61965-059) and 10% vol/vol fetal bovine serum (FBS) (Sigma, cat. no. F7524). Cells were maintained at 37 °C and 5% CO$_2$ in a humidified incubator. Cell counting was performed using the viability marker trypan blue (Thermo Fisher, cat. no. T10282) with the Countess automated cell counter (Thermo Fisher, cat. no. AMQAF2001). Before each experiment, cells were seeded at equal densities and continuously monitored for mycoplasma contamination.

### Plasmid and siRNA transfection

For transient transfection of siRNA or endoribonuclease-prepared siRNA (esiRNA), a reverse transfection was conducted with a total cell number of 4×10$^5$ cells per well in a 6-well plate. The experiments were conducted 48 h or 72 h after transfection with RNAiMAXX transfection reagent (Invitrogen, cat. no. 56532). The following siRNA and esiRNA targeting sequences against human proteins were used: si*AFG3L2* (cat. no. SASI_HS01_0023-1520, Merck), esi*TIMM44* (cat. no. EHU009281, Merck), esi*HSPA9* (cat. no. EHU011841, Merck), si*DNAJC19* (cat. no. SASI_Hs01_00055864, Sigma), si*DNAJC15* (cat. no. SASI_Hs01_00246981, Sigma), esi*TIMM17B* (cat. no. EHU032651, Merck), esi*TIMM17A* (cat. no. EHU023551, Merck), si*TIMM21* (cat. no. EHU031871, Merck), si*PAM16* (cat. no. SASI_Hs01_00168312, Sigma), esi*TIMM23* (cat. no. EHU106141, Merck), esi*ROMO1* (cat. no. EHU224321, Merck), esi*TIMM50* (cat. no. EHU043281, Merck), esi*GRPEL1* (cat. no. EHU006111, Merck), esi*GRPEL2* (cat. no. EHU065941, Merck) and esi*OMA1* (cat. no. EHU072451, Merck). For negative control, esiRNA eGFP-negative control (Merck, cat. no. EHUEGFP) and stealth RNAi siRNA negative control (Invitrogen, cat. no.12935300) were obtained. The efficiency of protein depletion was evaluated through immunoblotting.

### Generation of stable cell lines

Complementary cDNAs encoding human DNAJC15, S-DNAJC15 and DNAJC15Δ2 were cloned into the PiggyBac vector (PB-Cuo-MCS-IRES-GFP-EF1α-CymR-Puro) using the Q5 site-directed mutagenesis kit (NEB, cat. no. E0554S). After plasmid transfection with Genejuice Transfection Reagent (Merck, cat. no. 70967) and 0.2 µg µl$^{-1}$ of the PiggyBac Transposase Expression Vector (PB210PA-1-SBI), cells were selected with puromycin (InvivoGen, cat. no. ant-pr-1) for 4 days after reaching full confluence. Experiments were performed with the puromycin-selected polyclonal fraction in a *DNAJC15*$^{-/-}$ HeLa cell background. Expression was induced with 8 or 15 µg ml$^{-1}$ of 4-Isopropylbenzoic acid (Aldrich, cat. no. 268402) and confirmed by immunoblotting. An siRNA-resistant DNAJC15 construct was generated by introducing silent mutations into the DNA sequence targeted by the siRNA, thereby altering the nucleotide composition without affecting the corresponding amino acid sequence. This strategy ensures that the re-expressed DNAJC15 remains functionally intact while evading siRNA-mediated knockdown.

### CRISPR-Cas9 gene editing

To generate *DNAJC15*$^{-/-}$, *DNAJC19*$^{-/-}$, *AFG3L2*$^{-/-}$*SPG7*$^{-/-}$ and *OMA1*$^{-/-}$ HeLa cells, cells were transfected with px335 plasmid (Addgene cat. no. 42335), which contained guide RNAs for a nickase Cas9 deletion. The gRNAs were ordered from Merck. The gRNAs for *DNAJC15* were 5′-ACTTGCAGCCCTCGGCCAAA-3′ and 5′-GTGGTGT CATCGCTCCAGTT-3′. The gRNAs for *DNAJC19* were 5′-CGGGAAGCAG CATTAATACT-3′ and 5′-TCAGTGGTGGCTATTATAGA-3′. The gRNAs for *OMA1* were 5′-CCATATAGTAAATAAGTATCAGG-3′ and 5′-TGGG AGTAAATCAGTGTGACAGG-3′. The gRNAs for *AFG3L2* and *SPG7* were 5′-GCTGCTACCACACGCTCTTC-3′ and 5′-GGTACATCAAGGCTAG CCGC-3′, respectively. Following three consecutive transfections on

subsequent days, cells were seeded as single cells in 96-well plates and examined for protein expression after a two-week growth period. Deletions were further confirmed by PCR and subsequent genomic sequencing analysis.

### IncuCyte-based cell growth assays

Cells were grown at equal densities in 96-well plates, with 5,000 cells per well. All experiments were repeated as independent biological replicates. Cell proliferation was monitored by phase microscopy imaging at ×10 magnification using the IncuCyte System (Sartorius). To assess cell growth, the slope of the log-transformed phase area was determined. Significance was assessed by an unpaired two-tailed *t*-test. Data were visualized using GraphPadPrism Version 10.4.1 (532).

### Mitochondrial membrane potential measurement

Cells were seeded in 6-well plates at equal densities of 400,000 cells per well. Treatment with 50 µM CCCP was performed 2 h prior to staining. Tetramethylrhodamine, methyl ester (TMRM) staining (Thermo Fisher, cat. no. T668) was carried out following the manufacturer's protocol, using 200 nM TMRM for 30 min in the dark. After staining, cells were resuspended in FACS buffer (DPBS supplemented with 2% FBS), passed through a 50-µm cell strainer, and analyzed using flow cytometry (FACScanto II, BD Biosciences) using the appropriate fluorescence channels. Single cells were identified using a conventional FSC/SSC gating strategy, and the mean fluorescence intensity of the gated population was quantified.

### Protein stability assays

Cells were grown at equal densities in 6-well plates at 400,000 cells per well. Plasmid and siRNA transfection was performed for 48 h, and the medium was changed to complete DMEM media containing either 0.1 mg ml$^{-1}$ Emetine (Sigma, cat. no. E2375) or 100 µg ml$^{-1}$ cyclohex-imide (Sigma, cat. no. C7698) for the indicated periods of time. Cells were collected and washed with PBS. Quantification of the protein decay was acquired using ImageJ (version 2.1.0/1.53c) and visualized with GraphPadPrism Version 10.4.1 (532).

### Antibodies

Antibodies against the following proteins were used for immunoblotting: DNAJC15 (ProteinTech, cat. no. 16063-1-AP, 1:500), Tubulin (Sigma, cat. no. T6074, 1:2,000), AFG3L2 (Sigma, cat. no. HPA004480, 1:1,000), Actin (Santa Cruz, cat. no. SC-47778, 1:3,000), SDHA (abcam, cat. no. ab14715, 1:5,000), OMA1 (Santa Cruz, cat. no. SC-515788; 1:1,000) and OPA1 (BD Biosciences, cat. no. 612607, 1:2,000). Corresponding species-specific HRP-coupled antibodies were used for immunoblot (Biorad; cat. no. 1706515 and cat. no. 1706516).

### SDS–PAGE and immunoblotting

Cells were collected and washed with ice-cold PBS, before lysing with lysis buffer (50 mM Tris-HCl pH 7.4, 1 mM EDTA, 0.1 (wt/vol) SDS, 1% (wt/vol) Triton-X-100, 0.5% (wt/vol) sodium deoxycholate, 150 mM (wt/vol) NaCl) containing complete EDTA-free protease inhibitor cocktail (Roche, cat. no. 64755100). Protein concentration was determined using a Bradford assay (BioRad, cat. no. 5000006). Samples were boiled for 5 min at 95 °C before SDS–PAGE. Total proteins from cells (100 µg) were separated using 14% SDS–PAGE and were then transferred to a nitrocellulose membrane, followed by immunoblotting with the indicated antibodies. Western blot images were acquired using an Intas ChemoStar ECL Imager HR6.0 (Intas, cat. no. 114801) and ChemoStar Ts software.

### Measurement of cellular respiration

OCRs were measured in a Seahorse Extracellular Flux Analyzer XFe96 (Agilent), according to the manufacturer's instructions for cells grown in DMEM-GlutaMAXX containing 25 mM glucose. For each well, 3.6×10$^3$

cells were plated and incubated for 24 h at 37 °C and 5% $CO_2$ in a humidified incubator before measurement. OCRs were assessed after the addition of the corresponding OXPHOS inhibitors (oligomycin (2 µM), FCCP (0.5 µM), rotenone and antimycin A (0.5 µM)), included in the Seahorse XF Cell Mito Stress Test Kit (Agilent, cat. no. 103015-100).

For respiration measurements using OROBOROS, the OCR was analyzed at 37 °C using isolated mitochondria (400 µg), which were suspended in 2.1 ml of respiratory buffer (MiR05 buffer: 110 mM sucrose, 0.5 mM EGTA, 20 mM Taurine, 3 mM $MgCl_2$, 60 mM K-lactobionate, 10 mM $KH_2PO_4$, 20 mM HEPES-KOH; pH 7.1) or for cells ($1 \times 10^6$ cells) in DMEM, containing 4.5 g $L^{-1}$ D-glucose (Thermo Fisher, cat. no. 61965-059) and 10% (vol/vol) FBS (Sigma, cat. no. F7524). Measurements were performed with an Oxygraph-2k system (Oroboros Instruments). Substrate-driven oxygen consumption was assessed in the presence of succinate (10 mM), pyruvate (10 mM), glutamate (5 mM) and malate (5 mM). The OCR was determined under three conditions: the phosphorylating state after the addition of ADP (1.2 mM), the non-phosphorylating state after treatment with oligomycin (10 µM for mitochondria, 20 µM for cells) and the uncoupled state induced by the addition of carbonyl cyanide m-chlorophenylhydrazone (CCCP), which was added in incremental steps up to a concentration of 50 µM (for mitochondria) and 500 µM (for cells), to achieve maximal respiratory capacity. To ensure accurate assessment of respiratory-chain-dependent oxygen consumption, antimycin A (500 nM) was added at the end of each experiment to account for residual oxygen consumption.

## Quantitative PCR

Cells were collected and washed with ice-cold PBS prior to cell lysis using the lysis buffer provided in the RNA isolation kit (Macherey-Nagel, cat. no. 740955.250). The experiment was performed according to the manufacturer's instructions. RNA isolation was quantified using a NanoDrop spectrophotometer (Thermo Fisher, cat. no. ND-ONE-W), and complementary DNA (cDNA) was synthesized using the GoScript Reverse Transcriptase kit (Promega, cat. no. A2791), according to the manufacturer's protocol, with a total RNA concentration of 50 µg ml$^{-1}$. Quantitative PCR was performed using SYBR Green PCR mix (Applied Biosystems, cat. no. 4367659), containing 0.5 µM of each primer and 20 ng of cDNA for 40 cycles at 95 °C for denaturation and 60 °C for the elongation step. Hypoxanthine-guanine phosphoribosyltransferase (HPRT) was used as a housekeeping gene to normalize expression levels, which were determined using the $2^{-\Delta\Delta Ct}$ method. The sequences are the following: For HPRT, we used 5′-TGACACTGGCAAAACAATGCA and 5′-GGTCCTTTTCACCAGCAAGCT-3′, for ATF4, we used 5′-AGGTCTCTT AGATGATTACC-3′ and 5′-CAAGTCGAACTCCTTCAAATC-3′, for ASNS, we used 5′-AAAGGTCCCAGATCATATTG-3′ and 5′-ATATGACTTCAT CCAGAGCC-3′, for MTHFD2, we used 5′-GTGGATTTTGAAGGAGTCAG-3′ and 5′-CTTTAGACTTCAGCACTTCTC-3′, for PYCR1, we used 5′-ACAAGAT AATGGCTAGCTCC-3′ and 5′-CAATGTGTCTGTCCTCAATG-3′, for SHMT2, we used 5′-CATTTGAGGACCGAATCAAC-3′ and 5′-CACCTGATACCAG TGAGTAG-3′, for DNAJC15, we used 5′-ATGAGTAGGCGAGAAGCTGGTC-3′ and 5′-GGGTGATTCAAAATCATGACTCTC-3′.

## In vitro protein import into isolated mitochondria

Radiolabeled proteins were synthesized in a cell-free transcription/translation system (Promega, cat. no. L2080 and L1170) in the presence of $^{35}$S-methionine (Hartmann Analytic, cat. no. SCM-01) following the manufacturer's instructions. To block import, mitochondria were preincubated for 5 min with 8 µM antimycin A, 1 µM valinomycin and 20 µM oligomycin. Import assays were conducted at 30 °C in import buffer (250 mM sucrose, 5 mM Mg-acetate, 80 mM K-acetate, 20 mM HEPES-KOH (pH 7.4), 10 mM Na-succinate, 3 mM ATP and 1 mM DTT). Reactions were terminated by placing the samples on ice. Non-imported precursor proteins were digested

with proteinase K (20 µg ml$^{-1}$) for 20 min on ice; protease activity was quenched with 2 mM PMSF.

## CoQ extraction and measurement

CoQ was extracted from cells using freshly prepared extraction buffer consisting of methyl tertiary-butyl ether (MTBE), methanol and water (50:30:20, vol/vol/vol), which was cooled to −20 °C before use. Cells were seeded at equal density with 400,000 cells per well in a 6-well plate. CoQ extraction was performed 2 days later. Following trypsinization, cells were pelleted by centrifugation at 800g for 3 min and washed with phosphate-buffered saline. Each sample was incubated with 1 ml extraction buffer at 1,500 r.p.m. in a thermomixer for 30 min at 4 °C. After centrifugation at 21,000g for 10 min, protein levels in the pellet were quantified by BCA assay. The resulting supernatant was mixed with 200 µl MTBE and 150 µl LC−MS grade water, then incubated at 15 °C for 10 min in a thermomixer. After a final centrifugation at 16,000g for 10 min at 15 °C to achieve phase separation, the upper lipid phase was transferred to a new tube and dried in a Speed Vac concentrator at 20 °C and 1,000 r.p.m.

Lipid pellets were reconstituted in 750 µL UPLC-grade acetonitrile:isopropanol (70:30, vol/vol), vortexed for 10 s and incubated for 10 min at 4 °C on an orbital mixer. After centrifugation at 10,000g for 5 min at 4 °C, supernatants were transferred to 2 ml glass vials equipped with 200 µl glass inserts. Samples were analyzed on a Vanquish Flex quaternary Ultra High Perfomance Liquid Chromatography (Thermo Fisher Scientific) coupled to an Orbitrap High Resolution Mass Spectrometry equipped with a heated ESI source (ID-X, Thermo Fisher Scientific). For the analysis, 2 µl of each sample were injected onto a 100 × 2.1 mm ACQUITY Premier CSH C18 Column (1.7 µm particles) and separated at a flow rate of 400 µl min$^{-1}$. The buffers were: A (10 mM ammonium formate, 0.1% formic acid in 60:40 (vol/vol) water/acetonitrile) and B (10 mM ammonium formate, 0.1% formic acid in isopropanol:acetonitrile 90:10 (vol/vol)). The UHPLC gradient was: 0−1 min 25% B, 1−2 min 25−40% B, 2−4 min 40−58% B, 4−6 min 58−60% B, 6−9 min 60−70% B, 9−11 min 70−73% B, 11−13 min 73−88% B, 13−15 min 88−100% B and 15−16 min 100% B, followed by a 4 min re-equilibration at 25% B. The total run time per sample was 20 min.

The mass spectrometer operated in positive ion mode, scanning $m/z$ 150−1,500 at a resolution of ×120,000, with the RF lens at 50%, AGC target at 40%, maximum ion time 100 ms, spray voltage 3.5 kV, capillary temperature 300 °C, sheath gas 60 AU, auxiliary gas 20 AU and sweep gas 1 AU at 340 °C. All samples were analyzed in randomized order. Targeted quantification was performed using the quan module in TraceFinder 5.1 (Thermo Fisher Scientific) with reference to a custom compound database based on commercial standards.

## Subcellular fractionation

Cell pellets were resuspended in homogenization buffer (20 mM HEPES-KOH pH 7.4, 220 mM mannitol, 70 mM sucrose, 1 mM EGTA pH 7.4) containing complete EDTA-free protease inhibitor cocktail (Roche, cat. no. 64755100). The suspension was kept on ice for 15 min. Cell disruption was performed by applying 20 strokes in a glass mortar and PTFE pestle at 1,000 r.p.m. using a rotating homogenizer (Schuett Biotec, cat. no. 3201011). Two consecutive centrifugation steps at 600g for 5 min were conducted to remove nuclei. Mitochondrial fractions were pelleted at 8,000g for 10 min at 4 °C. When indicated in the figure legend, mitochondria were treated with 20 µg ml$^{-1}$ Proteinase K (Southern Cross Science, cat. no. 0219350491) for 20 min at 4 °C, followed by reaction termination with 2 mM PMSF (Roche, cat. no. 10837091001). For further subcellular fractionation, the supernatant of the mitochondrial-containing fraction was centrifuged at 40,000g for 1 h, to yield the heavy microsome fraction. Additional centrifugation of the remaining supernatant at 100,000g for 1 h allowed isolation of the light microsomal fraction. The supernatant, containing the soluble fraction was precipitated with acetone.

## Sodium carbonate extraction of the ER fraction

Sodium carbonate extraction was performed to distinguish integral from peripheral ER-associated proteins. Sodium carbonate buffer (0.1 M, pH 11.0) was prepared freshly in water and adjusted to pH 11.0 with 0.1 M HCl. For the extraction, the supernatant obtained after mitochondrial isolation was centrifuged at 40,000$g$ for 1 h to isolate the ER-enriched fraction. The resulting pellet was resuspended in 0.1 M sodium carbonate (pH 11.0) and incubated for 20 min on ice. Samples were then ultracentrifuged at 126,000xg for 30 min. Both the supernatant, containing carbonate-extracted peripheral proteins, and the pellet, containing integral membrane-associated proteins, were supplemented with sodium deoxycholate and subjected to acetone precipitation.

## Neo-N-terminal proteomics

Cells were lysed and proteins were reduced (10 mM TCEP) and alkylated (20 mM CAA) in the dark for 45 min at 45 °C. TMT labels (TMT10plex Label Reagent Set plus TMT11-131C, cat. no. A37725, or TMT6plex Label Reagent Set) were equilibrated to RT and solubilized in LC−MS-grade acetonitrile according to manufacturer's protocol. For 30 µg protein input, 20 µl of TMT labels were added and incubated for 1 h at 25 °C on a thermomixer at 300 r.p.m. The reaction was stopped by adding 100 mM Tris-HCL buffer to a final concentration of 20 mM. Samples were pooled to a total of 100 µg and subjected to SP3-based protein digestion. Peptides were incubated with activated NHS-magnetic beads (Pierce NHS-Activated Magnetic Beads, cat. no. 88826), according to the manufacturer's instructions, to capture free amine groups of internal peptides (peptide N-term). The supernatant was removed, acidified and desalted using SDB-RP stage tips and resuspended in 10 mM ammonium hydroxide and 5% acetonitrile. Peptides were then separated through offline high-pH peptide fractionation. The instrumentation consisted of a ZirconiumTM Ultra HPLC and a PAL RTC autosampler system using the binary buffer system, consisting out of buffer A (10 mM ammonium hydroxide) and buffer B (80% acetonitrile and 10 mM ammonium hydroxide). Peptides were separated according to their hydrophobicity using an in-house packed column (length, 40 cm; inner diameter, 175 µm, 2.7-µm beads, PoroShell, Agilent Technologies) column. The instruments communicated and were controlled using the software Chronos (Axel Semrau). The total gradient length was 40 min, and a total of 12 ($SPG7^{-/-}$, $AFG3L2^{-/-}$, $SPG7^{-/-}AFG3L2^{-/-}$ and WT) or 36 ($OMA1^{-/-}$ and WT) fractions were collected. The fraction collector moved every 30 s to the next fraction well over the complete gradient. The collected fractions were concentrated using a SpeedVac to complete dryness.

## DSP interaction analysis

Cells were seeded at a density $15 × 10^6$ in a 15-cm cell-culture dish. If indicated, DNAJC15 expression was induced with 15 µg ml$^{-1}$ of 4-isopropylbenzoic acid (Aldrich, cat. no. 268402) and confirmed using immunoblotting. After 48 h of incubation in a humidified incubator, cells were washed twice with PBS. PBS containing 2 mM DSP (Thermo Fisher, cat. no. 22585) was then added and the cells were incubated for 30 min at RT to allow cross-linking. For quenching, a 100 mM Tris pH 7.4 solution was added and the cells were incubated for a further 15 min at RT. Cells were then collected and lysed (20 mM HEPES-NaOH, 150 mM NaCl, 2 g/g digitonin (Merck, cat. no. 300410)). The protein concentration was determined using Bradford reagent (BioRad, cat. no. 5000006). For subsequent immunoprecipitation, 1.4 mg/1 mg (DNAJC15 interactome/$AFG3L2^{-/-}SPG7^{-/-}$ interactome) of total protein was used as input, with DNAJC15 antibody (Proteintech, cat. no. 16063-1-AP) or the IgG control (Cell Signaling, cat. no. 2729) pre-bound to Protein G magnetic beads (Thermo Fisher, cat. no. 10003D). The Protein G and antibody mixture was washed twice for 10 min each with 0.1% sodium deoxycholate, and then washed three times with PBS, including a 10-min incubation step with PBS. After adding the lysate, the mixture was incubated for 120 min. After several washes

with washing buffer (10 mM HEPES-NaOH, pH 7.5, 150 mM NaCl, 0.1% (wt/vol) Triton-X-100), proteins were eluted by incubating the beads in Laemmli buffer (50 mM Tris-HCL, pH 6.8, 2% (wt/vol) SDS, 10% (vol/vol) glycerol, 0.01% (wt/vol) bromophenol blue, 60 mM DTT) at 70 °C for 10 min at 300 r.p.m.

## SILAC-based mitoproteomics

HeLa cells were cultured in DMEM medium without arginine, lysine and glutamine (Silantes, cat. no. 282006500), supplemented with glutamine and 10% (vol/vol) dialyzed FBS. Cells were adapted for seven doublings to heavy isotope medium containing [$^{13}C_6$,$^{15}N_4$] arginine (Silantes, cat. no. 201604102) and [$^{13}C_6$,$^{15}N_2$] lysine (Silantes, cat. no. 211604102), to middle isotope media [$^{13}C_6$] arginine (Silantes, cat. no. 201204102) and [$D_4$] lysine (Silantes, cat. no. 211104113), to light isotope media arginine (Silantes, cat. no. 201004102) and lysine (Silantes, cat. no. 211004102). The same medium was used for the corresponding siRNA transfection. After 48 h of incubation, cells were trypsinized, and cell counts were performed using the viability marker trypan blue (Thermo Fisher, cat. no. T10282) with the Countess automated cell counter (Thermo Fisher, cat. no. AMQAF2001). Equal numbers of cells were pooled, and subcellular fractionation was performed.

## Acetone precipitation

Four times the volume of ice-cold (−20 °C) acetone was added to the supernatant and incubated overnight for 16 h. The samples were centrifuged at 20,000$g$ for 10 min at 4 °C. The pellet was washed twice with 400 µl of 80% (vol/vol) ice-cold acetone. The pellet was dried under the fume hood for 5 min and then resuspended in 100 µl 4% (wt/vol) SDS in 100 mM HEPES-KOH (pH 8.5).

## SP3 digestion protocol

For total proteome analysis, 60 µl of 4% (wt/vol) SDS in 100 mM HEPES-NaOH (pH, 8.5) was pre-warmed to 70 °C and added to the cell pellet for a further 10 min incubation at 70 °C on a thermomixer (shaking, 550 r.p.m.). Protein concentration was determined using the 660 nm Protein Assay (Thermo Fisher, cat. no. 22660). Twenty micrograms of protein were subjected to tryptic digestion. For immunoprecipitations, the LDS buffer eluate was directly used. Proteins were reduced (10 mM TCEP) and alkylated (20 mM CAA) for 45 min at 45 °C in the dark. Samples were subjected to an SP3-based digestion[60]. Washed SP3 beads (SP3 beads (Sera-Mag Magnetic Carboxylate Måodified Particles (Hydrophobic, GE44152105050250) and Sera-Mag Magnetic Carboxylate Modified Particles (Hydrophilic, GE24152105050250) from Sigma Aldrich) were mixed equally, and 3 µl of bead slurry was added to each sample. Acetonitrile was added to a final concentration of 50% and washed twice using 70% ethanol (volume ($V$) = 200 µl) on a custom-made magnet. After a further acetonitrile wash ($V$ = 200 µL), 5 µl of digestion solution (10 mM HEPES-NaOH, pH 8.5, containing 0.5 µg Trypsin (Sigma, cat. no. T6567-1mg) and 0.5 µg LysC (Wako, cat. no. 129-02541) was added to each sample and incubated at 37 °C overnight. Peptides were desalted on a magnet using 2 ×200 µL of acetonitrile. Peptides were eluted in 10 µl 5% (vol/vol) DMSO in LC-MS water (Sigma Aldrich, cat. no. 900682) in an ultrasonic bath for 10 min and subjected to StageTip desalting using the SDB-RPS material (Affinisep, AttractSPE Disks SDB-RP, cat. no. SPE-Disks-Bio-DVB-47.20)[61]. Formic acid and acetonitrile were added to a final concentration of 2.5% (vol/vol) and 2% (vol/vol), respectively. Samples were stored at −20 °C prior to LC−MS/MS analysis.

## Liquid chromatography and mass spectrometry for Neo N-terminal proteome

LC−MS/MS instrumentation consisted of an Easy-LC 1200 (Thermo Fisher Scientific) coupled through a nano-electrospray ionization source to an QExactive HF-x mass spectrometer (Thermo Fisher Scientific). For peptide separation, an in-house packed column (inner diameter: 75 µm,

length: 20 cm) was used. A binary buffer system (A: 0.1% (vol/vol) formic acid and B: 0.1% (vol/vol) formic acid in 80% (vol/vol) acetonitrile) was applied as follows: linear increase of buffer B from 4% (vol/vol) to 28% (vol/vol) within 33 min, followed by a linear increase to 55% (vol/vol) within 5 min. The buffer B content was further ramped to 95% (vol/vol) within 2 min. Then, 95% (vol/vol) buffer B was kept for 3 min to wash the column. Before each sample was added, the column was washed using 6 μl buffer A, and the sample was loaded using 7 μl buffer A. The mass spectrometer operated in a data-dependent mode and acquired MS1 spectra at a resolution of 60,000 (at 200 $m/z$) using a maximum injection time of 20 ms and an AGC target of $3 \times 10^6$. The scan range was defined from 350–1,650 $m/z$, and data type was set to profile. MS2 spectra were acquired in a Top 15 mode at a ×45,000 resolution (at 200 $m/z$) using an isolation window of 0.8 $m/z$ and a normalized collision energy of 32. The first mass was set to 110 $m/z$. Dynamic exclusion was enabled and set to 20 s.

### Liquid chromatography and mass spectrometry for data-independent acquisition (whole proteome, immunoprecipitation, subcellular fractionation)
The LC–MS/MS instrumentation consisted of an Easy-LC 1200 (Thermo Fisher Scientific) coupled through a nano-electrospray ionization source to an Exploris 480 mass spectrometer (Thermo Fisher Scientific). An Aurora Frontier column (60 cm length, 1.7 μm particle diameter, 75 μm inner diameter, Ionopticks). A gradient-based buffer system (A: 0.1% (vol/vol) formic acid; B: 0.1% (vol/vol) formic acid in 80% (vol/vol) acetonitrile) was used at a flow rate of 185 nl min⁻¹ as follows: a linear increase of buffer B from 4% (vol/vol) to 28% (vol/vol) within 100 min, followed by a linear increase to 40% within 10 min. The buffer B content was further increased to 50% (vol/vol) within 4 min and then to 65% (vol/vol) within 3 min. Then, 95% (vol/vol) buffer B was maintained for a further 3 min to wash the column. The RF lens amplitude was set to 45% (vol/vol), the capillary temperature was set to 275 °C and the polarity was set to positive. MS1 profile spectra were acquired at a resolution of 30,000 (at 200 $m/z$) over a mass range of 450-850 $m/z$ and an AGC target of $1 \times 10^6$.

For MS/MS-independent spectra acquisition, 34 equally spaced windows were acquired with an isolation $m/z$ range of 7 Th, and the isolation windows overlapped by 1 Th. The first fixed mass was 200 $m/z$. The isolation center range covered a mass range of 500–740 $m/z$. Fragmentation spectra were acquired with a 30,000 resolution at 200 $m/z$ using a maximum injection time setting of 'auto' and stepped normalized collision energies (NCEs) of 24, 28 and 30. The default charge state was set to 3, and the AGC target was $3 \times 10^6$ (900%, Exploris 480). MS2 spectra were acquired in centroid mode. FAIMS was activated with an inner electrode temperature of 100 °C and an outer electrode temperature of 90 °C. The compensation voltage was set at −45 V.

### RNA-sequencing data
For eukaryotic mRNA sequencing, samples were processed by Novogene. Sequencing was performed on the Illumina NovaSeq X Plus platform (PE150) using a paired-end 150 bp (PE150) read strategy. Messenger RNA was purified from total RNA using poly-T oligo-attached magnetic beads. After fragmentation, the first strand of the cDNA was synthesized using random hexamer primers followed by the second-strand cDNA synthesis. The library was ready after end repair, A-tailing, adapter ligation, size selection, amplification and purification. The library was checked with Qubit and real-time PCR for quantification and Bioanalyzer for size distribution detection. Quantified libraries were pooled and sequenced on Illumina platforms, according to effective library concentration and data amount.

### Data Analysis
**Statistics and reproducibility.** All the independent experiments or biological samples are represented in the graphs. Instant Clue software 12.2 and GraphPad Prism V.10.4.1 were used to analyze all the datasets[62].

Data are presented as the 95% confidence interval of the mean, to show statistically significant differences between the groups. To compare two groups, $P < 0.05$ was considered significant, unless otherwise indicated. No statistical method was used to pre-determine sample size. The investigators for proteomics and metabolomics measurement were blinded to allocation during experiments and samples were randomized. The investigators were not blinded for all other experiments, and samples were not randomized. The center line of the visualized boxplots represents the median; edges of the box represent 25th and 75th quantiles; whiskers show minimum and maximum values excluding outliers. Outliers are defined by a greater distance from the median and 1.5× the inter-quartile range (IQR), if not indicated otherwise. The investigators were not blinded to allocation during experiments and outcome assessment. Replicate 5 of the si*DNAJC15*-treated samples of the RNA sequencing were excluded, as they did not pass the quality check owing to fewer reads. Data distribution was assumed to be normal, but this was not formally tested.

**RNA sequencing.** rRNA transcripts were removed from the annotation file by depleting all lines with 'rrna' tag on it. cDNA index was built using kallisto (kallisto/0.46.1) and RSeQC/4.0.0 used to identify mapping strands: a strand was identified by having more than 60% of reads mapped to it. Cases with less than 60% of reads in each strand are defined as unstranded. After normalization of read counts by making use of the standard median-ratio for estimation of size factors, pair-wise differential gene expression was performed using DESeq2/1.24.0. After removal of genes with fewer than ten overall reads, $\log_2$(fold changes) were shrunk using approximate posterior estimation for GLM coefficients. ATF4 targets were extracted from Torrence et al.[63].

**Neo N-terminal proteome.** Raw files were analysed using MaxQuant (v.1.6.7 and v.1.6.12)[64], and the implemented Andromeda search engine. TMT 10-plex (WT and *OMA1*⁻/⁻) or TMT-6plex (*SPG7*⁻/⁻, *AFG3L2*⁻/⁻, *SPG7*⁻/⁻*AFG3L2*⁻/⁻ and WT) was set as a quantification setting. ArgC with semi-specificity (free N terminus) was used. MS2 spectra were correlated against the Uniprot human reference proteome (UP000005640, number sequences: 21,000, downloaded: August 2022). Match-between runs algorithm was enabled. To identify Neo-N termini, the peptides.txt file of the MaxQuant output folder was utilized and TMT intensities between conditions were compared using a two-sided *t*-test. Raw files were analyzed using MaxQuant (v.1.6.4)[64] and the implemented Andromeda search engine. TMT-10plex was set as a quantification setting. ArgC with semi-specificity (free N-terminus) was used. MS2 spectra were correlated against the Uniprot human reference proteome. Match-between runs algorithm was enabled. To identify Neo-N termini, the peptides.txt file of the MaxQuant output folder was utilized and TMT intensities between conditions were compared using a two-sided *t*-test in the Perseus software.

**Whole proteome and crosslinking interaction studies.** The Spectronaut (18.7.240325.55695) directDIA+ (Deep) analysis tool was used to correlate the acquired MS2 spectra in data-independent mode with the Uniprot reference human proteome (UP000005640, number sequences: 21,000, downloaded: August 2022). The MaxLFQ algorithm was used. The precursor, peptide and protein $q$-value cutoff were 0.01 in the Pulsar search. A total number of two missed cleavages of two were allowed, and the minimum peptide length was seven amino acids. Acetyl (Protein N-term), and Oxidation (M) were defined as variable modifications, and Carbamidomethyl at cysteines was set as a fixed modification. The mean precursor and mean peptide quantities were used for agglomeration to protein quantities. The $\log_2$ protein group LFQ intensities were used for a pairwise comparison using a two-tailed unpaired *t*-test followed by a permutation-based FDR calculation (FDR < 0.05 is considered significantly different, s0 = 0.1, number of permutations = 500). The experiment was performed with 5 (whole proteome) and 4 (interaction studies) biologically independent replicates.

**SILAC-based subcellular fractionation.** The Spectronaut (18.7.240325.55695) directDIA tool was used to analyse the acquired raw files. channel 2 (Arg6, Lys4) and channel 3 (Arg10, Lys8) were defined. The option 'exclude interference' was enabled to exclude fragments with the same mass of different SILAC channels (for example co isolated b-ions). Otherwise, the default settings were utilized. The SILAC protein group ratio was then calculated by aggregating the precursor data to the mean of the $\log_2$ SILAC ratio at the elution group level and further calculating the median of the peptides $\log_2$ SILAC ratio to obtain the protein group H/L, H/M and M/L ratios. The ratio distributions were then shifted to a median of 0. Significantly different protein groups were identified using a one-sample t-test on the $\log_2$ protein group SILAC ratio. The P values were adjusted using the Benjamini-Hochberg correction (adj. P value < 0.05) in the Instant Clue software. MTS score and length of MTS were extracted from MTSviewer[65].

**Quantification and statistical analysis.** Densitometry data were generated by Fiji for Western blot quantification. Representative images from at least three independent experiments are shown. Graphs were generated using Prism (GraphPad, v.10.4.1). Volcano plots and heat maps were generated using InstantClue (v.0.12.2)[62]. Error bars represent the s.d. of the mean. Statistical significance of the data was assessed by using the two-sided unpaired t-test or the Mann−Whitney U-test, comparing control and test conditions, as described in the figure legend. Data were visualized in Instant Clue (v.0.12.2).

### Reporting summary

Further information on research design is available in the Nature Portfolio Reporting Summary linked to this article.

### Data availability

RNA sequencing data was uploaded to Gene Expression Omnibus (GEO) and is available under the accession number GSE299431. The proteomics data were uploaded to the following PRIDE projects under the following IDs: The cycloheximide chase in WT and $AFG3L2^{-/-}$ cells proteomics data are available under the identifier PXD056318. The Neo-N termiome of $OMA1^{-/-}$ HeLa cells are available under the identifier PXD061134. The Neo-N termiome of $AFG3L2^{-/-}$, $SPG7^{-/-}$, $AFG3L2^{-/-} SPG7^{-/-}$ and WT HeLa cells is available on PRIDE under the identifier PXD061137. The SILAC-based subcellular fraction proteomics experiment is available under the identifiers PXD061449 and PXD061185. Data from TIMM17A and TIMM17B knock-down in WT and $DNAJC15^{-/-}$ HeLa cells were deposited to PRIDE under the identifier PXD061131. The DNAJC15 DSP crosslinking interactome is available under the PRIDE identifier PXD061165. The DNAJC15 DSP crosslinking in $AFG3L2^{-/-} SPG7^{-/-}$ cells is available under the PRIDE identifier PXD070836. The SILAC-based subcellular fraction proteomics data, including carbonate extraction, are available under the identifier PXD070854. The expression proteomics data for $DNAJC15^{-/-}$ HeLa cells are deposited to PRIDE under the identifier PXD072822. All other data supporting the findings of this study are available from the corresponding author on reasonable request. Source data are provided with this paper.

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

### Acknowledgements

We would like to thank D. Diehl for excellent MS sample preparation and continuous support. We also appreciate the technical and bioinformatic assistance provided by the MPI Age Bioinformatics Facility, as well as the support from the MPI Age FACS Imaging Facility for cell growth-based analyses and flow cytometry analysis. We are grateful to the MPI Age Metabolomics Facility for their technical support and valuable discussions. This work was supported by the Deutsche Forschungsgemeinschaft (DFG. German Research Foundation) as part of the CRC1218 (grant number 269925409, project A01 and A05).

### Author contributions

T.L. designed and initiated the study. L.K., Y.L., I.A., D.C.P. and K.L. performed biochemical experiments and cellular studies. L.K. and H.N. performed MS experiments. L.K. and H.N. analysed proteomics and RNA-sequencing experiments. T.M. and E.R. assisted in bioenergetic measurements and interpretation. L.K., H.N. and T.L. wrote the manuscript with input from all authors.

### Funding

### Competing interests

The authors declare no competing interests.

### Additional information

**Extended data** is available for this paper at https://doi.org/10.1038/s41594-026-01756-0.

**Correspondence and requests for materials** should be addressed to Thomas Langer.

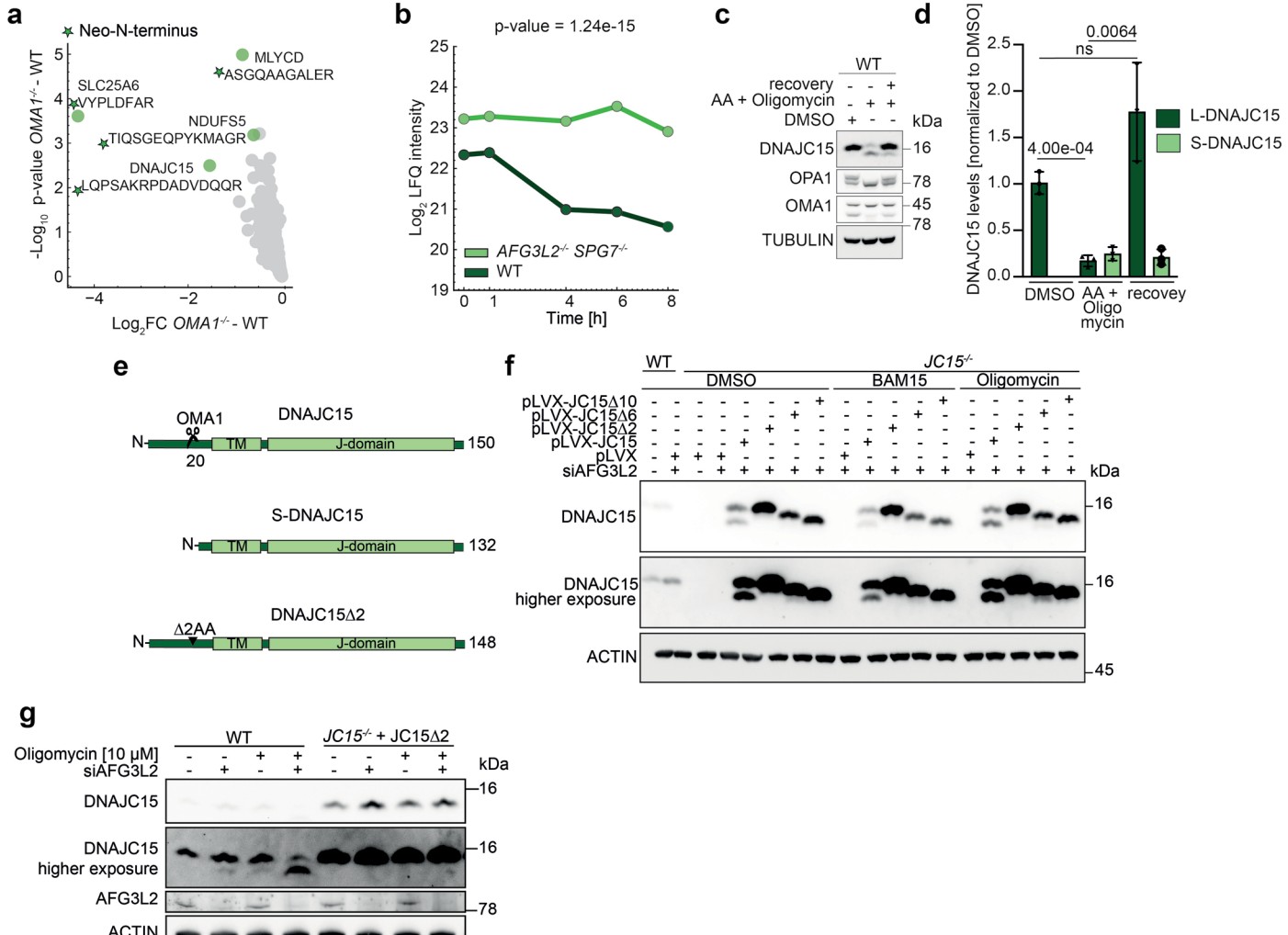

**Extended Data Fig. 1 | OMA1 cleaves DNAJC15 at AA position 19. a**, Volcano plot of the Neo-N-terminal proteome of wild-type (WT) and $OMA1^{-/-}$ HeLa cells, highlighting significantly affected mitochondrial proteins in green (permutation-based FDR < 0.05; MitoCarta 3.0). (n = 5, biologically independent samples). *P* values (y-axis) were calculated using an unpaired two-sided *t*-test. **b**, Stability of DNAJC15 in WT and $AFG3L2^{-/-} SPG7^{-/-}$ HeLa cells (FDR < 0.05; n = 5, biologically independent samples). *P* value for the interaction time x genotype was calculated using a random mixed linear model (not corrected for multiple testing). **c**, Western Blot analysis of WT HeLa cells treated with Antimycin A (AA, 100 nM) and Oligomycin (100 nM) for 16 h. Media was changed after treatment for additional 24 h containing no stress-inducing agents. **d**, Relative

DNAJC15 levels of (c) normalized to DMSO control. Data are means ± s.d. (n = 3, biologically independent samples). *P* values were calculated using a two-sided unpaired *t*-test. **e**, Schematic representation of the different DNAJC15 variants (TM = transmembrane domain). **f**, Western blot analysis of WT and $DNAJC15^{-/-}$ HeLa cells, which transiently express DNAJC15 or DNAJC15 variants lacking two (19/20), six (17-22) or ten (15-24) amino acids. After siRNA-mediated depletion of AFG3L2 for 72 h and treatment with oligomycin (10 μM) or Bam15 (10 μM) for 2 h, cells were analyzed by SDS-PAGE and immunoblotting (n = 1). **g**, Western blot analysis of WT and $DNAJC15^{-/-}$ HeLa cells stably expressing DNAJC15Δ2 following treatment with siRNA directed against AFG3L2 for 72 h and with oligomycin (10 μM) for 16 h as indicated (n = 1).

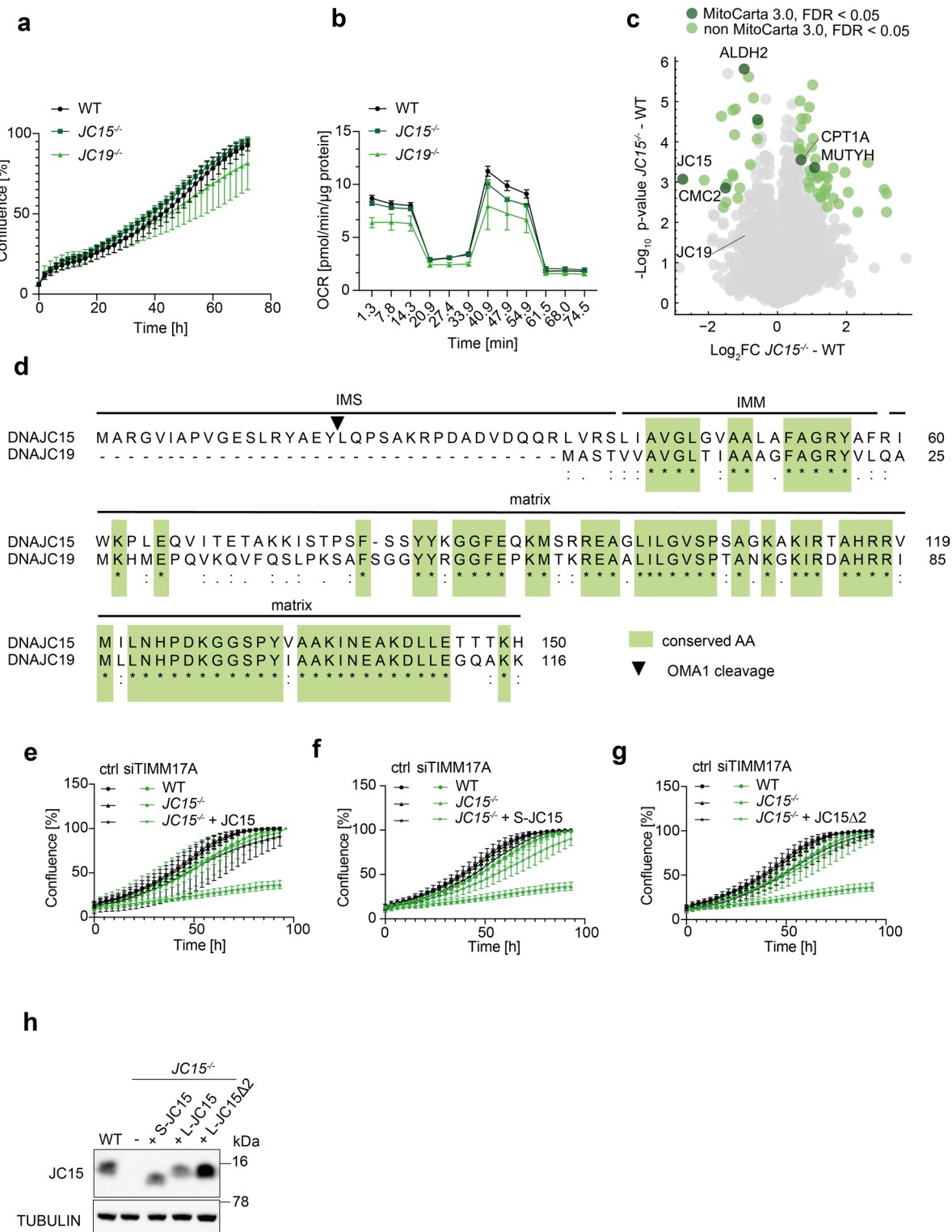

**Extended Data Fig. 2 | Genetic interactions of DNAJC15 with DNAJC19 and TIMM17A. a**, Cell growth of wild-type (WT), *DNAJC15*[−/−], and *DNAJC19*[−/−] HeLa cells (n = 3, biologically independent samples). Data are presented as means ± s.d. **b**, Oxygen consumption rate (OCR) of WT, *DNAJC15*[−/−], and *DNAJC19*[−/−] cells after the addition of oligomycin, FCCP, and antimycin A and rotenone (n = 3, biologically independent samples). **c**, Volcano plot of cellular proteome of WT and *DNAJC15*[−/−] HeLa cells. Significantly affected proteins (FDR < 0.05) are shown in green, significantly changed MitoCarta 3.0-annotated proteins in dark green (n = 5, biologically independent samples). *P* values (y-axis) were calculated using

an unpaired two-sided *t*-test. **d**, Sequence alignment of human DNAJC15 and DNAJC19. Conserved amino acids are highlighted in green. IMS, intermembrane space; IMM, inner mitochondrial membrane. **e**–**g**, Growth of wild-type (WT) cells, *DNAJC15*[−/−] cells and *DNAJC15*[−/−] cells expressing (e) DNAJC15 (JC15), (f) cleaved DNAJC15 (S-JC15) or (g) DNAJC15Δ2 (JC15Δ2), which were treated with siRNA targeting TIMM17A for 72 h (n = 3, biologically independent samples). Data are presented as means ± s.d. **h**, Expression of DNAJC15 variants in different HeLa cell lines monitored by SDS-PAGE and immunoblotting.

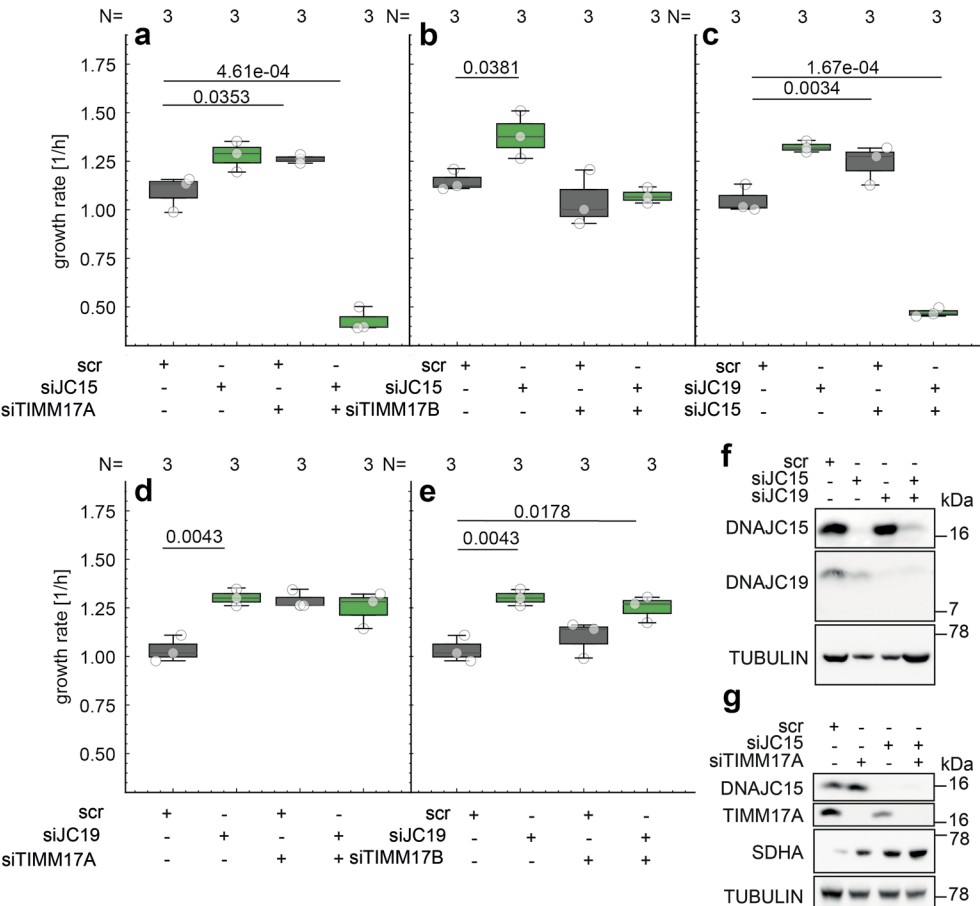

**Extended Data Fig. 3 | Confirmation of the genetic interaction by using acute depletion models. a–e,** Cell growth of wild-type (WT) HeLa cells after siRNA-mediated depletion of DNAJC15 (siJC15) and TIMM17A (siTIMM17A) for 72 h in (a), DNAJC15 and TIMM17B (siTIMM17B) in (b), DNAJC15 and DNAJC19 (siJC19) in (c), DNAJC19 and TIMM17A in (d) and DNAJC19 and TIMM17B in (e). Data are means ± s.d. *P* values were calculated using an unpaired two-sided *t*-test. Quantile box plot show median (center line), 25th and 75th percentiles, whiskers show minimum and maximum values (1.5 * IQR distance from median, outliers not shown, n = 3, biologically independent samples). **f,** Expression of DNAJC15 and DNAJC19 in (c) after siRNA-mediated depletion monitored by SDS-PAGE and immunoblotting. **g,** Expression of DNAJC15 and TIMM17A in (a) after siRNA-mediated depletion monitored by SDS-PAGE and immunoblotting.

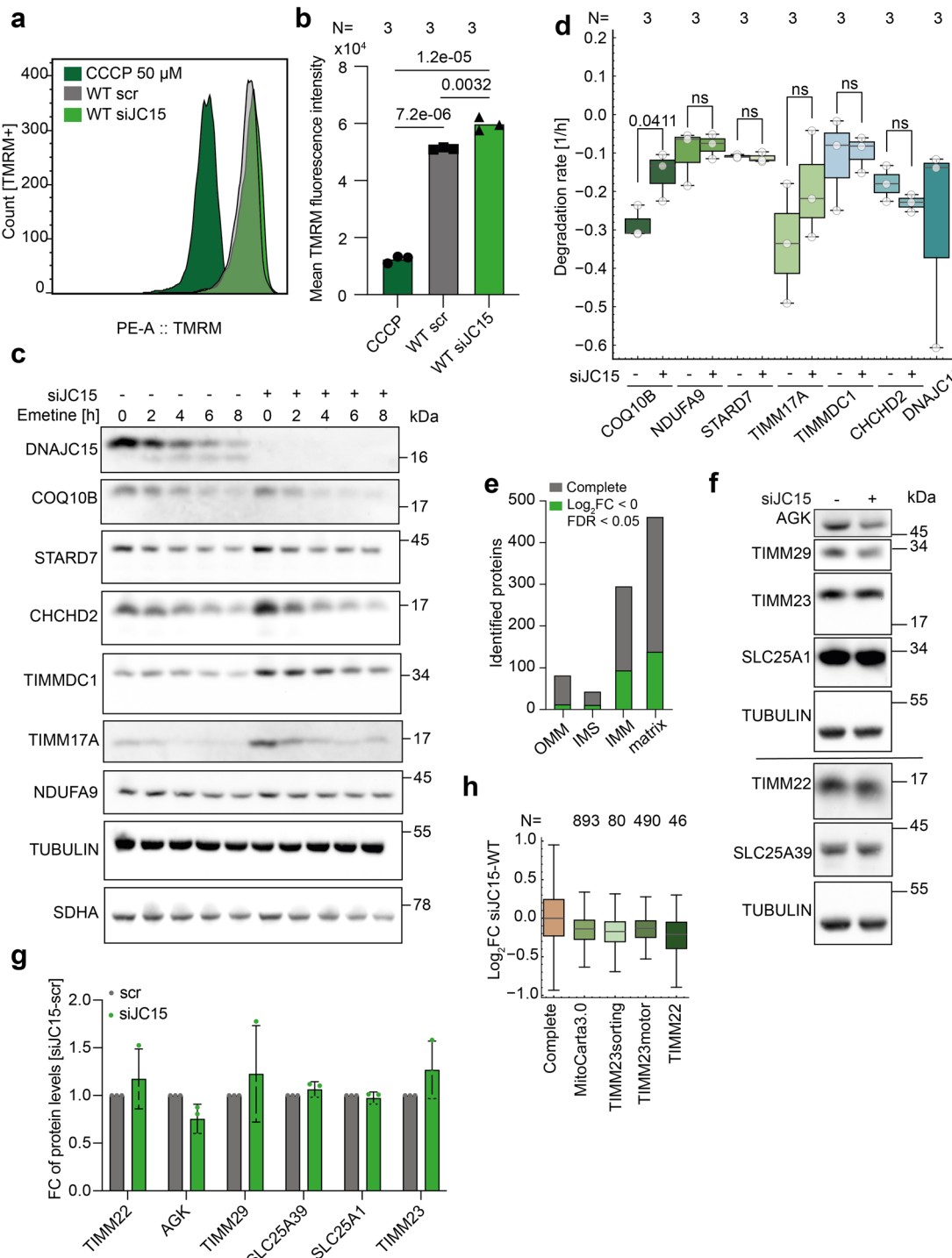

**Extended Data Fig. 4 | DNAJC15 does not affect MMP, TIM22 levels and protein turnover rates. a**, Histogram plot of the TMRM fluorescence levels in wild-type (WT) HeLa cells after siRNA-mediated depletion of DNAJC15 (siJC15) or CCCP treatment for 2 h (50 µM). A representative experiment is shown (n = 3, biologically independent samples). Single cells were identified using a conventional FSC/SSC gating strategy, and the mean fluorescence intensity of the gated population was quantified. **b**, Quantification of the mean TMRM fluorescence intensity in (a). Data are means ± s.d. (n = 3, biologically independent samples). *P* values were calculated using a two-sided unpaired *t*-test. **c**, Stability of several mitochondrial proteins in wild-type (WT) HeLa cells depleted of DNAJC15 (siJC15) for 48 h after inhibition of cytosolic translation with emetine. A representative experiment is shown (n = 3, biologically independent samples). **d**, Degradation rates of the proteins in (c) normalized to time point 0 h. Data are means ± s.d. (n = 3, biologically independent samples). *P* values were calculated using a two-sided unpaired *t*-test. Quantile box plot show median

(center line), 25th and 75th percentiles, whiskers show minimum and maximum values (1.5 * IQR distance from median, outliers not shown). **e**, Fraction of significantly changed mitochondrial proteins (FDR < 0.05) within all identified proteins of various mitochondrial subcompartments. **f**, Expression levels of TIM22 components and TIMM22 substrates in wild-type (WT) HeLa cells after siRNA-mediated depletion of DNAJC15 (siJC15) monitored by Western Blot. **g**, Quantification of the protein levels in (f). Data are means ± s.d. (n = 3, biologically independent samples). *P* values were calculated using a two-sided unpaired *t*-test. **h**, Boxplot visualizing the distribution of MitoCarta 3.0-annotated proteins according to their mitochondrial protein import in ref. 45. Quantile box plot show median (center line), 25th and 75th percentiles, whiskers show minimum and maximum values after outlier removal (1.5 * IQR distance from median, outliers not shown). Mann-Whitney *U*-test was performed (n = 5, biologically independent samples, including outliers).

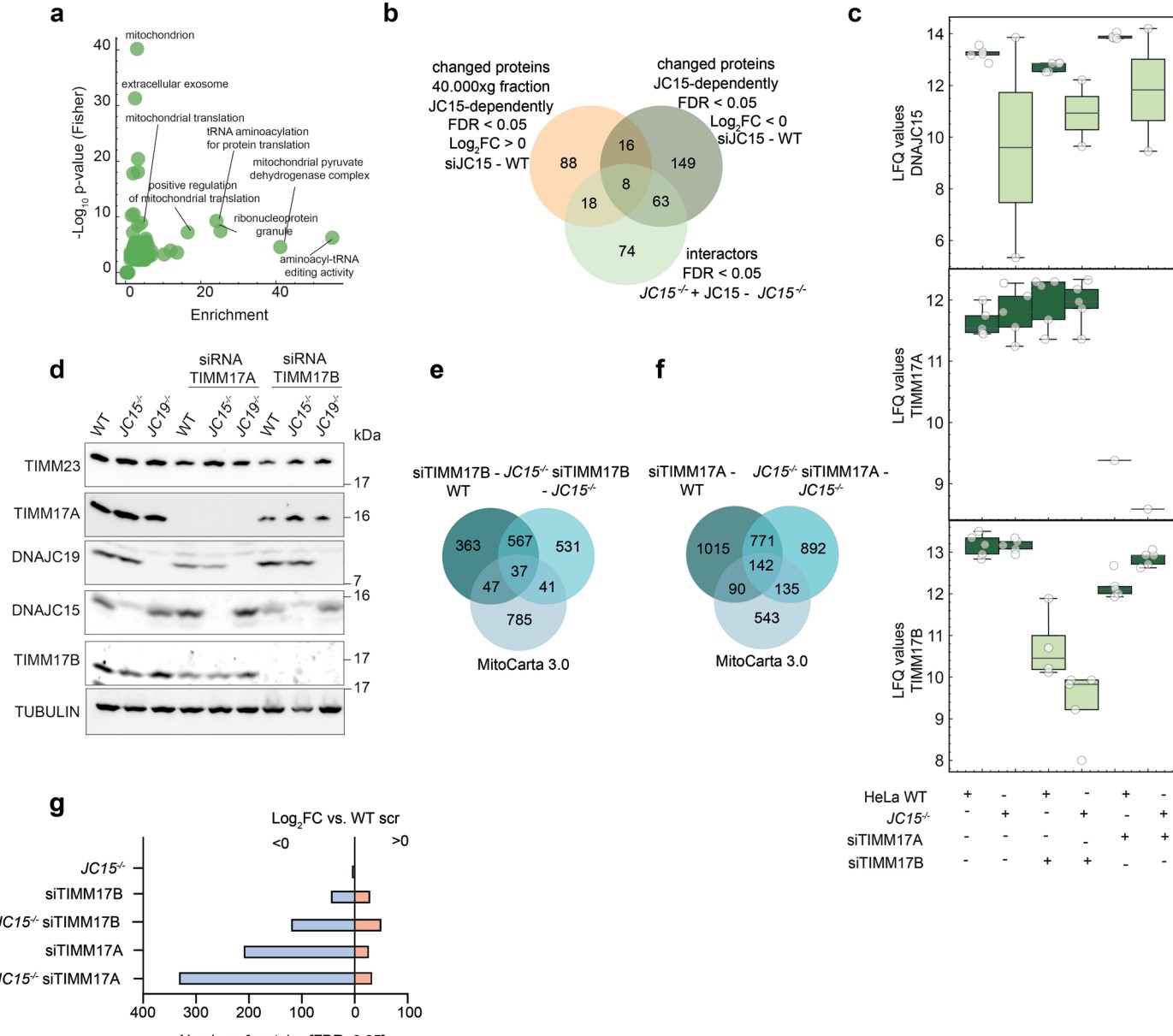

**Extended Data Fig. 5 | DNAJC15 and TIMM17A depletion affects OXPHOS biogenesis. a**, Gene ontology enrichment analysis of all significant protein groups in the mitochondrial, Proteinase K-treated fraction after DNAJC15 depletion relative to WT (FDR < 0.02, Fisher exact test, corrected by Benjamini Hochberg procedure). **b**, Venn-diagram visualizing the overlap of significantly changed mitochondrial proteins after DNAJC15 depletion in HeLa cells (mitochondrial fraction treated with Proteinase K; Fig. 4b, n = 5, biologically independent samples), DNAJC15 interactors (Fig. 3d) (FDR < 0.05, n = 4, biologically independent samples) and significantly changed mitochondrial proteins after DNAJC15 depletion in HeLa cells (40.000xg fraction; Fig. 6b, n = 5, biologically independent samples). **c**, Boxplot visualizing the label-free quantification (LFQ) values of DNAJC15, TIMM17A and TIMM17B levels after siRNA-mediated depletion. Data are means ± s.d. Quantile box plot show median (center line), 25th and 75th percentiles, whiskers show minimum and

maximum values. (1.5 * IQR distance from median, outliers not shown, n = 5, biologically independent samples). **d**, Expression levels of proteins in wild-type (WT), *DNAJC19⁻/⁻* and *DNAJC15⁻/⁻* HeLa cells after siRNA-mediated depletion of TIMM17A (siTIMM17A) and TIMM17B (siTIMM17B) monitored by SDS-PAGE and immunoblotting. **e**, Venn diagram visualizing the overlap of the significantly changed proteins of WT and *DNAJC15⁻/⁻* cells, treated with siRNA targeting TIMM17B, and MitoCarta 3.0-annotated protein groups (FDR < 0.05). **f**, Venn diagram visualizing the overlap of the significantly changed proteins of WT and *DNAJC15⁻/⁻* cells, treated with siRNA targeting TIMM17A, and MitoCarta 3.0-annotated protein groups (FDR < 0.05). **g**, Boxplot visualizing all significant mitochondrial proteins (FDR < 0.05, MitoCarta 3.0) compared to all identified proteins in the whole cell fraction in HeLa WT and *DNAJC15⁻/⁻* cells, treated with siRNA targeting TIMM17A or TIMM17B.

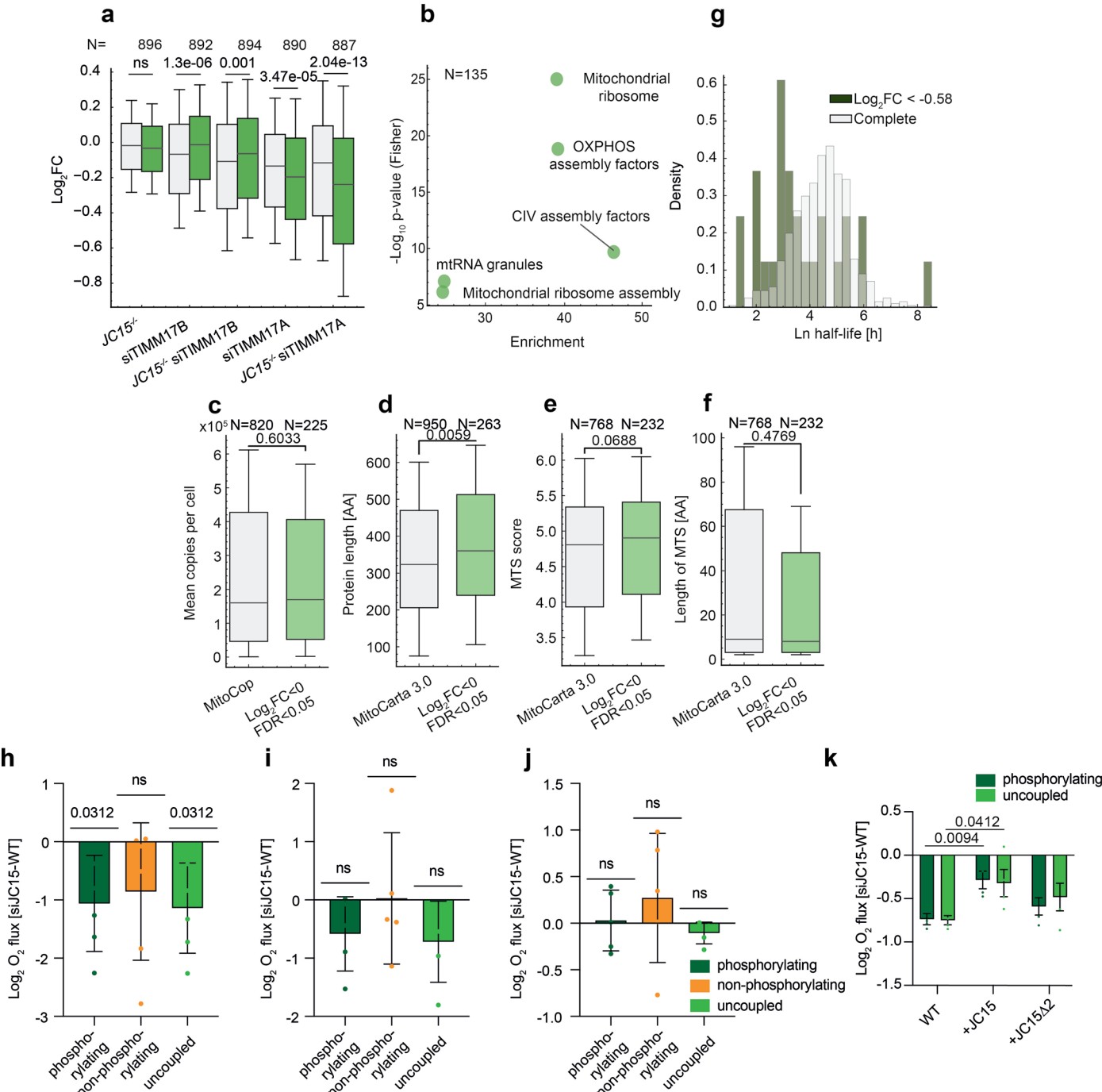

**Extended Data Fig. 6 | Acute DNAJC15 depletion affects protein import capacity and OXPHOS function.** The quantile box plot of all presented boxplots show median (center line), 25th and 75th percentiles, whiskers show minimum and maximum values after outlier removal (0.5 * IQR distance from median, outliers not shown). Two-sided Mann-Whitney *U*-test was performed (n = 5, biologically independent samples, including outliers). **a**, Boxplot visualizing the distribution of mitochondrial proteins (MitoCarta 3.0) compared to all identified proteins in the whole cell fraction in HeLa WT and *DNAJC15*[-/-] cells, treated with siRNA targeting TIMM17A or TIMM17B. **b**, MitoPathways enrichment analysis of the significant mitochondrial (according to MitoCarta 3.0) proteins, that are specifically affected in *DNAJC15*[-/-] treated with siRNA targeting TIMM17A (FDR < 0.05) (Fisher exact test, corrected by Benjamini Hochberg procedure, FDR < 0.02). **c-f**, Boxplot visualizing the distribution of significantly downregulated proteins in the mitochondrial fraction (log₂FC < 0, FDR < 0.05) compared to all MitoCop or MitoCarta 3.0-annotated fractions.

The distribution indicates the mean copies per cell, extracted from Morgenstern et al.[31], in **c**, the protein length in **d**, the highest calculated MTS score in **e** and the length of the MTS with the highest MTS score in **f**, both extracted from the MTSviewer platform. **g**, Density plot visualizing the Ln half-life distribution of the mostly affected proteins (log₂FC < -0.58) (n = 5, biologically independent samples). Half-life was extracted from Morgenstern et al.[31]. **h–j**, Quantification of the oxygen flux for Fig. 5a in **h**, Fig. 5d in **i** and Fig. 5e in **j**. Statistical significance in the log₂ fold-changes in the oxygen flux between DNAJC15-depleted cells and WT cells was assessed using a paired two-sided t-test. Data are means ± SD (n = 5 for **h**, n = 4 for **i** and **j**, biologically independent samples). **k**, Statistical analysis of Fig. 5b, c of wild-type (WT) and siRNA-resistant DNAJC15Δ2 (JC15Δ2), or DNAJC15 (JC15) complemented HeLa cells after treatment with siRNA-mediated depletion of DNAJC15 (siJC15) compared to wild-type HeLa cells in phosphorylating and uncoupled conditions. Data are means ± s.d. (n = 4, biologically independent samples). *P* values were calculated using a two-sided unpaired *t*-test.

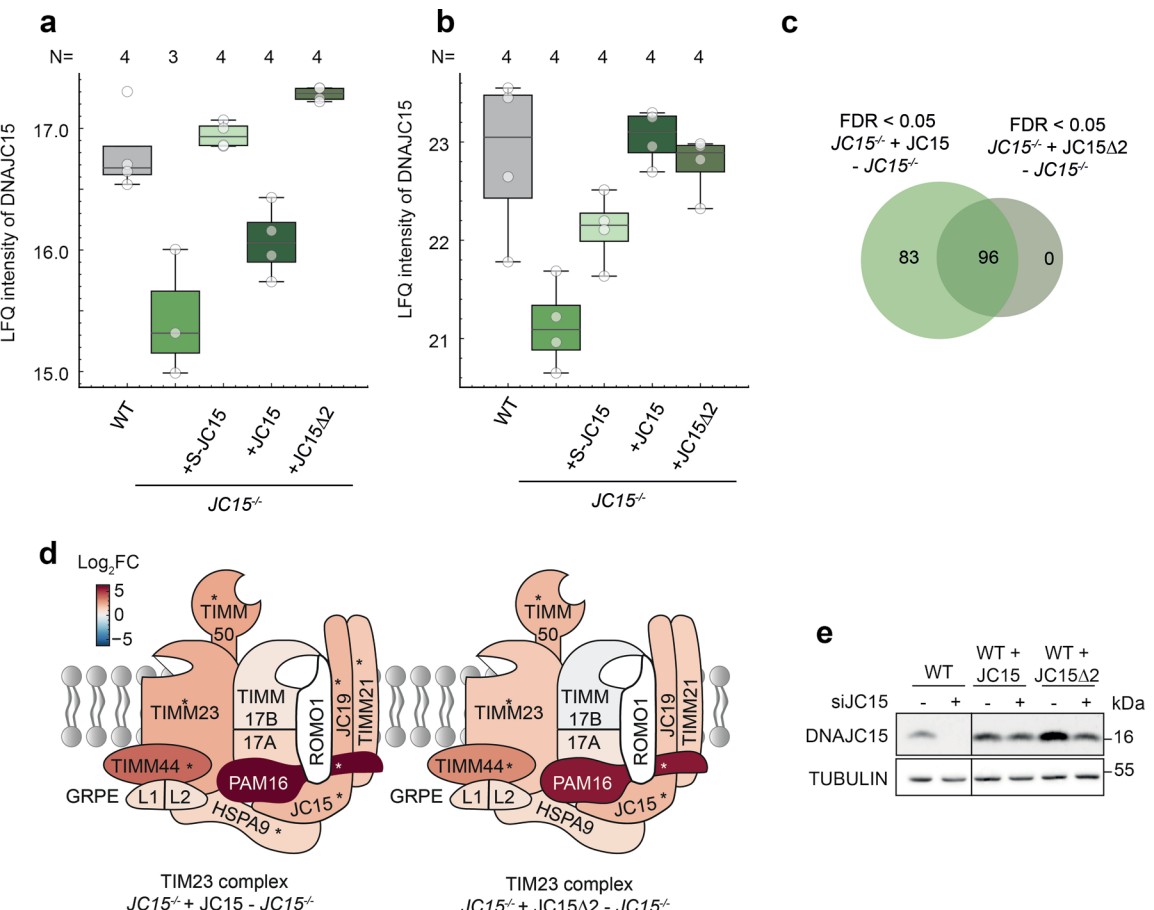

**Extended Data Fig. 7 | The interactome of DNAJC15Δ2. a**, Boxplot visualizing the label-free quantification (LFQ) intensity of all identified DNAJC15 peptides in the different cell lines from the input fraction after crosslinking with 2 mM DSP (n = 4, biologically independent samples). Quantile box plot show median (center line), 25th and 75th percentiles, whiskers show minimum and maximum values. **b**, Boxplot visualizing the LFQ intensity of all identified DNAJC15 peptides in the different cell lines in the elution fraction after crosslinking with 2 mM DSP (n = 4, biologically independent samples). Quantile box plot show median (center line), 25th and 75th percentiles, whiskers show minimum and maximum values. **c**, Venn diagram visualizing the overlap of the significantly identified interactors

of DNAJC15 (JC15) and DNAJC15Δ2 (JC15Δ2) (FDR < 0.05). **d**, Schematic visualization of the TIM23 complex subunits interacting with DNAJC15. The scale indicates fold-enrichment between *DNAJC15^{-/-}* cells expressing DNAJC15 or DNAJC15Δ2 relative to *DNAJC15^{-/-}* cells (log₂FC). The asterisk indicates all significantly affected proteins (FDR < 0.05, n = 4, biologically independent samples). **e**, Expression levels of DNAJC15 protein in wild-type (WT) expressing DNAJC15 (JC15) and non-cleavable DNAJC15 (JC15Δ2) after siRNA-mediated depletion of DNAJC15 (siJC15) in (Fig. 5h, i) monitored by SDS-PAGE and immunoblotting.

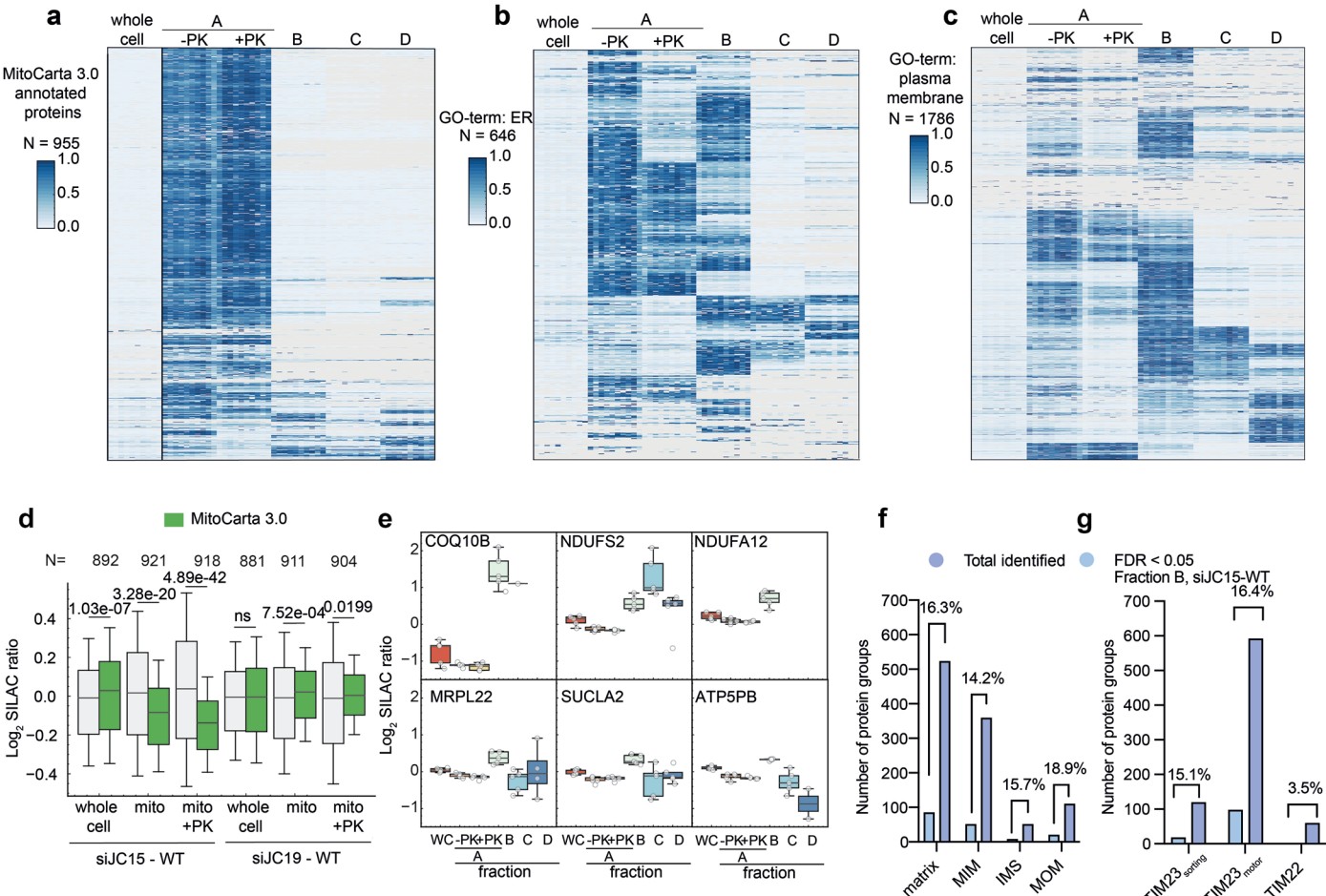

**Extended Data Fig. 8 | Mitochondrial proteins relocalize to the ER. a**, Heat map showing the intensity-based absolute quantification values of all MitoCarta 3.0-annotated proteins in all extracted fractions according to Fig. 6a (n = 5, biologically independent samples). **b**, Heat map showing the intensity-based absolute quantification values of all proteins annotated with the GO-term 'ER' in all extracted fractions according to Fig. 6a (n = 5, biologically independent samples). **c**, Heat map showing the intensity-based absolute quantification values of all proteins annotated with the GO-term 'plasma membrane' in all extracted fractions according to Fig. 6a (n = 5, biologically independent samples). **d**, Boxplot visualizing the distribution of selected mitochondrial proteins (MitoCarta 3.0) and all identified proteins in the whole cell and in the mitochondrial fractions, which were treated with Proteinase K (PK) when indicated. Two-sided Mann-Whitney *U*-test was performed. Quantile box plot show median, 25th and 75th percentiles, whiskers show minimum and maximum values after outlier removal (defined as distance from median 0.5 * IQR, outliers not shown) (n = 5, biologically independent samples). **e**, Panel of significantly downregulated proteins in the mitochondrial fraction treated with PK, that were also significantly upregulated in the 40,000xg fraction upon depletion of DNAJC15 (FDR < 0.05, n = 5, biologically independent samples). Quantile box plot show median (center line), 25th and 75th percentiles, whiskers show minimum and maximum values (1.5 * IQR distance from median, outliers not shown). **f**, Fraction of significantly changed mitochondrial proteins (FDR < 0.05) within all identified proteins according to the submitochondrial localization of MitoCarta 3.0. **g**, Fraction of significantly changed mitochondrial proteins (FDR < 0.05) within all identified proteins according to their mitochondrial protein import[45].

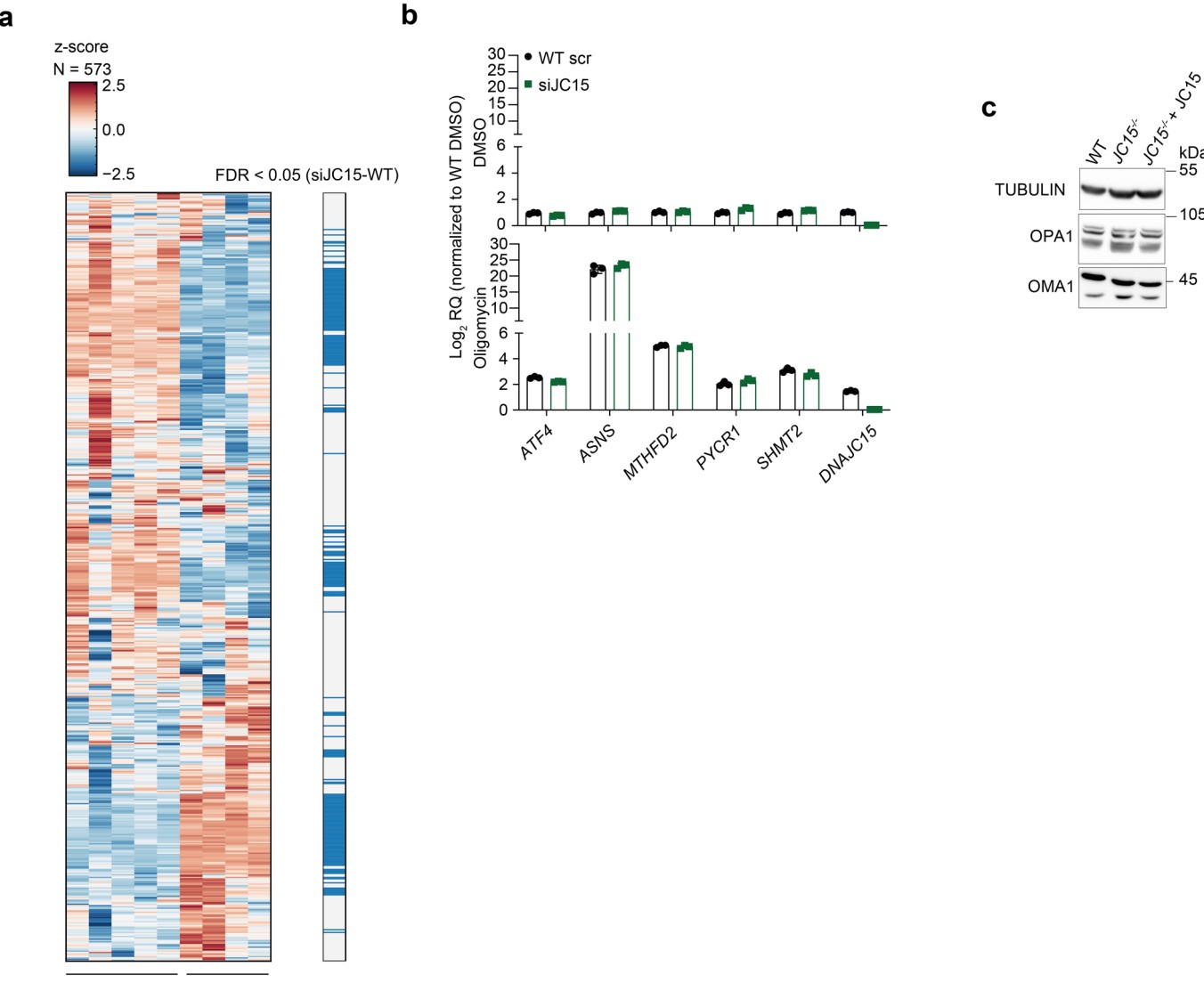

**Extended Data Fig. 9 | DNAJC15 depletion does not induce an integrated stress response. a**, Heat map of ATF4-dependent target genes whose expression in DNAJC15-depleted relative to wild-type (WT) cells (log$_2$FC) is shown. Genes whose expression was significantly changed after DNAJC15 depletion (adjusted p < 0.05, unpaired *t*-test followed by Benjamini Hochberg correction, FDR < 0.05) are marked in blue (n = 5 for wildtype cells, n = 4 for siJC15 cells, biologically independent samples). **b**, log$_2$ RQ values, normalized to WT cells treated with DMSO (upper panel) or oligomycin (10 μM, lower panel), for cells depleted of DNAJC15 for 48 h (n = 3, biologically independent samples). **c**, Steady-state levels of OMA1 and OPA1 in wild-type (WT), *DNAJC15$^{-/-}$* and *DNAJC15$^{-/-}$* cells expressing DNAJC15 (JC15) analyzed by SDS-PAGE and immunoblotting (n = 3, biologically independent samples).

# Reporting Summary

## Statistics

For all statistical analyses, confirm that the following items are present in the figure legend, table legend, main text, or Methods section.

| n/a | Confirmed | |
|---|---|---|
| ☐ | ☒ | The exact sample size (*n*) for each experimental group/condition, given as a discrete number and unit of measurement |
| ☐ | ☒ | A statement on whether measurements were taken from distinct samples or whether the same sample was measured repeatedly |
| ☐ | ☒ | The statistical test(s) used AND whether they are one- or two-sided *Only common tests should be described solely by name; describe more complex techniques in the Methods section.* |
| ☒ | ☐ | A description of all covariates tested |
| ☐ | ☒ | A description of any assumptions or corrections, such as tests of normality and adjustment for multiple comparisons |
| ☐ | ☒ | A full description of the statistical parameters including central tendency (e.g. means) or other basic estimates (e.g. regression coefficient) AND variation (e.g. standard deviation) or associated estimates of uncertainty (e.g. confidence intervals) |
| ☐ | ☒ | For null hypothesis testing, the test statistic (e.g. *F*, *t*, *r*) with confidence intervals, effect sizes, degrees of freedom and *P* value noted *Give P values as exact values whenever suitable.* |
| ☒ | ☐ | For Bayesian analysis, information on the choice of priors and Markov chain Monte Carlo settings |
| ☒ | ☐ | For hierarchical and complex designs, identification of the appropriate level for tests and full reporting of outcomes |
| ☒ | ☐ | Estimates of effect sizes (e.g. Cohen's *d*, Pearson's *r*), indicating how they were calculated |

*Our web collection on statistics for biologists contains articles on many of the points above.*

## Software and code

Policy information about availability of computer code

| Data collection | For mass spec experiments, XCalibur (v.4.5, Thermo Fisher) and Tune (v.4.0.309,Thermo Fisher) was used. Cell growth data was acquired using the Incucyte System (V.2022B.Rev2). Mitochondrial respiration was acquired and analyzed using the Seahorse Agilent Software Wave (V2.2.4.3) or the Oroboros oxygraph2k DatLab (V.6.1.0.7). Western Blot Images were acquired using the Intas ChemoStar ECL Imager HR6.9. |
|---|---|
| Data analysis | Data was analyzed using InstantClue software (v.0.12.2, https://github.com/hnolCol/instantclue). Mass spec data was analyzed using Spectronaut (18.7.240325.55695). Pair-wise differential gene expression was performed using DESeq2/1.24.0. Raw file analysis was analyzed using MaxQuant (1.6.7 and 1.6.12). Incucyte Analysis Software (V.2022.Rev2) was used for cell growth assay analysis, visualized with GraphPadPrism Version 10.4.1(532). Seahorse data was analyzed by Seahorse Wave Desktop Software (V.2.2.4.3). Western Blot quantification was acquired using ImageJ (version: 2.1.0/1.53c) and visualized with GraphPadPrism Version 10.4.1(532). |

For manuscripts utilizing custom algorithms or software that are central to the research but not yet described in published literature, software must be made available to editors and reviewers. We strongly encourage code deposition in a community repository (e.g. GitHub). See the Nature Portfolio guidelines for submitting code & software for further information.

## Data

Policy information about availability of data

All manuscripts must include a data availability statement. This statement should provide the following information, where applicable:
- Accession codes, unique identifiers, or web links for publicly available datasets
- A description of any restrictions on data availability
- For clinical datasets or third party data, please ensure that the statement adheres to our policy

RNA sequencing data was uploaded to Gene Expression Omnibus (GEO) and is available under the accession number (link: https://www.ncbi.nlm.nih.gov/geo/query/acc.cgi?acc=GSE299431, ID: GSE299431. The proteomics data were uploaded to the following PRIDE projects under the IDs: The cycloheximide chase in WT and AFG3L2-/- cells proteomics data are available under the identifier (link: https://www.ebi.ac.uk/pride/archive/projects/ PXD056318, ID: PXD056318). The Neo-N termiome of OMA1-/- HeLa cells are available under the identifier (link: https://www.ebi.ac.uk/pride/archive/projects/PXD061134, ID: PXD061134. The Neo-N termiome of AFG3L2-/-, SPG7-/-, AFG3L2-/- SPG7-/-, and WT HeLa cells is available on PRIDE under the identifier (link: https://www.ebi.ac.uk/pride/archive/projects/PXD061137, ID: PXD061137. The SILAC-based subcellular fraction proteomics experiment is available under the identifier: (link: https://www.ebi.ac.uk/pride/archive/projects/PXD061449, ID: PXD061449, and (link: https://www.ebi.ac.uk/pride/archive/projects/PXD061185, ID: PXD061185. TIMM17A and TIMM17B knock-down in WT and DNAJC15-/-, HeLa cells were deposited to PRIDE under the identifier (link: https://www.ebi.ac.uk/pride/archive/projects/PXD061131, ID: PXD061131. The DNAJC15 DSP crosslinking interactome is available under the PRIDE identifier (link: https://www.ebi.ac.uk/pride/archive/projects/PXD061165, ID: PXD061165. The DNAJC15 DSP crosslinking in AFG3L2-/- SPG7-/- cells is available under the PRIDE identifier (link: https://www.ebi.ac.uk/pride/archive/projects/PXD070836, ID: PXD070836. The SILAC-based subcellular fraction proteomics including carbonate extraction is available under the identifier (link: https://www.ebi.ac.uk/pride/archive/projects/PXD070854, ID: PXD070854). The expression proteomics of DNAJC15-/- HeLa cells is deposited to PRIDE under the identifier (link: https://www.ebi.ac.uk/pride/archive/projects/PXD072822, ID:PXD072822). All other data supporting the findings of this study are available from the corresponding author on reasonable request. Source data are available with the manuscript online.

## Research involving human participants, their data, or biological material

Policy information about studies with human participants or human data. See also policy information about sex, gender (identity/presentation), and sexual orientation and race, ethnicity and racism.

| | |
|---|---|
| Reporting on sex and gender | *Use the terms sex (biological attribute) and gender (shaped by social and cultural circumstances) carefully in order to avoid confusing both terms. Indicate if findings apply to only one sex or gender; describe whether sex and gender were considered in study design; whether sex and/or gender was determined based on self-reporting or assigned and methods used.*<br>*Provide in the source data disaggregated sex and gender data, where this information has been collected, and if consent has been obtained for sharing of individual-level data; provide overall numbers in this Reporting Summary. Please state if this information has not been collected.*<br>*Report sex- and gender-based analyses where performed, justify reasons for lack of sex- and gender-based analysis.* |
| Reporting on race, ethnicity, or other socially relevant groupings | *Please specify the socially constructed or socially relevant categorization variable(s) used in your manuscript and explain why they were used. Please note that such variables should not be used as proxies for other socially constructed/relevant variables (for example, race or ethnicity should not be used as a proxy for socioeconomic status).*<br>*Provide clear definitions of the relevant terms used, how they were provided (by the participants/respondents, the researchers, or third parties), and the method(s) used to classify people into the different categories (e.g. self-report, census or administrative data, social media data, etc.)*<br>*Please provide details about how you controlled for confounding variables in your analyses.* |
| Population characteristics | *Describe the covariate-relevant population characteristics of the human research participants (e.g. age, genotypic information, past and current diagnosis and treatment categories). If you filled out the behavioural & social sciences study design questions and have nothing to add here, write "See above."* |
| Recruitment | *Describe how participants were recruited. Outline any potential self-selection bias or other biases that may be present and how these are likely to impact results.* |
| Ethics oversight | *Identify the organization(s) that approved the study protocol.* |

Note that full information on the approval of the study protocol must also be provided in the manuscript.

# Field-specific reporting

Please select the one below that is the best fit for your research. If you are not sure, read the appropriate sections before making your selection.

☒ Life sciences        ☐ Behavioural & social sciences        ☐ Ecological, evolutionary & environmental sciences

For a reference copy of the document with all sections, see nature.com/documents/nr-reporting-summary-flat.pdf

# Life sciences study design

All studies must disclose on these points even when the disclosure is negative.

| | |
|---|---|
| Sample size | No sample size was calculated. No statistical test was performed to determine the sample size. The sample size included 3 biological independent replicates for Western Blot analysis, on which a statistical test was performed on. The sample size included 4 biological replicates |

for the crosslinking experiment, and 5 biological replicates for all other remaining mass spec data. The sample size was determined by experience.

| | |
|---|---|
| Data exclusions | Replicate 5 of the siRNA DNAJC15 treated samples of the RNA sequencing were excluded, as they did not pass the quality check due to fewer reads. |
| Replication | All replicates are shown separately in each figure. All attempts in repeating the experiments were successful and included in the figure. |
| Randomization | Proteomics data were measured in random order. For cell culture experiments, samples were allocated in different groups either based on genotypes or based on different treatments at different time points. Whenever possible, different experimental groups were handled together. Covariates do not apply to cell culture experiments, since all the experiments were performed in the same cell culture room, all the cells were cultured in the same DMEM medium and were put in the same incubator. Samples were allocated to experimental groups according to predefined experimental conditions rather than randomization. Covariate control was not applicable, as all samples were derived from a homogeneous source and were identical with respect to relevant biological and technical variables. |
| Blinding | Proteomics and RNA sequencing experiments were blinded. Tissue culture experiments were not blinded. The first author had to label the genotypes and treatement/time points in tissue culture. |

# Reporting for specific materials, systems and methods

We require information from authors about some types of materials, experimental systems and methods used in many studies. Here, indicate whether each material, system or method listed is relevant to your study. If you are not sure if a list item applies to your research, read the appropriate section before selecting a response.

### Materials & experimental systems

| n/a | Involved in the study |
|---|---|
| ☐ | ☒ Antibodies |
| ☐ | ☒ Eukaryotic cell lines |
| ☒ | ☐ Palaeontology and archaeology |
| ☒ | ☐ Animals and other organisms |
| ☒ | ☐ Clinical data |
| ☒ | ☐ Dual use research of concern |
| ☒ | ☐ Plants |

### Methods

| n/a | Involved in the study |
|---|---|
| ☒ | ☐ ChIP-seq |
| ☒ | ☐ Flow cytometry |
| ☒ | ☐ MRI-based neuroimaging |

## Antibodies

| | |
|---|---|
| Antibodies used | DNAJC15 (ProteinTech, Cat# 16063-1-AP, 1:500), TUBULIN (Sigma, Cat# T6074, 1:2000), AFG3L2 (Sigma, Cat# HPA004480, 1:1000), ACTIN (Santa Cruz, Cat# SC-47778, 1:3000), SDHA (abcam, Cat# ab14715, 1:5000), OMA1 (Santa Cruz, Cat# SC-515788; 1:1,000). Corresponding species-specific HRP-coupled antibodies were used for immunoblot (Biorad; Cat# 1706515 and Cat# 1706516, 1:10000 dilution). |
| Validation | DNAJC15, AFG3L2, OMA1 antibodies were validated by observing no bands in the knockout cell line of those proteins. SDHA antibody has been validated before in our lab (Saita et al., 2017 Nature Cell Biology). ACTIN has been validated in our lab (MacVicar et al., 2020, Nature Cell Biology). Tubulin has been verified by the manufacturer´s: https://www.sigmaaldrich.com/DE/de/product/sigma/t6074 |

## Eukaryotic cell lines

Policy information about cell lines and Sex and Gender in Research

| | |
|---|---|
| Cell line source(s) | HeLa (CCL-2) cells were purchased from ATCC. Information regarding generation of knockout cell line details are included in the Methods under CRISPR-Cas9 gene editing section. |
| Authentication | Cell lines were not authenticated |
| Mycoplasma contamination | All the cell lines were routinely tested for mycoplasma contamination and only if tested negative, used for experiments. |
| Commonly misidentified lines (See ICLAC register) | No commonly misidentified lines were used in this study. |

# Plants

Seed stocks

*Report on the source of all seed stocks or other plant material used. If applicable, state the seed stock centre and catalogue number. If plant specimens were collected from the field, describe the collection location, date and sampling procedures.*

Novel plant genotypes

*Describe the methods by which all novel plant genotypes were produced. This includes those generated by transgenic approaches, gene editing, chemical/radiation-based mutagenesis and hybridization. For transgenic lines, describe the transformation method, the number of independent lines analyzed and the generation upon which experiments were performed. For gene-edited lines, describe the editor used, the endogenous sequence targeted for editing, the targeting guide RNA sequence (if applicable) and how the editor was applied.*

Authentication

*Describe any authentication procedures for each seed stock used or novel genotype generated. Describe any experiments used to assess the effect of a mutation and, where applicable, how potential secondary effects (e.g. second site T-DNA insertions, mosiacism, off-target gene editing) were examined.*

