## [Peer Review File · Nature Structural & Molecular Biology]

Stress adaptation of mitochondrial protein import by OMA1-mediated degradation of DNAJC15

Corresponding Author: Professor Thomas Langer

Version 0:

Reviewer comments:

Reviewer #1

(Remarks to the Author)

The study by Kroczek et al. (2025) presents a detailed omics-enriched investigation into the role of DNAJC15 in the mitochondrial protein biogenesis. The authors identify DNAJC15 as a substrate of OMA1, which, upon cleavage, undergoes degradation by the m-AAA protease. The authors conclude that the loss of DNAJC15 alters mitochondrial protein import via the TIM23 complex, containing TIMM17A, affecting the respiratory chain biogenesis, leading to the subsequent accumulation of preproteins at the endoplasmic reticulum (ER) and triggering an ATF6-mediated stress response.

The study provides the strong biochemical evidence regarding DNAJC15 processing and degradation, and also examines the homolog JC19. However, the authors introduce multiple hypotheses about its role in OXPHOS biogenesis, altered protein import, and stress responses but the appropriate evidence is missing. This broad scope results in a somewhat fragmented narrative. The study should be strengthened by biochemically linking DNAJC15 function to mitochondrial import and its downstream effects, such as the suggested changes in OXPHOS and in cellular stress pathways.

Major points:

1. Adaptation of DNAJC15 Knock-Out Cells

Mutant DNAJC15^{-/-} and depletion behave opposite in different aspects, raising the question of whether the proposed effect on the protein import via the TIMM17A containing TIM23 complex is due to the adaptation in the knock-out cells. To address this point, authors should compare knock-out and knock-down models side by side and show which phenotype can be reproduced in a double knock-down mutant of TIMM17A and JC15. The adaptations and different phenotypes observed for KOs and depletions are relatively commonly seen, however this should be addressed along specificity of the observed effects.

2. Protein import via DNAJC15

Authors conclude “from these experiments that DNAJC15 mediates the import of many matrix and IM proteins”. This part requires more evidence such as:

-using in vitro imports to show changes in import of matrix and IM proteins and their preferential dependence on TIM17A and DNAJC15 versus TIMM17B.

- using appropriate controls of a membrane potential and the TIMM22 to show that the overall quality of mitochondria is not impaired.

Another point that requires additional evidence is differentiation between protein turnover and protein import. A protein turnover rate in DNAJC15 mutants should be examined.

3. Precursor accumulation and mislocalization to the ER and UPR activation

The study suggests that mitochondrial import defects caused by DNAJC15 loss lead to precursor accumulation in the cytosol and subsequent mislocalization to the ER, triggering an ATF6-mediated UPR. However, the direct evidence for these events is lacking.

The authors are recommended to use some of the suggested ways for providing more evidence on this interesting concept: show that mitochondrial proteins (substrates of DNAJC15) accumulate not in mitochondria by fractionation and/or proximity labeling approaches. Is this potential localization on the ER surface caused by a closer tethering of the ER and mitochondria?

The authors should also confirm the UPR activation with additional evidence beyond transcriptomics.

Minor points and recommendations:

- Show that L-JC15 interacts with TIM23 not S-LC15, using IPs.
- Can JC19 take over for JC15? Does the overexpression of JC19 rescue JC15^{-/-}?
- Check if knock-down of components of complex I and coenzyme Q synthesis would have the same effect as TIMM17A knock-downs in JC15^{-/-}

- Label JC19 in Vulcano blot of JC15^{-/-}, show if there is a general shift in mitochondrial proteins by labeling all mitochondrial proteins in volcano blot, since authors say it plays a role in mitochondrial import
- Comment why the S-JC15 can rescue siTIMM17a in JC15^{-/-}, but rescue is only partial in the KO mutant – could the already mentioned adaptations be the reason for it?
- Mark proteins identified and mentioned in the text also in the interactome of DNAJC15 IP (Fig. 3A)
- The authors state that matrix and IMM proteins are most affected by DNAJC15 depletion, suggesting a specific impairment in TIM23-dependent import. However: 1/3 of IMS proteins are also reduced, which suggests that the defect is broader than just matrix/IMM proteins.
- The recent papers from the Chacinska and Youle groups are not cited. Given the focus of this work, these papers provide an additional introductory information on the topic of TIM23 protein import regulation.

Reviewer #2

(Remarks to the Author)

The article describes the OMA1-mediated degradation of DNAJC15, which regulates its functions in mitochondria protein import. The authors demonstrate that the activation of OMA1 often occurs during mitochondrial stress, inducing the cleavage of DNAJC15 (L-DNAJC15) into a shorter version, S-DNAJC15. This shorter version is subsequently degraded by the m-AAA protease AFG3L2. The cleavage event compromises the import of mitochondrial proteins mediated by DNAJC15. Moreover, this OMA1-mediated cleavage appears to regulate the transition of L-DNAJC15 from its role in mitochondrial protein import to that of S-DNAJC15 in CoQ biosynthesis and supporting OXPHOS activity. The data presented in the article are of high quality. However, while the study provides new insight into OMA1's role in DNAJC15 degradation, it is difficult to see the broader impact of this regulation on mitochondrial activity, particularly protein import, beyond our current understanding. Additionally, some of the claims made need additional experimental data for their substantiation.

Major Concerns:

1. Clarification of OMA1's role in import regulation of DNAJC15: The main premise of the study is that OMA1-mediated cleavage of DNAJC15 links mitochondrial stress to the regulation of mitochondrial protein import. This conclusion is primarily supported by Figure 4, which shows that knockdown of DNAJC15 reduces mitochondrial import efficiency. However, this data does not sufficiently demonstrate that the observed effect is specifically mediated by OMA1 cleavage of DNAJC15. To validate this claim, rescue experiments comparing overexpression of wild-type (WT) DNAJC15 and a non-cleavable DNAJC15 mutant (JC15 Δ 2) in a DNAJC15 knockdown background are required.
2. Discrepancy between mitochondrial import and proteome stability: The authors used different methods to assess the impact of DNAJC15 depletion, including knockdown and knockout approaches. In the knockdown condition, mitochondrial import efficiency decreases, whereas the overall mitochondrial proteome remains unchanged, presumably due to the shorter duration of knockdown. However, even in the knockout condition, the mitochondrial proteome remains unaffected (Fig. 5c). How do the authors explain why reduced mitochondrial import does not translate into changes in the mitochondrial proteome over time, even under knockout conditions? Additionally, the authors show that only upon TIMM17A depletion in DNAJC15 knockout cells significant changes occur in the mitochondrial proteome. This raises concerns about the significance of DNAJC15's role in mitochondrial import regulation under OMA1 activation during stress, as observed in Figure 4.
3. Insufficient evidence for the functional transition of DNAJC15 mediated by cleavage: The authors present changes in the interactome of WT DNAJC15 and the JC15 Δ 2 mutant (Fig. 6) and propose that OMA1 cleavage of DNAJC15 mediates a functional shift from protein import to CoQ biosynthesis, which is critical for OXPHOS activity. While this is a compelling hypothesis, the data presented in the study do not provide sufficient support for this claim. A direct comparison of the interactomes of L-DNAJC15 and S-DNAJC15 is necessary to substantiate this conclusion. The authors argue that S-DNAJC15 is unstable, preventing such an analysis. However, this limitation could be overcome by knocking out AFG3L2, as shown in Fig. 1b. Additionally, if S-DNAJC15 is rapidly degraded, does it make sense to assume that it plays a functional role in sustaining OXPHOS activity during stress? Could it instead be a transient byproduct of DNAJC15 degradation rather than an active participant in stress adaptation?
4. Specificity of DNAJC15's role in protein import: The role of DNAJC15 in mitochondrial protein import as component of TIM23 complex is already established (PMID: 23263864). It remains unclear whether the import function of DNAJC15 highlighted in this study is unique to DNAJC15 or if it just reflects a broader effect on the TIM23 complex. To clarify this, the authors should examine how the depletion of other TIM23 components impacts mitochondrial import and compare these findings to the effects observed in DNAJC15 knockdown conditions.

Reviewer #3

(Remarks to the Author)

Kroczek et al have discovered a novel mechanism that regulates mitochondrial biogenesis via processing and degradation of DNAJC15, a J-protein involved in protein import via the TIM23 translocase. The authors find that JC15 displays negative genetic interactions with TIMM17A, a stress-responsive subunit of the TIM23 presequence translocase. The study further demonstrates that deletion of JC15 leads to specific depletion of proteins of the mitochondrial ribosome as well as OXPHOS, and that non-imported proteins accumulate at the ER where they induce the UPR.

The findings are highly interesting and shed light on the regulation of mitochondrial biogenesis via stress-responsive translocase components. They rely on robust, high quality data from a variety of elegant, appropriate methods. The study warrants publication but a few conceptual aspects require further analysis:

1. OMA1 is a stress-regulated protease, however no particular stress seems to be required for DNAJC15 processing. Rather it appears that DNAJC15 is continually turned over by this mechanism, resulting in a short half-life. If this is indeed not a stress-induced pathway, the authors need to re-work their interpretations and conclusions accordingly. To clarify the physiological context of this mechanism, addressing some of the following points would be helpful: Is DNAJC15 degraded quantitatively upon (full) OMA1 activation, with similar consequences on mitochondrial biogenesis as JC15 deletion? Upon termination of OMA1 activity, are DNAJC15 levels perhaps stabilized relative to non-stressed conditions, resulting in a boost for mitochondrial biogenesis? Do JC15 Δ 2 expressing cells perform better in OCR measurements than the wildtype?
2. The authors find that several CoQ synthesis enzymes are uniquely dependent, for their interaction with DNAJC15, on the ability of DNAJC15 to be processed by OMA1, and conclude that these proteins preferentially interact with the processed form. Are there any functional consequences for CoQ synthesis in light of their finding that processed DNAJC15 is rapidly degraded by the m-AAA protease? Are CoQ levels affected in S-JC15 cells or upon OMA1 activation relative to wildtype and JC15 Δ 2 cells? What do the authors think is the purpose of this interaction in the context of (continuous or stress-induced) JC15 turnover?
3. The authors demonstrate convincingly that deletion of JC15 specifically interacts with TIMM17A knockdown in a synthetic negative manner. They interpret this to mean that JC15 functions as the J-protein for TIM23 translocase complexes that include TIMM17A. However, a synthetic negative interaction is usually indicative of the components being in parallel pathways rather than in the same pathway. How can the authors exclude that the negative genetic interaction is due to JC15 acting preferentially with TIMM17B (not A)-containing complexes, so depleting JC15 and TIMM17A functionally disrupts both types of TIM23 translocase complexes? In this context it is also relevant that JC19 levels are significantly decreased in JC15 KO + siTIMM17A cells (Fig. 5c), which makes the interpretation even less straightforward. In summary: Is it really justified to conclude that JC15 cooperates with TIMM17A? If not, the conclusions should be toned down accordingly.
4. Some controls are missing that are required to judge the efficacy of RNAi and the resulting status of the TIM23 translocase: Levels of TIMM17A, TIMM17B, DNAJC19 +/- knockdown of these components in WT or JC15-/- background; import rates for a standard matrix-targeted precursor under these conditions, or alternatively levels of assembled TIM23 complexes under the different conditions (if possible distinguishing between complexes containing 17A vs. B).
5. The Venn diagram in Fig. 4g should be extended to account for the mitochondrial proteins that are mislocalized to the ER in JC15-/- cells. Do proteins with a TIMM17A import dependence also display a bias towards short half-lives?

Decision Letter:

28th Mar 2025

Dear Thomas,

Thank you again for submitting your manuscript "Stress adaptation of mitochondrial protein import by OMA1-mediated degradation of DNAJC15" to NSMB. We now have comments (below) from the 3 reviewers who evaluated your paper. In light of those reports, we remain interested in your study and would like to see your response to the comments of the referees in the form of a revised manuscript.

You will see that the reviewers found the study interesting but also indicated that further evidence would be required to support the model. We have discussed the reviewers' feedback; we find that the concerns from the reviewers are significant and would need to be addressed thoroughly experimentally for the core conclusions and advance to stand. Reconsideration of the study for this journal and re-engagement of referees will depend on strength of these revisions. We are committed to providing a fair and constructive peer-review process and our standard revision is about 6 months; we are happy to discuss if you have any questions or anticipate any issues or delays. If you cannot send the revision back within this time, please let us know. We will be happy to consider your revision as long as nothing similar has been accepted for publication at NSMB or published elsewhere. Should your manuscript be substantially delayed without notifying us in advance and your article is eventually published, the received date would be that of the revised, not the original, version.

To guide the scope of the revisions, we list below a prioritized set of referee points that should be addressed in the revision, which we hope will be helpful to you. Please be sure to address/respond to all concerns of the referees in full in a point-by-point response and highlight all changes in the revised manuscript text file.

1—The reviewers were not convinced yet of the evidence linking DNAJC15 processing to mitochondrial protein import and to the TIMM23-TIMM17A translocase-dependent pathway. This aspect is central to the conclusions and needs to be strengthened according to the reviewers' points:

Rev#1 points #1, 2

Rev#2 points #1, 2, 3, 4
Rev#3 points #3, 4

2-Similarly, the functional consequences on mitochondria functions and proteome as well as the interplay with stress signaling should be clarified and bolstered:

Rev#1 point #3
Rev#3 point #1, 2, 5

3-Please also address the reviewers' minor points, requests for discussion and text edits/clarifications.

Reporting Summary:
<https://www.nature.com/documents/nr-reporting-summary.pdf>

When submitting the revised version of your manuscript, please pay close attention to our [href="https://www.nature.com/nature-portfolio/editorial-policies/image-integrity">Digital Image Integrity Guidelines. and to the following points below:](https://www.nature.com/nature-portfolio/editorial-policies/image-integrity)

EXTENDED DATA FIGURES

Please note that all key data shown in the main figures as cropped gels or blots should be presented in uncropped form, with molecular weight markers. These data can be aggregated into a single supplementary figure. While these data can be displayed in a relatively informal style, they must refer back to the relevant figures. These data should be submitted with the last revision, prior to acceptance, but you may want to start putting it together at this point.

We require deposition of coordinates (and, in the case of crystal structures, structure factors) into the Protein Data Bank with the designation of immediate release upon publication (HPUB). Electron microscopy-derived density maps and coordinate data must be deposited in EMDB and released upon publication. Deposition and immediate release of NMR chemical shift assignments are highly encouraged. Deposition of deep sequencing and microarray data is mandatory, and the datasets must be released prior to or upon publication. To avoid delays in publication, dataset accession numbers must be supplied with the final accepted manuscript and appropriate release dates must be indicated at the galley proof stage. Please find the complete NRG policies on data availability at <http://www.nature.com/authors/policies/availability.html>.

Nature Structural & Molecular Biology is committed to improving transparency in authorship. As part of our efforts in this direction, we are now requesting that all authors identified as 'corresponding author' on published papers create and link their Open Researcher and Contributor Identifier (ORCID) with their account on the Manuscript Tracking System (MTS), prior to acceptance. This applies to primary research papers only. ORCID helps the scientific community achieve unambiguous attribution of all scholarly contributions. You can create and link your ORCID from the home page of the MTS by clicking on 'Modify my Springer Nature account'. For more information please visit please visit <a

<http://www.springernature.com/orcid>>www.springernature.com/orcid.

Link Redacted

Sincerely,

Melina Casadio, PhD
Locum Chief Editor, Nature Structural & Molecular Biology
ORCID ID: <https://orcid.org/0000-0003-2389-2243>

Referee expertise:

Referee #1: mitochondria, protein import

Referee #2: mitochondria, proteostasis, proteomics

Referee #3: mitochondria, metabolism, stress responses

Reviewers' Comments:

Reviewer #1 (Remarks to the Author):

The study by Krocze et al. (2025) presents a detailed omics-enriched investigation into the role of DNAJC15 in the mitochondrial protein biogenesis. The authors identify DNAJC15 as a substrate of OMA1, which, upon cleavage, undergoes degradation by the m-AAA protease. The authors conclude that the loss of DNAJC15 alters mitochondrial protein import via the TIM23 complex, containing TIMM17A, affecting the respiratory chain biogenesis, leading to the subsequent accumulation of preproteins at the endoplasmic reticulum (ER) and triggering an ATF6-mediated stress response. The study provides the strong biochemical evidence regarding DNAJC15 processing and degradation, and also examines the homolog JC19. However, the authors introduce multiple hypotheses about its role in OXPHOS biogenesis, altered protein import, and stress responses but the appropriate evidence is missing. This broad scope results in a somewhat fragmented narrative. The study should be strengthened by biochemically linking DNAJC15 function to mitochondrial import and its downstream effects, such as the suggested changes in OXPHOS and in cellular stress pathways.

Major points:

1. Adaptation of DNAJC15 Knock-Out Cells

Mutant DNAJC15^{-/-} and depletion behave opposite in different aspects, raising the question of whether the proposed effect on the protein import via the TIMM17A containing TIM23 complex is due to the adaptation in the knock-out cells. To address this point, authors should compare knock-out and knock-down models side by side and show which phenotype can be reproduced in a double knock-down mutant of TIMM17A and JC15. The adaptations and different phenotypes observed for KOs and depletions are relatively commonly seen, however this should be addressed along specificity of the observed effects.

2. Protein import via DNAJC15

Authors conclude "from these experiments that DNAJC15 mediates the import of many matrix and IM proteins". This part requires more evidence such as:

-using in vitro imports to show changes in import of matrix and IM proteins and their preferential dependence on TIM17A and DNAJC15 versus TIMM17B.

- using appropriate controls of a membrane potential and the TIMM22 to show that the overall quality of mitochondria is not impaired.

Another point that requires additional evidence is differentiation between protein turnover and protein import. A protein turnover rate in DNAJC15 mutants should be examined.

3. Precursor accumulation and mislocalization to the ER and UPR activation

The study suggests that mitochondrial import defects caused by DNAJC15 loss lead to precursor accumulation in the cytosol and subsequent mislocalization to the ER, triggering an ATF6-mediated UPR. However, the direct evidence for these events is lacking.

The authors are recommended to use some of the suggested ways for providing more evidence on this interesting concept: show that mitochondrial proteins (substrates of DNAJC15) accumulate not in mitochondria by fractionation and/or proximity labeling approaches. Is this potential localization on the ER surface caused by a closer tethering of the ER and mitochondria?

The authors should also confirm the UPR activation with additional evidence beyond transcriptomics.

Minor points and recommendations:

- Show that L-JC15 interacts with TIM23 not S-LC15, using IPs.
- Can JC19 take over for JC15? Does the overexpression of JC19 rescue JC15^{-/-}?
- Check if knock-down of components of complex I and coenzyme Q synthesis would have the same effect as TIMM17A knock-downs in JC15^{-/-}
- Label JC19 in Vulcano blot of JC15^{-/-}, show if there is a general shift in mitochondrial proteins by labeling all mitochondrial proteins in volcano blot, since authors say it plays a role in mitochondrial import
- Comment why the S-JC15 can rescue siTIMM17a in JC15^{-/-}, but rescue is only partial in the KO mutant – could the already mentioned adaptations be the reason for it?
- Mark proteins identified and mentioned in the text also in the interactome of DNAJC15 IP (Fig. 3A)
- The authors state that matrix and IMM proteins are most affected by DNAJC15 depletion, suggesting a specific impairment in TIM23-dependent import. However: 1/3 of IMS proteins are also reduced, which suggests that the defect is broader than just matrix/IMM proteins.
- The recent papers from the Chacinska and Youle groups are not cited. Given the focus of this work, these papers provide an additional introductory information on the topic of TIM23 protein import regulation.

Reviewer #2 (Remarks to the Author):

The article describes the OMA1-mediated degradation of DNAJC15, which regulates its functions in mitochondria protein import. The authors demonstrate that the activation of OMA1 often occurs during mitochondrial stress, inducing the cleavage of DNAJC15 (L-DNAJC15) into a shorter version, S-DNAJC15. This shorter version is subsequently degraded by the m-AAA protease AFG3L2. The cleavage event compromises the import of mitochondrial proteins mediated by DNAJC15. Moreover, this OMA1-mediated cleavage appears to regulate the transition of L-DNAJC15 from its role in mitochondrial protein import to that of S-DNAJC15 in CoQ biosynthesis and supporting OXPHOS activity. The data presented in the article are of high quality. However, while the study provides new insight into OMA1's role in DNAJC15 degradation, it is difficult to see the broader impact of this regulation on mitochondrial activity, particularly protein import, beyond our current understanding. Additionally, some of the claims made need additional experimental data for their substantiation.

Major Concerns:

1. Clarification of OMA1's role in import regulation of DNAJC15: The main premise of the study is that OMA1-mediated cleavage of DNAJC15 links mitochondrial stress to the regulation of mitochondrial protein import. This conclusion is primarily supported by Figure 4, which shows that knockdown of DNAJC15 reduces mitochondrial import efficiency. However, this data does not sufficiently demonstrate that the observed effect is specifically mediated by OMA1 cleavage of DNAJC15. To validate this claim, rescue experiments comparing overexpression of wild-type (WT) DNAJC15 and a non-cleavable DNAJC15 mutant (JC15 Δ 2) in a DNAJC15 knockdown background are required.
2. Discrepancy between mitochondrial import and proteome stability: The authors used different methods to assess the impact of DNAJC15 depletion, including knockdown and knockout approaches. In the knockdown condition, mitochondrial import efficiency decreases, whereas the overall mitochondrial proteome remains unchanged, presumably due to the shorter duration of knockdown. However, even in the knockout condition, the mitochondrial proteome remains unaffected (Fig. 5c). How do the authors explain why reduced mitochondrial import does not translate into changes in the mitochondrial proteome over time, even under knockout conditions? Additionally, the authors show that only upon TIMM17A depletion in DNAJC15 knockout cells significant changes occur in the mitochondrial proteome. This raises concerns about the significance of DNAJC15's role in mitochondrial import regulation under OMA1 activation during stress, as observed in Figure 4.
3. Insufficient evidence for the functional transition of DNAJC15 mediated by cleavage: The authors present changes in the interactome of WT DNAJC15 and the JC15 Δ 2 mutant (Fig. 6) and propose that OMA1 cleavage of DNAJC15 mediates a functional shift from protein import to CoQ biosynthesis, which is critical for OXPHOS activity. While this is a compelling hypothesis, the data presented in the study do not provide sufficient support for this claim. A direct comparison of the interactomes of L-DNAJC15 and S-DNAJC15 is necessary to substantiate this conclusion. The authors argue that S-DNAJC15 is unstable, preventing such an analysis. However, this limitation could be overcome by knocking out AFG3L2, as shown in Fig. 1b. Additionally, if S-DNAJC15 is rapidly degraded, does it make sense to assume that it plays a functional role in sustaining OXPHOS activity during stress? Could it instead be a transient byproduct of DNAJC15 degradation rather than an active participant in stress adaptation?
4. Specificity of DNAJC15's role in protein import: The role of DNAJC15 in mitochondrial protein import as component of TIM23 complex is already established (PMID: 23263864). It remains unclear whether the import function of DNAJC15 highlighted in this study is unique to DNAJC15 or if it just reflects a broader effect on the TIM23 complex. To clarify this, the authors should examine how the depletion of other TIM23 components impacts mitochondrial import and compare these findings to the effects observed in DNAJC15 knockdown conditions.

Reviewer #3 (Remarks to the Author):

Kroczek et al have discovered a novel mechanism that regulates mitochondrial biogenesis via processing and degradation of DNAJC15, a J-protein involved in protein import via the TIM23 translocase. The authors find that JC15 displays negative genetic interactions with TIMM17A, a stress-responsive subunit of the TIM23 presequence translocase. The study further demonstrates that deletion of JC15 leads to specific depletion of proteins of the mitochondrial ribosome as well as OXPHOS, and that non-imported proteins accumulate at the ER where they induce the UPR.

The findings are highly interesting and shed light on the regulation of mitochondrial biogenesis via stress-responsive translocase components. They rely on robust, high quality data from a variety of elegant, appropriate methods. The study warrants publication but a few conceptual aspects require further analysis:

1. OMA1 is a stress-regulated protease, however no particular stress seems to be required for DNAJC15 processing. Rather it appears that DNAJC15 is continually turned over by this mechanism, resulting in a short half-life. If this is indeed not a stress-induced pathway, the authors need to re-work their interpretations and conclusions accordingly. To clarify the physiological context of this mechanism, addressing some of the following points would be helpful: Is DNAJC15 degraded quantitatively upon (full) OMA1 activation, with similar consequences on mitochondrial biogenesis as JC15 deletion? Upon termination of OMA1 activity, are DNAJC15 levels perhaps stabilized relative to non-stressed conditions, resulting in a boost for mitochondrial biogenesis? Do JC15 Δ 2 expressing cells perform better in OCR measurements than the wildtype?

2. The authors find that several CoQ synthesis enzymes are uniquely dependent, for their interaction with DNAJC15, on the ability of DNAJC15 to be processed by OMA1, and conclude that these proteins preferentially interact with the processed form. Are there any functional consequences for CoQ synthesis in light of their finding that processed DNAJC15 is rapidly degraded by the m-AAA protease? Are CoQ levels affected in S-JC15 cells or upon OMA1 activation relative to wildtype and JC15 Δ 2 cells? What do the authors think is the purpose of this interaction in the context of (continuous or stress-induced) JC15 turnover?

3. The authors demonstrate convincingly that deletion of JC15 specifically interacts with TIMM17A knockdown in a synthetic negative manner. They interpret this to mean that JC15 functions as the J-protein for TIM23 translocase complexes that include TIMM17A. However, a synthetic negative interaction is usually indicative of the components being in parallel pathways rather than in the same pathway. How can the authors exclude that the negative genetic interaction is due to JC15 acting preferentially with TIMM17B (not A)-containing complexes, so depleting JC15 and TIMM17A functionally disrupts both types of TIM23 translocase complexes? In this context it is also relevant that JC19 levels are significantly decreased in JC15 KO + siTIMM17A cells (Fig. 5c), which makes the interpretation even less straightforward. In summary: Is it really justified to conclude that JC15 cooperates with TIMM17A? If not, the conclusions should be toned down accordingly.

4. Some controls are missing that are required to judge the efficacy of RNAi and the resulting status of the TIM23 translocase: Levels of TIMM17A, TIMM17B, DNAJC19 +/- knockdown of these components in WT or JC15 $^{-/-}$ background; import rates for a standard matrix-targeted precursor under these conditions, or alternatively levels of assembled TIM23 complexes under the different conditions (if possible distinguishing between complexes containing 17A vs. B).

5. The Venn diagram in Fig. 4g should be extended to account for the mitochondrial proteins that are mislocalized to the ER in JC15 $^{-/-}$ cells. Do proteins with a TIMM17A import dependence also display a bias towards short half-lives?

Version 1:

Reviewer comments:

Reviewer #1

(Remarks to the Author)

The authors addressed the concerns and comments more than adequately. This is an exciting study of the high interest.

Reviewer #2

(Remarks to the Author)

The authors have answered my comments sufficiently and have strengthened the manuscript significantly. I congratulate them on their interesting story and manuscript.

Reviewer #3

(Remarks to the Author)

The authors have adequately addressed my comments and improved the quality of the manuscript. However, the presentation of the new import results (log₂FC of signal at 0h) is unusual and appears contrived; the import signal after zero incubation time should be close to the range of the noise and so it is unclear why it was chosen as the base for all calculations. Also, presumably each curve was normalized to its own 0h signal so the relative amounts of proteins imported in the two strains are unknown. The authors should include actual import results or graphs calculated in a more standard manner.

With this small correction, the manuscript is ready for publication.

Decision Letter:

Our ref: NSMB-A50679A

11th Dec 2025

Dear Thomas,

Thank you for submitting your revised manuscript "Stress adaptation of mitochondrial protein import by OMA1-mediated degradation of DNAJC15" (NSMB-A50679A). It has now been seen by the original referees and their comments are below. The reviewers find that the paper has improved in revision, and therefore we'll be happy in principle to publish it in Nature Structural & Molecular Biology, pending minor revisions to satisfy the referees' final requests and to comply with our editorial and formatting guidelines.

Thank you again for your interest in Nature Structural & Molecular Biology. Please do not hesitate to contact me if you have any questions.

Sincerely,

Melina Casadio, PhD
Consulting Senior Editor, Nature Structural & Molecular Biology
ORCID ID: <https://orcid.org/0000-0003-2389-2243>

Reviewer #1 (Remarks to the Author):

The authors addressed the concerns and comments more than adequately. This is an exciting study of the high interest.

Reviewer #2 (Remarks to the Author):

The authors have answered my comments sufficiently and have strengthened the manuscript significantly. I congratulate them on their interesting story and manuscript.

Reviewer #3 (Remarks to the Author):

The authors have adequately addressed my comments and improved the quality of the manuscript. However, the presentation of the new import results (log₂FC of signal at 0h) is unusual and appears contrived; the import signal after zero incubation time should be close to the range of the noise and so it is unclear why it was chosen as the base for all calculations. Also, presumably each curve was normalized to its own 0h signal so the relative amounts of proteins imported in the two strains are unknown. The authors should include actual import results or graphs calculated in a more standard manner.

With this small correction, the manuscript is ready for publication.

Version 2:

Decision Letter:

22nd Jan 2026

Dear Dr. Langer,

We are now happy to accept your revised paper "Stress adaptation of mitochondrial protein import by OMA1-mediated degradation of DNAJC15" for publication as an Article in Nature Structural & Molecular Biology.

Over the next few weeks, your paper will be copyedited to ensure that it conforms to Nature Structural & Molecular Biology

style. Once your paper is typeset, you will receive an email with a link to choose the appropriate publishing options for your paper and our Author Services team will be in touch regarding any additional information that may be required.

Your paper will be published online soon after we receive proof corrections and will appear in print in the next available issue. You can find out your date of online publication by contacting the production team shortly after sending your proof corrections.

Authors may need to take specific actions to achieve compliance with funder and institutional open access mandates. If your research is supported by a funder that requires immediate open access (e.g. according to <https://www.springernature.com/gp/open-science/plan-s-compliance> Plan S principles or the <https://www.springernature.com/gp/open-science/us-federal-agency-compliance> NIH public access policy) then you should select the gold OA route, and we will direct you to the compliant route where possible. Because authors warrant under our subscription licensing terms that they haven't committed to licensing any version of their article under a licence inconsistent with the terms of our agreement – including the applicable embargo period – publication under the subscription model isn't suitable for authors whose funders require no embargo.

Sincerely,

Yue Feng, PhD

(she/her)

Associate Editor

Nature Structural & Molecular Biology

ORCID: 0009-0002-9697-1903

Click here if you would like to recommend Nature Structural & Molecular Biology to your librarian:

<http://www.nature.com/subscriptions/recommend.html#forms>

Reply to Reviewers' Comments

Reply to reviewer #1:

The study by Kroczeck et al. (2025) presents a detailed omics-enriched investigation into the role of DNAJC15 in the mitochondrial protein biogenesis. The authors identify DNAJC15 as a substrate of OMA1, which, upon cleavage, undergoes degradation by the m-AAA protease. The authors conclude that the loss of DNAJC15 alters mitochondrial protein import via the TIM23 complex, containing TIMM17A, affecting the respiratory chain biogenesis, leading to the subsequent accumulation of preproteins at the endoplasmic reticulum (ER) and triggering an ATF6-mediated stress response.

The study provides the strong biochemical evidence regarding DNAJC15 processing and degradation, and also examines the homolog JC19. However, the authors introduce multiple hypotheses about its role in OXPHOS biogenesis, altered protein import, and stress responses but the appropriate evidence is missing. This broad scope results in a somewhat fragmented narrative. The study should be strengthened by biochemically linking DNAJC15 function to mitochondrial import and its downstream effects, such as the suggested changes in OXPHOS and in cellular stress pathways.

The motivation of our study was to define how OMA1 orchestrates mitochondrial responses to stress. Our identification of DNAJC15 as a novel OMA1 substrate (besides OPA1 and DELE1) suggested regulation of mitochondrial protein import, which we therefore characterized in more detail. As suggested by the reviewer, we have now consolidated the role of DNAJC15 during protein import and in the coenzyme Q metabolism and streamlined the text.

Although not the focus, we have found in the course of our study that loss of DNAJC15 also induces an UPR in the ER. Although this exciting finding warrants further investigation, we decided to include the data in our manuscript as they highlight the broad consequences of mitochondrial import deficiencies and OMA1-mediated stress responses.

Major points:

1. Adaptation of DNAJC15 Knock-Out Cells

Mutant DNAJC15^{-/-} and depletion behave opposite in different aspects, raising the question of whether the proposed effect on the protein import via the TIMM17A containing TIM23 complex is due to the adaptation in the knock-out cells. To address this point, authors should compare knock-out and knock-down models side by side and show which phenotype can be reproduced in a double knock-down mutant of TIMM17A and JC15. The adaptations and different phenotypes observed for KOs and depletions are relatively commonly seen, however this should be addressed along specificity of the observed effects.

We have shown genetic interactions of DNAJC15 with DNAJC19 and TIMM17A, but not with TIMM17B using DNAJC15^{-/-} cells in the original version of the manuscript. We have now extended this analysis using DNAJC15 knockdown experiments (new

Extended Data Fig. 3). These experiments confirmed our findings that DNAJC15 interacts genetically with DNAJC19 and TIMM17A but not TIMM17B. A functional interaction of DNAJC15 with TIMM17A is further supported by our proteomic experiments (Fig. 4g). We would like to emphasize, however, that this genetic evidence does not distinguish whether DNAJC15 binds exclusively to TIMM17A or TIMM17B or interacts with both.

2. Protein import via DNAJC15

Authors conclude “from these experiments that DNAJC15 mediates the import of many matrix and IM proteins”. This part requires more evidence such as:

-using in vitro imports to show changes in import of matrix and IM proteins and their preferential dependence on TIM17A and DNAJC15 versus TIMM17B.

Our conclusion that DNAJC15 broadly affects mitochondrial protein import is based on proteome-wide SILAC experiments, which revealed reduced levels of 265 mitochondrial proteins (94 mitochondrial IM proteins and 138 mitochondrial matrix proteins) after acute depletion of DNAJC15 in mitochondrial fractions that were treated with proteinase K to degrade non-imported proteins (Fig. 4b). Importantly, these proteins accumulated at normal levels in whole cell extracts, demonstrating normal synthesis. We therefore conclude that DNAJC15 depletion limits the import of these proteins. This is further supported by the physical interaction of 73 of these proteins with DNAJC15, indicating a direct effect of DNAJC15 on their protein import. These findings are consistent with and significantly extend previous reports using yeast complementation studies and showing an import function of DNAJC15 for the artificial model substrate Su9-DHFR^{mut} (PMID:23263864).

To comply with the request of the reviewer, we have extended our analysis and performed protein import experiments into isolated mitochondria *in vitro*. Despite the expected variability in these knockdown experiments, they substantiated our proteome-wide SILAC experiments and showed reduced mitochondrial protein import for several mitochondrial proteins, such as NDUFA9, TIMMDC1, COQ10B and COX16 upon acute knockdown of DNAJC15 (new Fig. 4d, e; new Extended Data Fig. 4e, f).

Further *in vitro* import experiments revealed that depletion of TIMM17A but not TIMM17B reduced mitochondrial import into isolated mitochondria. However, concomitant depletion with DNAJC15 revealed variable results and no further effects on protein import efficiency, likely reflecting technical limitations of the experimental

setup (mitochondrial isolation after siRNA-mediated depletion of two proteins). Since our manuscript focuses on the regulation of mitochondrial import upon stress-induced degradation of DNAJC15, we did not include these data in the manuscript. Dissecting the stoichiometry and precise composition of the functional import machinery is likely more complex and extends beyond the scope of the present study.

- using appropriate controls of a membrane potential and the TIMM22 to show that the overall quality of mitochondria is not impaired.

We have determined the mitochondrial membrane potential using TMRM labeling but did not observe any impairment after DNAJC15 depletion, excluding indirect effects (new Extended Data Fig. 4a, b). This is consistent with the observation that only a subset of matrix and IM proteins are affected by DNAJC15 depletion. Moreover, the steady state levels of the components of TIMM22, as well as its substrates remained unaffected after DNAJC15 depletion (new Extended Data Fig. 4h-j).

Another point that requires additional evidence is differentiation between protein turnover and protein import. A protein turnover rate in DNAJC15 mutants should be examined.

We agree with the reviewer that changes of either protein import or protein turnover could explain differences in the accumulation of mitochondrial proteins after DNAJC15 depletion (a limitation which is also true for classical *in vitro* import experiments). We therefore monitored in emetine chase experiments the turnover of selected short-lived proteins after DNAJC15 depletion, including both proteins affected (COQ10B, STARD7, TIMM17A, TIMMDC1) and unaffected (CHCHD2, NDUFA9) by the loss of DNAJC15 (new Extended Data Fig. 4c, d). Depletion of DNAJC15 did not significantly affect the turnover of any of these proteins. These results further substantiate a role of DNAJC15 during protein import.

3. Precursor accumulation and mislocalization to the ER and UPR activation
The study suggests that mitochondrial import defects caused by DNAJC15 loss lead to precursor accumulation in the cytosol and subsequent mislocalization to the ER, triggering an ATF6-mediated UPR. However, the direct evidence for these events is lacking. The authors are recommended to use some of the suggested ways for providing more evidence on this interesting concept: show that mitochondrial proteins (substrates of DNAJC15) accumulate not in mitochondria by fractionation and/or

proximity labeling approaches. Is this potential localization on the ER surface caused by a closer tethering of the ER and mitochondria?

As suggested by the reviewer, we have performed cell fractionation experiments and detected mitochondrial proteins in the ER fraction of DNAJC15-depleted cells (Fig. 7a-c and Extended Data Fig. 4 of the original manuscript). These cell fractionation experiments combined with mass spectrometry demonstrate unambiguously that 130 mitochondrial proteins are detected in the ER fraction specifically after loss of DNAJC15 but not of DNAJC19 (Fig. 6c, d). We have identified 24 proteins in the ER fraction, whose mitochondrial import was reduced in DNAJC15-depleted cells (revised Extended Data Fig. 5b). Because many of the enriched proteins were mitochondrial membrane proteins, we extended our analysis by a carbonate extraction of the ER fraction. We observed that the mitochondrial preproteins accumulating at the ER were predominantly detected in the pellet fraction, indicating integral membrane association of the mitochondrial proteins with the ER (new Fig. 6e)

We would like to emphasize that only a fraction of any affected protein is expected to accumulate in the ER, since DNAJC15 degradation does not completely block import (in contrast to $\Delta\psi_m$ dissipation), as expected for a physiological relevant stress response.

Our microscopic analysis of DNAJC15 depleted cells did not reveal significant alterations in ER-mitochondria tethering. Moreover, depletion of proteins known to be involved in ER-mito tethering (such as MFN1, MFN2, VDAC1, RMDN3) did not affect UPR induction in DNAJC15 depleted cells (Reviewer Fig. 1). Since these data are negative and a more thorough analysis is required to exclude any effect of membrane tethering, we prefer not to show these data in the manuscript.

The authors should also confirm the UPR activation with additional evidence beyond transcriptomics

The analysis of transcript levels of UPR signature genes (PMID:30821953) indicates the induction of ATF6-dependent UPR genes (Fig. 6g). We have extended this analysis and performed an unbiased analysis of our RNAseq data, which confirmed UPR induction (new Fig. 6f). Consistently, we observe a moderate accumulation of ATF6 at the protein level (Reviewer Fig. 2a-c). However, DNAJC15 depletion did not induce significant ATF6 processing. Consistently, inhibitors of ATF6 processing did not affect UPR induction. Notably, while ATF6 strongly accumulates in a processed form

upon tunicamycin treatment of the cells as expected, treatment of the cells with another known UPR inducer, thapsigargin, did not cause ATF6 processing (Reviewer Fig. 2a-f). Therefore, a more complex regulation of this pathway appears to exist, which warrants further experimentation but is outside the scope of this manuscript. We have excluded a role of various ER-mitochondria tethering factors (MFN1, MFN2, VDAC, RMND3) (Reviewer Fig. 1b-e) or other proteins that have been previously linked to mitochondrial stress signaling (XBP1, ATAD1, HSF1, DNAJA1, DNAJB6) (Reviewer Fig. 3a-e). Since our interpretation of an ATF6-dependent UPR relies only on a previously identified signature gene set (PMID:30821953) and since we cannot demonstrate unambiguously ATF6 dependence, we use more careful wording throughout the manuscript and emphasize only UPR induction as part of the stress response to tailor mitochondrial biogenesis.

Minor points and recommendations:

- *Show that L-JC15 interacts with TIM23 not S-LC15, using IPs.*

The N-terminal region of DNAJC15, which is exposed to the IMS and partially cleaved off by OMA1, has been reported to promote binding of DNAJC15 to the C-terminal domain of TIMM17A within the TIM23 complex (PMID:24636990). However, we were unable to confirm this observation using glycerol gradient fractionation of mitochondrial extracts, which did not show co-migration of DNAJC15 or DNAJC19 with TIM23 complexes (Reviewer Fig. 4a, b). Immunoprecipitation experiments combined with chemical crosslinking identified TIMM23 and other components of the TIM23 complex but not TIMM17A or TIMM17B (Fig. 3d). These experiments are hampered by the short half-life of S-DNAJC15 and TIMM17A. We therefore also performed interaction studies in mitochondria lacking the m-AAA protease AFG3L2, stabilizing S-DNAJC15 (new Fig. 5g). Although these experiments substantiated the link of DNAJC15 to coenzyme Q metabolism, we did not detect specific crosslinks of L- or S-DNAJC15 to TIMM17A or TIMM17B. We speculate that the pleiotropic functions of AFG3L2 precluded further insight into the assembly of TIM23 complexes in the IM. It therefore remains open whether processing of DNAJC15 by OMA1 is sufficient to destabilize the interaction with TIM23 complexes or whether the import defect depends on the complete degradation of DNAJC15 initiated by OMA1 cleavage. We would like to emphasize,

however, that this does not limit the main conclusion of our manuscript that the degradation of DNAJC15 modulates mitochondrial protein import.

- *Can JC19 take over for JC15? Does the overexpression of JC19 rescue JC15^{-/-}?*

DNAJC15^{-/-} cells do not show any obvious phenotype. However, our genetic analysis shows that DNAJC19 is essential in these cells (Fig. 2a, f), demonstrating that DNAJC19 can replace DNAJC15. This is consistent with previous observations in cells expressing high levels of Magma (PMID:34715125).

- *Check if knock-down of components of complex I and coenzyme Q synthesis would have the same effect as TIMM17A knock-downs in JC15^{-/-}*

We have depleted the complex I subunits NDUFS1 and NDUFS4 and the CoQ biosynthetic enzyme COQ2 in DNAJC15^{-/-} cells and monitored cell growth, but did not observe any genetic interaction with DNAJC15 (Reviewer Fig. 4c-e). However, we prefer not to include these negative data in the manuscript.

- *Label JC19 in Volcano blot of JC15^{-/-}, show if there is a general shift in mitochondrial proteins by labeling all mitochondrial proteins in volcano blot, since authors say it plays a role in mitochondrial import*

Similar to other mitochondrial proteins, the steady state level of DNAJC19 is not affected in DNAJC15^{-/-} cells as now shown in the Volcano blot in Extended Data Fig. 2c. We also include a boxplot analysis of the mitochondrial proteome in the revised manuscript, which shows no decrease in mitochondrial proteins in the DNAJC15^{-/-} cells (new Extended Fig. 6a).

- *Comment why the S-JC15 can rescue siTIMM17a in JC15^{-/-}, but rescue is only partial in the KO mutant – could the already mentioned adaptations be the reason for it?*

The expression of S-DNAJC15 only partially restores cell growth of DNAJC15^{-/-} cell depleted of DNAJC19 (Fig. 2d) but more efficiently the cells growth of DNAJC15^{-/-} cells depleted of TIMM17A (Extended Data Fig. 2f). We speculate that the apparent adaptations in DNAJC15^{-/-} cells are related to DNAJC19, which has overlapping functions to DNAJC15 (Fig. 2a).

- *Mark proteins identified and mentioned in the text also in the interactome of DNAJC15 IP (Fig. 3A)*

We have marked the identified and mentioned proteins in Fig. 3a.

• *The authors state that matrix and IMM proteins are most affected by DNAJC15 depletion, suggesting a specific impairment in TIM23-dependent import. However: 1/3 of IMS proteins are also reduced, which suggests that the defect is broader than just matrix/IMM proteins.*

We agree with the notion of the reviewer and use more careful wording. It is conceivable that some proteins are affected indirectly via other proteins that are directly affected by DNAJC15 depletion.

• *The recent papers from the Chacinska and Youle groups are not cited. Given the focus of this work, these papers provide an additional introductory information on the topic of TIM23 protein import regulation.*

We apologize for this oversight and refer now to the recent papers of the Chacinska and Youle groups.

Reply to Reviewer #2:

The article describes the OMA1-mediated degradation of DNAJC15, which regulates its functions in mitochondria protein import. The authors demonstrate that the activation of OMA1 often occurs during mitochondrial stress, inducing the cleavage of DNAJC15 (L-DNAJC15) into a shorter version, S-DNAJC15. This shorter version is subsequently degraded by the m-AAA protease AFG3L2. The cleavage event compromises the import of mitochondrial proteins mediated by DNAJC15. Moreover, this OMA1-mediated cleavage appears to regulate the transition of L-DNAJC15 from its role in mitochondrial protein import to that of S-DNAJC15 in CoQ biosynthesis and supporting OXPHOS activity.

The data presented in the article are of high quality. However, while the study provides new insight into OMA1's role in DNAJC15 degradation, it is difficult to see the broader impact of this regulation on mitochondrial activity, particularly protein import, beyond our current understanding. Additionally, some of the claims made need additional experimental data for their substantiation.

OMA1 regulates mitochondrial dynamics (via OPA1 processing) and stress signaling (via DELE1 processing) and therefore controls adaptive responses to stress. The identification of DNAJC15 as a novel proteolytic substrate extends the function of OMA1 to the regulation of mitochondrial protein import. Moreover, our analysis demonstrates preferred effects of OMA1-mediated DNAJC15 degradation on OXPHOS biogenesis, demonstrating that the degradation of selected subunits of the TIM23 complex under stress tailors the mitochondrial proteome. We propose that limiting the general import capacity by degrading components of protein translocases provides means to reshape the mitochondrial proteome according to protein half-lives.

We are convinced that these findings significantly extend our current understanding of mitochondrial stress adaptation and introduce a novel concept for mitochondrial proteome regulation.

Major Concerns:

1. Clarification of OMA1's role in import regulation of DNAJC15: The main premise of the study is that OMA1-mediated cleavage of DNAJC15 links mitochondrial stress to the regulation of mitochondrial protein import. This conclusion is primarily supported by Figure 4, which shows that knockdown of DNAJC15 reduces mitochondrial import efficiency. However, this data does not sufficiently demonstrate that the observed effect is specifically mediated by OMA1 cleavage of DNAJC15. To validate this claim, rescue experiments comparing overexpression of wild-type (WT) DNAJC15 and a non-cleavable DNAJC15 mutant (JC15 Δ 2) in a DNAJC15 knockdown background are required.

Our results demonstrate that OMA1 cleavage destabilizes DNAJC15 and facilitates its degradation by AFG3L2. We therefore have used DNAJC15 depletion experiments to define consequences of DNAJC15 degradation. We present several lines of evidence supporting the regulation of mitochondrial protein import by DNAJC15 degradation: 1. Synthetic growth defects caused by that loss of DNAJC15 and other components of the TIM23 complex that can be suppressed by expression of the stable DNAJC15 Δ 2 variant (Fig. 2); 2. Reduced accumulation of 265 mitochondrial proteins in mitochondria which are synthesized at normal rates at cytosolic ribosomes (Fig. 4). 3. Specific crosslinking of numerous OXPHOS-related proteins to DNAJC15 likely during their protein translocation (Fig. 3). 4. To further substantiate the role of DNAJC15 in protein import, we now performed mitochondrial protein import experiments in vitro (new Fig. 4d, e; new Extended Data Fig. 4e, f). Together, these findings demonstrate unambiguously that DNAJC15 functions in mitochondrial protein import and that its loss upon proteolytic degradation impairs the accumulation of OXPHOS-related proteins.

To further elucidate functional differences between L- and S-DNAJC15, we've established cell lines expressing siRNA-resistant DNAJC15 Δ 2 or S-DNAJC15 and depleted the cells of endogenous DNAJC15. SILAC experiments (similar to Fig. 4) should provide insight how OMA1-mediated cleavage of L- to S-DNAJC15 affects protein import. However, the rapid turnover of DNAJC15 hampered these experiments and precluded SILAC labeling which requires prolonged cell cultivation. While we succeeded to monitor Coenzyme Q levels in these cells (new Fig. 5h, i), requiring only limited cell cultivation, we were not able to preserve L Δ 2- and S-DNAJC15 levels at constant levels for a sufficient time to allow SILAC labeling. It therefore remains open

whether processing of DNAJC15 by OMA1 is sufficient to destabilize the interaction with TIM23 complexes or whether the import defect depends on the complete degradation of DNAJC15 initiated by OMA1 cleavage. We would like to emphasize, however, that this does not limit the main conclusion of our manuscript that OMA1 activation and degradation of DNAJC15 modulates mitochondrial protein import.

2. Discrepancy between mitochondrial import and proteome stability: The authors used different methods to assess the impact of DNAJC15 depletion, including knockdown and knockout approaches. In the knockdown condition, mitochondrial import efficiency decreases, whereas the overall mitochondrial proteome remains unchanged, presumably due to the shorter duration of knockdown. However, even in the knockout condition, the mitochondrial proteome remains unaffected (Fig. 5c). How do the authors explain why reduced mitochondrial import does not translate into changes in the mitochondrial proteome over time, even under knockout conditions? Additionally, the authors show that only upon TIMM17A depletion in DNAJC15 knockout cells significant changes occur in the mitochondrial proteome. This raises concerns about the significance of DNAJC15's role in mitochondrial import regulation under OMA1 activation during stress, as observed in Figure 4.

We apologize for making this point not sufficiently clear. The acute depletion of DNAJC15 does not affect the accumulation of mitochondrial proteins when whole cells are analyzed by mass spectrometry, demonstrating that mitochondrial proteins are synthesized at normal rates. However, upon isolation of mitochondria from these cells, we observed changes in the mitochondrial proteome: 265 mitochondrial proteins were present at decreased levels in a protease-protected form in the mitochondrial fraction of DNAJC15-depleted cells. We have identified 71 of these proteins to be crosslinked to DNAJC15, suggesting a direct interaction (new Extended Data Fig. 5b). These experiments demonstrate that not all newly synthesized mitochondrial proteins are transported to mitochondria and established a function of DNAJC15 in mitochondrial protein import. Non-imported mitochondrial proteins are either degraded in the cytosol or accumulate at the ER.

In contrast, the steady state levels of mitochondrial proteins in *DNAJC15^{-/-}* cells were not altered, neither in whole cell extracts nor mitochondrial fractions. This suggests adaptive processes that are relatively commonly seen, as also pointed out by reviewer 1. However, it should be noted that *DNAJC15^{-/-}* cells are more susceptible than wildtype cells to depletion of TIMM17A (similar to DNAJC15 siRNA-depleted cells), indicating an increased sensitivity of *DNAJC15^{-/-}* cells to specific impairments in mitochondrial biogenesis.

3. Insufficient evidence for the functional transition of DNAJC15 mediated by cleavage: The authors present changes in the interactome of WT DNAJC15 and the JC15Δ2 mutant (Fig. 6) and propose that OMA1 cleavage of DNAJC15 mediates a functional shift from protein import to CoQ biosynthesis, which is critical for OXPHOS activity. While this is a compelling hypothesis, the data presented in the study do not provide sufficient support for this claim. A direct comparison of the interactomes of L-DNAJC15 and S-DNAJC15 is necessary to substantiate this conclusion. The authors argue that S-DNAJC15 is unstable, preventing such an analysis. However, this limitation could be overcome by knocking out AFG3L2, as shown in Fig. 1b. Additionally, if S-DNAJC15 is rapidly degraded, does it make sense to assume that it plays a functional role in sustaining OXPHOS activity during stress? Could it instead be a transient byproduct of DNAJC15 degradation rather than an active participant in stress adaptation?

We agree with the reviewer that evidence for functional differences between L- and S-DNAJC15 was limited and restricted to changes in the interactome between WT DNAJC15 and DNAJC15Δ2. Despite repeated efforts, we failed to establish cell lines that stably and exclusively express S-DNAJC15, mainly due to its inherent instability. Moreover, expression of S-DNAJC15 ceased quickly after repeated cell division, pointing to deleterious effects of S-DNAJC15 accumulation (rationalizing its short half-life). Consistently, overexpression of DNAJC15 has been reported to induce MPTP opening and cell death (PMID:24603329).

To comply with the request of the reviewer, we have analyzed the interactome of DNAJC15 in *AFG3L2*^{-/-} cells, which accumulate mainly S-DNAJC15 (Fig. 1B), using immunoprecipitation combined with crosslinking. Strikingly, we identified several subunits of the Coenzyme Q biosynthetic machinery, COQ4, COQ8A, and COQ10B exclusively in cells lacking AFG3L2 and accumulating S-DNAJC15 (new Fig. 5g). These experiments strongly suggest that S-DNAJC15 can interact with Coenzyme Q biosynthetic enzymes and therefore complement our previous findings showing that DNAJC15 but not non-cleavable DNAJC15Δ2 interacts with CoQ biosynthetic enzymes (Fig. 5f). It should be noted that COQ8A, in contrast to COQ4 and COQ10B, is not a substrate of AFG3L2 and does not accumulate in *AFG3L2*^{-/-} cells, excluding confounding effects of the loss of the protease. Regardless, the loss of AFG3L2 broadly affects mitochondrial gene expression and OXPHOS assembly (PMID:23041622; PMID:38012514) and Coenzyme Q synthesis is impaired in OXPHOS deficiency; (PMID:29132502; PMID:27374853; PMID:25126048), making an interpretation of these findings more difficult.

We next analyzed cellular CoQ9 and CoQ10 levels and found that expression of the non-cleavable (siRNA-resistant) DNAJC15 variant resulted in a notable increase in the

total CoQ content (new Fig. 5h, i). This finding supports a functional link between OMA1-mediated DNAJC15 cleavage and CoQ metabolism. Given that S-DNAJC15 is likely the isoform interacting with CoQ-associated proteins, S-DNAJC15 may inhibit CoQ biosynthesis. Accordingly, rapid turnover of S-DNAJC15 may allow regulation (please note that CoQ also serves other metabolic functions, including protection against lipid peroxidation, besides its role for OXPHOS activity). However, a direct validation of this hypothesis would require stabilization of S-DNAJC15 by AFG3L2 depletion, which however has confounding effects, since AFG3L2 depletion impairs OXPHOS activity (see above). Although further studies are required to establish how DNAJC15 affects CoQ synthesis, our results unambiguously link OMA1-dependent processing of DNAJC15 to the regulation of both protein import and CoQ metabolism.

4. Specificity of DNAJC15's role in protein import: The role of DNAJC15 in mitochondrial protein import as component of TIM23 complex is already established (PMID: 23263864). It remains unclear whether the import function of DNAJC15 highlighted in this study is unique to DNAJC15 or if it just reflects a broader effect on the TIM23 complex. To clarify this, the authors should examine how the depletion of other TIM23 components impacts mitochondrial import and compare these findings to the effects observed in DNAJC15 knockdown conditions.

Our results reveal the selective degradation of DNAJC15 by OMA1 and AFG3L2 as a novel stress-regulated pathway to reshape the mitochondrial proteome. As pointed out by the reviewer, the role of DNAJC15 in mitochondrial protein import has been previously reported based on yeast complementation studies and *in vitro* import experiments using the model substrate Su9-DHFR^{mut} (PMID:23263864). Our experiments significantly extend these findings showing broad effects of DNAJC15 proteolysis on the mitochondrial proteome, in particular on OXPHOS related processes, demonstrating specific reshaping of the mitochondrial proteome under stress.

Beyond these findings, our study revealed novel general insight into mitochondrial protein import, such as functional differences between DNAJC15/DNAJC19 and TIMM17A/B, which will be of interest to the field. However, given the dynamic nature of TIM23 complexes, a characterization how the loss of individual subunits impacts mitochondrial protein import would require a systematic analysis of TIM23 complexes under the various conditions (including the characterization of assembly states, homo- vs. heterodimerization of individual components, etc.), which is beyond the scope of the present manuscript. Regardless, would like to emphasize, that several

observations highlight the specificity of DNAJC15 function: DNAJC15 but not DNAJC19 depletion affects the mitochondrial proteome and OXPHOS related pathways (without completely blocking mitochondrial protein import). Similarly, TIMM17A but not TIMM17B depletion aggravated the effect of DNAJC15 loss.

Reply to reviewer #3:

Kroczek et al have discovered a novel mechanism that regulates mitochondrial biogenesis via processing and degradation of DNAJC15, a J-protein involved in protein import via the TIM23 translocase. The authors find that JC15 displays negative genetic interactions with TIMM17A, a stress-responsive subunit of the TIM23 presequence translocase. The study further demonstrates that deletion of JC15 leads to specific depletion of proteins of the mitochondrial ribosome as well as OXPHOS, and that non-imported proteins accumulate at the ER where they induce the UPR.

The findings are highly interesting and shed light on the regulation of mitochondrial biogenesis via stress-responsive translocase components. They rely on robust, high quality data from a variety of elegant, appropriate methods. The study warrants publication but a few conceptual aspects require further analysis:

1. OMA1 is a stress-regulated protease, however no particular stress seems to be required for DNAJC15 processing. Rather it appears that DNAJC15 is continually turned over by this mechanism, resulting in a short half-life. If this is indeed not a stress-induced pathway, the authors need to re-work their interpretations and conclusions accordingly. To clarify the physiological context of this mechanism, addressing some of the following points would be helpful: Is DNAJC15 degraded quantitatively upon (full) OMA1 activation, with similar consequences on mitochondrial biogenesis as JC15 deletion? Upon termination of OMA1 activity, are DNAJC15 levels perhaps stabilized relative to non-stressed conditions, resulting in a boost for mitochondrial biogenesis? Do JC15 Δ 2 expressing cells perform better in OCR measurements than the wildtype?

We thank the reviewer for their positive evaluation and thoughtful comments. Our data demonstrate that OMA1 cleavage facilitates complete proteolysis of DNAJC15. OMA1 activation accelerates the (already high) turnover of DNAJC15. This is illustrated, first, by the increased stability of DNAJC15 Δ 2 when compared to L-DNAJC15 (Fig. 1f, g) and, second, by the increased stability of L-DNAJC15 upon OMA1 depletion when compared to the proteolysis of L-DNAJC15 in cells depleted of AFG3L2 (Fig. 1b, c). Therefore, we conclude that stress conditions accelerate DNAJC15 turnover by OMA1 and AFG3L2. Please note that OMA1 is activated in cells depleted of AFG3L2 (PMID: 20038678), likely by the accumulation of misfolded proteins (PMID: 30683687). Similarly, emetine treatment causes OMA1 activation and accelerates L-DNAJC15 cleavage (Fig. 1d, e), which hampers the direct comparison of turnover rates under normal and mitochondrial stress conditions.

The reviewer raised the interesting possibility that this proteolytic pathway stimulated by mitochondrial stress (including OXPHOS deficiency) also facilitates the recovery from stress. As speculated by the reviewer, alleviation of stress conditions (and restoration of the membrane potential after removal of Oligomycin and Antimycin A) results in the rapid accumulation of DNAJC15, supporting OXPHOS related pathways (new Extended Data Fig. 1c, d). It is therefore conceivable that the termination of OMA1 activity promotes mitochondrial biogenesis by stabilizing DNAJC15. We thank the reviewer for this comment and discuss this possibility now on p. 21 of the manuscript.

We additionally assessed mitochondrial respiration by measuring the oxygen consumption rate (OCR) upon expression of DNAJC15 (new Fig. 5b), which fully rescued the respiratory phenotype despite only partial restoration of DNAJC15 protein levels. In contrast, expression of the DNAJC15 Δ 2 variant failed to fully rescue the OCR phenotype (new Fig. 5c), even though its expression levels were comparable to those of the wildtype construct and CoQ content remained elevated (new Fig. 5h, i). Thus, DNAJC15 Δ 2 does not enhance OXPHOS activity, which appears to depend on the balanced accumulation of L- and S-DNAJC15. This reminiscent of the role of OPA1 for mitochondrial dynamics. Only the balanced accumulation of L- and S-OPA1 (generated by OMA1) preserves normal mitochondrial morphology.

2. The authors find that several CoQ synthesis enzymes are uniquely dependent, for their interaction with DNAJC15, on the ability of DNAJC15 to be processed by OMA1, and conclude that these proteins preferentially interact with the processed form. Are there any functional consequences for CoQ synthesis in light of their finding that processed DNAJC15 is rapidly degraded by the m-AAA protease? Are CoQ levels affected in S-JC15 cells or upon OMA1 activation relative to wildtype and JC15 Δ 2 cells? What do the authors think is the purpose of this interaction in the context of (continuous or stress-induced) JC15 turnover?

Despite multiple attempts, we were unable to establish cell lines that stably and exclusively express S-DNAJC15, likely due to the intrinsic instability of this isoform, precluding a direct analysis of S-DNAJC15 function for CoQ metabolism. Expression of S-DNAJC15 declined rapidly over successive cell divisions, indicating that its accumulation is detrimental to cell viability (Reviewer Fig. 4f). To further examine how impaired DNAJC15 processing affects CoQ metabolism, we compared CoQ levels in DNAJC15-depleted cells expressing either siRNA-resistant wildtype DNAJC15 or the non-cleavable DNAJC15 Δ 2 variant. Expression of DNAJC15 Δ 2, but not of DNAJC15,

resulted in a marked increase in total CoQ9 and CoQ10 levels (new Fig. 5h, i). Given that S-DNAJC15 is likely the isoform interacting with proteins of the CoQ metabolism, S-DNAJC15 may inhibit CoQ biosynthesis, while its rapid turnover may allow regulation (please note that CoQ also serves other metabolic functions, including protection against lipid peroxidation, besides its role for OXPHOS activity). These results suggest that OMA1 activation under stress limits import of OXPHOS-related proteins and reduces OXPHOS activity and CoQ accumulation. Collectively, these findings uncover a previously unrecognized role of DNAJC15 in directly modulating CoQ accumulation, independent of secondary consequences arising from OXPHOS impairment.

3. The authors demonstrate convincingly that deletion of JC15 specifically interacts with TIMM17A knockdown in a synthetic negative manner. They interpret this to mean that JC15 functions as the J-protein for TIM23 translocase complexes that include TIMM17A. However, a synthetic negative interaction is usually indicative of the components being in parallel pathways rather than in the same pathway. How can the authors exclude that the negative genetic interaction is due to JC15 acting preferentially with TIMM17B (not A)-containing complexes, so depleting JC15 and TIMM17A functionally disrupts both types of TIM23 translocase complexes? In this context it is also relevant that JC19 levels are significantly decreased in JC15 KO + siTIMM17A cells (Fig. 5c), which makes the interpretation even less straightforward. In summary: Is it really justified to conclude that JC15 cooperates with TIMM17A? If not, the conclusions should be toned down accordingly.

We agree with the reviewer that a negative genetic interaction (as seen between DNAJC15 and TIMM17A) is indicative of parallel pathways and difficult to reconcile with a complex formation of both proteins. We were led in our interpretation by previous studies proposing an interaction of DNAJC15 via its N-terminal region, which is cleaved off at least partially by OMA1, specifically with TIMM17A (PMID: 24636990; PMID:34715125). Our crosslink immunoprecipitation did not identify TIMM17A or TIMM17B as interaction partners of DNAJC15 (Fig. 3d). Moreover, glycerol gradient centrifugation failed to resolve complex formation between TIMM17A, TIMM17B, DNAJC15, and DNAJC19 (Reviewer Fig. 4a, b). We would also like to point out that binding of DNAJC15 and DNAJC19 homo- or heterodimers appears conceivable, adding additional complexity to the interpretation of genetic interactions. Since we have not analyzed the (likely dynamic) physical interaction of DNAJC15 with different TIM23 complexes in the manuscript, we avoided to directly comment on the assembly but used “cooperate” when referring to the negative genetic interaction (Fig. 2f) and combined effects on the mitochondrial proteome (Fig. 4). We realized that this is

misleading and, to avoid misunderstandings, we use now more careful wording when referring to the functional interaction of DNAJC15 with TIMM17A.

On a different note, we have noticed in the course of our experiments that commercially available antibodies recognize specifically the TIMM17B isoform 2 but not isoform 1 (PMID:24636990, Reviewer Fig. 4a, b), making the interpretation of the data even more complex. We feel that a detailed analysis of different assembly states of TIM23 complexes is required to unambiguously demonstrate the interaction of DNAJC15 with TIMM17A, TIMM17B or both.

4. Some controls are missing that are required to judge the efficacy of RNAi and the resulting status of the TIM23 translocase: Levels of TIMM17A, TIMM17B, DNAJC19 +/- knockdown of these components in WT or JC15-/- background; import rates for a standard matrix-targeted precursor under these conditions, or alternatively levels of assembled TIM23 complexes under the different conditions (if possible distinguishing between complexes containing 17A vs. B).

To comply with the request of the reviewer, we show now the steady state levels of various protein translocase subunits and the levels of DNAJC15 in WT, *DNAJC15^{-/-}*, *DNAJC19^{-/-}* combined with depletion of TIMM17A or B (new Extended Data Fig. 5c, d, Reviewer Fig. 1a). We also performed *in vitro* import assays using mitochondria depleted of DNAJC15, which substantiated our proteome-wide SILAC experiments and showed reduced mitochondrial protein import for several mitochondrial proteins, such as NDUFA9, TIMMDC1, COQ10B and COX16 upon acute knockdown of DNAJC15 (new Fig. 4d, e; new Extended Data Fig. 4e, f)

Further experiments revealed that depletion of TIMM17A but not TIMM17B reduced mitochondrial import into isolated mitochondria. However, concomitant depletion with DNAJC15 revealed variable results and no further effects on protein import efficiency, likely reflecting technical limitations of the experimental setup. Since our manuscript focuses on the regulation of mitochondrial import upon stress-induced degradation of DNAJC15, we did not include these data in the manuscript.

5. The Venn diagram in Fig. 4g should be extended to account for the mitochondrial proteins that are mislocalized to the ER in JC15-/- cells. Do proteins with a TIMM17A import dependence also display a bias towards short half-lives?

We have extended Fig. 4g as suggested by the reviewer (new Extended Data Fig. 5b). Proteins with a TIMM17A import dependence only display a bias towards short half-lives, if DNAJC15 is missing (new Fig. 4i, Reviewer Fig. 4g, h).

Reviewer Figure 1

Reviewer Figure 2

Reviewer Figure 3

Reviewer Figure 4

Reviewer Fig. 1 a, Protein levels in cells after siRNA-mediated depletion of TIMM17A monitored by Western Blot (n=3). **b-e**, Transcript levels of ATF6 target genes (\log_2 Fold change) of the unfolded protein response (UPR), normalized to wildtype (WT) HeLa cells following siRNA-mediated depletion of **b**, DNAJC15, MFN1, and combined treatment; **c**, DNAJC15, MFN2, and combined treatment; **d**, DNAJC15, RMDN3, and combined treatment; **e**, DNAJC15, VDAC1, and combined treatment. Data are means \pm SD (n=3, biologically independent samples).

Reviewer Fig. 2 a, Protein levels in cells after siRNA-mediated depletion of DNAJC15 and treatment for 16 h with thapsigargin (10 μ M) or tunicamycin (10 μ M), monitored by Western Blot. **b**, Quantification of ATF6 α protein levels in (a). Only the cleaved ATF6 α form was quantified after tunicamycin treatment. Data are means \pm SD (n=3, biologically independent samples). *P* values were calculated using a two-sided unpaired t-test. **c**, Quantification of ATF6 β protein levels in (a). Data are means \pm SD (n=3, biologically independent samples). *P* values were calculated using a two-sided unpaired t-test. **d-f**, Transcript levels of **d**, ATF6 (**d**), IRE1 (**e**) and PERK (**f**) target genes (\log_2 FC) of the unfolded protein response (UPR) following siRNA-mediated depletion of DNAJC15 and treatment with tunicamycin (10 μ M) or thapsigargin (10 μ M). Data are means \pm SD (n=3, biologically independent samples).

Reviewer Fig. 3 Transcript levels of ATF6 target genes (\log_2 FC) of the UPR, normalized to wildtype (WT) HeLa cells following siRNA-mediated depletion of **a**, DNAJC15, ATAD1, and combined treatment; **b**, DNAJC15, DNAJA1, and combined treatment; **c**, DNAJC15, DNAJB6, and combined treatment; **d**, DNAJC15, XBP1, and combined treatment; **e**, DNAJC15, HSF1, and combined treatment. Data are means \pm SD (n=3, biologically independent samples).

Reviewer Fig. 4 a, Analysis of TIM23 complex assembly. Mitochondrial lysates (WT, HeLa) were subjected to glycerol gradient centrifugation (20-50% glycerol; 160.000xg; 16 h) and fractions were analyzed by immunoblotting. Input and pellet represent 10% and 5% of the total gradient fraction, respectively. **b**, Heatmap showing the distribution of TIM23 complex subunits in (a) quantified by mass spectrometry. **c-e**, Cell growth of wild-type (WT) and *DNAJC15*^{-/-} HeLa cells after siRNA-mediated depletion of COQ2

(h), NDUFS4 (i), NDUFS1 (j). Data are means \pm SD. *P* values were calculated using a Mann-Whitney U-test. Quantile box plot shows median, 25th and 75th percentiles (n = 3, biologically independent samples) **f**, Expression levels of DNAJC15 protein in wild-type (WT) expressing DNAJC15 (JC15), S-DNAJC15 (S-JC15) and non-cleavable DNAJC15 (JC15 Δ 2) after siRNA-mediated depletion of DNAJC15 (siJC15) monitored by SDS-PAGE and immunoblotting. **g**, **h**, Boxplot visualizing the distribution of the half-life of mitochondrial proteins (according to MitoCop) and proteins significantly downregulated ($\text{Log}_2\text{FC} < -0.58$, $\text{FDR} < 0.05$) in (g) WT or in (h) *DNAJC15*^{-/-} HeLa cells after siRNA-mediated depletion of TIMM17A. Mann-Whitney U-test was performed. Quantile box plot shows median (n=5, biologically independent samples).

Reviewer #1:

Remarks to the Author:

The authors addressed the concerns and comments more than adequately. This is an exciting study of the high interest.

We thank the reviewer for the positive view on our findings.

Reviewer #2:

Remarks to the Author:

The authors have answered my comments sufficiently and have strengthened the manuscript significantly. I congratulate them on their interesting story and manuscript.

We thank the reviewer for the positive view on our findings.

Reviewer #3:

Remarks to the Author:

The authors have adequately addressed my comments and improved the quality of the manuscript. However, the presentation of the new import results (log₂FC of signal at 0h) is unusual and appears contrived; the import signal after zero incubation time should be close to the range of the noise and so it is unclear why it was chosen as the base for all calculations. Also, presumably each curve was normalized to its own 0h signal so the relative amounts of proteins imported in the two strains are unknown. The authors should include actual import results or graphs calculated in a more standard manner.

With this small correction, the manuscript is ready for publication.

To comply with the request of the reviewer, we follow the field's standard representation and have set the average value (three independent biological replicates) of the last timepoint of the wild type scr to 100% and scaled the other data points of scr and siJC15 accordingly.

We have updated the Figure 4 d and e and append the updated figure panels below.

d

e